# Personalized genome assembly for accurate cancer somatic mutation discovery using tumor-normal paired reference samples

Chunlin Xiao[1*] , Zhong Chen[2], Wanqiu Chen[2], Cory Padilla[3], Michael Colgan[4], Wenjun Wu[5], Li-Tai Fang[6], Tiantian Liu[2], Yibin Yang[5], Valerie Schneider[1], Charles Wang[2*] and Wenming Xiao[4*]

*Correspondence:
xiao2@mail.nih.gov; chwang@llu.
edu; Wenming.Xiao@fda.hhs.gov

[1] National Center
for Biotechnology Information,
National Library of Medicine,
National Institutes of Health,
45 Center Drive, Bethesda, MD
20894, USA
[2] Center for Genomics, Loma
Linda University School
of Medicine, 11021 Campus St.,
Loma Linda, CA 92350, USA
[3] Dovetail Genomics, 100
Enterprise Way, Scotts Valley, CA
95066, USA
[4] The Center for Drug Evaluation
and Research, U.S. Food
and Drug Administration, Silver
Spring, MD, USA
[5] Blood Cell Development
and Function Program, Fox Chase
Cancer Center, Philadelphia, PA
19111, USA
[6] Bioinformatics Research &
Early Development, Roche
Sequencing Solutions Inc., 1301
Shoreway Road, Belmont, CA
94002, USA

## Abstract

**Background:** The use of a personalized haplotype-specific genome assembly, rather than an unrelated, mosaic genome like GRCh38, as a reference for detecting the full spectrum of somatic events from cancers has long been advocated but has never been explored in tumor-normal paired samples. Here, we provide the first demonstrated use of de novo assembled personalized genome as a reference for cancer mutation detection and quantifying the effects of the reference genomes on the accuracy of somatic mutation detection.

**Results:** We generate de novo assemblies of the first tumor-normal paired genomes, both nuclear and mitochondrial, derived from the same individual with triple negative breast cancer. The personalized genome was chromosomal scale, haplotype phased, and annotated. We demonstrate that it provides individual specific haplotypes for complex regions and medically relevant genes. We illustrate that the personalized genome reference not only improves read alignments for both short-read and long-read sequencing data but also ameliorates the detection accuracy of somatic SNVs and SVs. We identify the equivalent somatic mutation calls between two genome references and uncover novel somatic mutations only when personalized genome assembly is used as a reference.

**Conclusions:** Our findings demonstrate that use of a personalized genome with individual-specific haplotypes is essential for accurate detection of the full spectrum of somatic mutations in the paired tumor-normal samples. The unique resource and methodology established in this study will be beneficial to the development of precision oncology medicine not only for breast cancer, but also for other cancers.

## Background

Accurately detecting somatic mutations and subsequently understanding genomic instability in cancer are critical for precision cancer therapies [1–4]. Many genomics studies, including tremendous efforts from the well-known TCGA and ICGC consortia,

have greatly improved our understanding of genomic instability of cancer and cancer biology in general [3, 4]. Most recently, paired tumor-normal reference samples [5–7] and reference call sets were established by the Sequencing Quality Control-2 (SEQC-2) consortium for benchmarking somatic mutation detections using different sequencing platforms and bioinformatic analysis methods [5, 6]. These studies have provided a critically important resource, not just to cancer biology in general, but for assessing the accuracy and reproducibility of somatic mutation detection in cancer diagnostics, designing personalized cancer immunotherapies, and for analyzing potential off-target effects that may interfere with successful therapies based on gene editing.

To date, discovering somatic events and defining high-confidence reference somatic call sets rely mainly on a standard human reference assembly (such as GRCh38) as a benchmark for sequence analysis. However, GRCh38 has its own limitations. Despite its high quality, it remains incomplete due to some unresolved assembly issues and persistent gaps, including those at centromeres, telomeres, and heterochromatic regions [8–11]. Incorrect or missing sequences in the GRCh38 reference assembly may lead to failed or spurious read mapping and unreliable subsequent analysis results, namely reference bias [12]. Moreover, the human reference assembly was constructed based on DNAs derived from multiple individuals, though approximately 70% of the GRCh38 sequences were contributed by a single African-European admixed male (RP11) [9]. Such mosaic haplotype representation in the reference assembly may complicate the identification of somatic variants from cancer samples. Therefore, use of a de novo assembly of a personalized genome rather than the standard reference assembly for confident cancer mutation discovery has been advocated [8, 10, 13], because there is more to be learned from direct comparison of the tumor genome to the normal genome from which it is derived, than to an unrelated, random, mosaic genome like GRCh38.

Recent advancements in DNA sequencing technologies provide an extraordinary opportunity to generate a high-quality de novo assembly for an individual genome at affordable cost. Specifically, the breakthrough of long-range DNA sequencing from next-generation sequencing technologies now makes it possible to accurately assemble individual genomes to near completion, as has been done for several samples, including HX1 [14], AK1 [15], NA12878 [16], CHM13 [17], and HG002 [18]. These studies demonstrated the utilities of these recent advancements to genome assembly methods and subsequent germline variant detection. However, there has been no systematic investigation of the use of personalized genomes as references for somatic mutation detection, particularly in paired tumor-normal samples. The HCC1395 breast cancer cell line and a matched B lymphocyte cell line HCC1395BL [19], derived from the same individual, are one of the most important tumor-normal models for triple negative breast cancers (TNBC), and represent reference samples that have been characterized extensively by previous studies through the SEQC2 consortium [5–7]. Here we combined multiple sequencing technologies, including Illumina short reads, 10X Genomics linked reads, PacBio long reads, and Hi-C (high-throughput chromosome conformation capture) reads, to reconstruct what, to our knowledge, is the first direct comparison of paired tumor-normal genomes [10]. This study enables us to assess the qualities of the de novo assembled personalized genome, to evaluate the performance of somatic mutation detection comprehensively with respect to the underlying reference assemblies being

used, and to interrogate the complete spectrum of genomic alterations more accurately, using a personal genome as the reference.

## Results

### Overall study design and construction of a reference-grade personal genome assembly

Sequencing data from five different platforms were used to assemble genome for the normal reference sample (HCC1395BL B Lymphocyte cell line), and data from three platforms were used for the tumor reference sample (HCC1395 breast cancer cell line from the same donor) (Fig. 1, top panel). Using data from multiple sequencing technologies, including short reads, linked reads, and long reads (Additional file 1: Table S1), we built a workflow to generate a de novo assembled personal genome (Fig. 1, middle panel), known as HCC1395BL_v1.0. We then used this assembled genome and GRCh38 as references for read mapping and somatic variant analyses (Fig. 1, bottom panel).

In this workflow, we first generated two initial assemblies for HCC1395BL, using canu [20] with PacBio long reads and Supernova [21] with 10X Genomics linked reads, respectively. In contrast to the PacBio canu assembly, the Supernova assembly contained many small contigs (< 10 kb) (Additional file 1: Table S2). Although the N50s

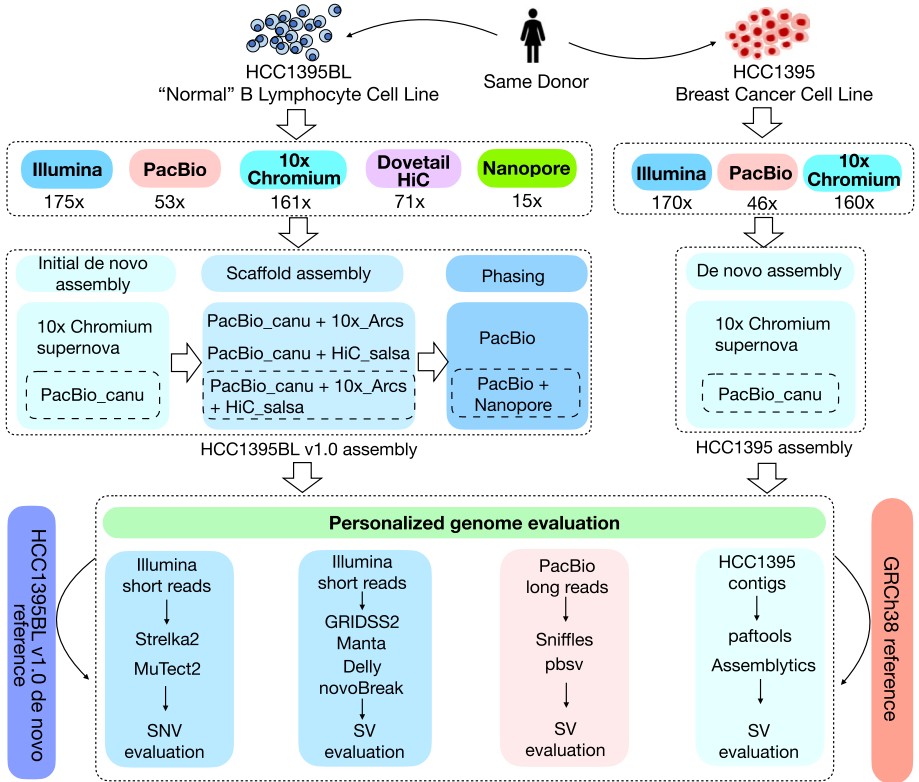

**Fig. 1** Schematic diagram of study design. Sequencing data from five different platforms were used for initial de novo assembly, assembly evaluation, scaffolding, and phasing for the normal reference sample (HCC1395BL B Lymphocyte cell line), while sequencing data from three platforms were used for de novo assembly and assembly evaluation for the tumor reference sample (HCC1395 breast cancer cell line from the same donor). The final assembled personal genome, known as HCC1395BL_v1.0, was used as reference for read mapping with both short and long reads, and assessment of somatic SNVs and SVs as compared to that using GRCh38 as reference

and the largest scaffold of the Supernova assembly were much larger than the PacBio contig assembly (Additional file 2: Figs. S1 and S2), the latter was more complete as measured via Benchmarking Universal Single-Copy Orthologue (BUSCO) genes (Additional file 2: Fig. S3). Additionally, a greater number of complete RefSeq protein-coding genes mapped to the PacBio assembly, and more base pairs from this assembly could be mapped to the GRCh38 reference (Additional file 1: Table S2). Taken together, these findings indicated that the overall quality of the PacBio canu assembly was higher than that of the Supernova assembly, particularly when gene content and completeness were the primary concerns.

We selected the PacBio contig assembly of HCC1395BL for further scaffolding. In general, two steps of scaffolding, using 10X Genomics linked reads with ARCS followed by Hi-C reads with SALSA (PacBio_canu + ARCS + SALSA), produced a better scaffolded assembly than using one-step scaffolding only (either PacBio_canu + ARCS or PacBio_canu + SALSA). The final scaffold assembly (hereafter referred to as HCC1395BL_v1.0) was the one with the highest Top50 (see "Methods") and scaffold N50 values, the largest scaffold size, and the greatest numbers of mapped complete BUSCOs and RefSeq transcripts (Table 1). HCC1395BL_v1.0 consisted of 1645 scaffolds totaling 2.9 Gb, of which 2.69 Gb (92.62%) were from Top50 scaffolds, with a scaffold N50 size of 69.97 Mb, in comparison to scaffold N50s of 67.79 Mb for GRCh38 [9] and 44.84 Mb for AK1, a recent assembly from a diploid sample [15], respectively. Both HCC1395BL_v1.0 and AK1 had a very similar assembly size (2.90 Gb), but HCC1395BL_v1.0 had fewer scaffolds (1,645 vs. 2,832), a smaller L50 (14 vs. 21), and a much greater Top50 (2.69 Gb vs. 2.26 Gb), N50 (69.97 Mb vs. 44.84 Mb), and the largest scaffold size (181.21Mb vs. 113.92 Mb). Moreover, the HCC1395BL_v1.0 assembly contained more complete RefSeq NM (protein-coding) transcripts (49,613 vs. 49,432) and RefSeq NR (non-protein-coding) transcripts (15,227 vs. 15,089).

Consistency analysis (see "Methods") with the GRCh38 primary assembly (alternate loci excluded) showed that five chromosomes (chr4, chr8, chr14, chr18, and chr20) were almost completely covered by single scaffolds. The largest HCC1395BL_v1.0 scaffold (Scaffold_1 181.21Mb) covered more than 95% of GRCh38 chromosome 4. Four chromosomes (chr2, chr3, chr12, and chr19) were broken only in centromeric regions. Several other chromosomal arms (chr1p, chr5p, chr6q, chr9p, chr10p, chr21q, and chrXq) were also covered by single HCC1395BL_v1.0 scaffolds (Fig. 2).

Phasing analysis showed that 3.13 out of 3.17 million heterozygous sites were considered phased, and 6368 phased blocks accounted for 2.42 Gb of HCC1395BL_v1.0. The longest phased block was 6.37 Mb (Additional file 1: Table S3). Approximately 15-fold coverage of Nanopore long reads was used to further extend phasing to 2.54 Gb. The total number of phased blocks was subsequently decreased to 3204 from 6368 blocks, and the longest phased block was greatly improved, increasing from 6.37 to 20.45 Mb (Additional file 1: Table S3). With the phased assemblies (haplotype1 and haplotype2) for the HCC1395BL cell line, we were able to call 4,115,622 germline SNVs in diploid regions of autosomal chromosomes using dipcall [22].

For a comparison, we also generated de novo assemblies for the HCC1395 cancer cell line, using canu with PacBio long reads and Supernova with 10X Genomics' linked reads, respectively. The resulting HCC1395 assembly was more fragmented than the

**Table 1** Summary of quality assessments of assemblies from different scaffolding strategies. The PacBio contig assembly of HCC1395BL was selected for further scaffolding. Generally, two steps of scaffolding, using 10X Genomics linked reads with ARCS followed by Hi-C reads with SALSA (PacBio_canu + ARCS + SALSA), produced a better scaffolded assembly than using one-step scaffolding only (either PacBio_canu + ARCS or PacBio_canu + SALSA). The final scaffold assembly (HCC1395BL_v1.0) is the one with the highest Top50 and scaffold N50 values, the largest scaffold size, and the greatest numbers of mapped complete BUSCOs and RefSeq transcripts

| | # Scaffolds | # bp from scaffolds | Top50 | N50 | L50 | Largest scaffold length (bp) | # Scaffolds on GRCh38 | # novel scaffolds | # bp from novel scaffolds | # Complete BUSCO (4,104 BUSCO) | # NMs 95+% mapped (50,052 NMs) | # NRs 95+% mapped (15,544 NRs) |
|---|---|---|---|---|---|---|---|---|---|---|---|---|
| **PB_canu (contigs)** | 2828 | 2,904,842,414 | 1,356,447,278 (46.69%) | 13,480,407 | 57 | 62,208,403 | 2,526 | 302 | 16,403,702 | 3890 | 49,287 | 15,115 |
| **PB_canu + 10X_arcs (scaffolds)** | 2032 | 2,904,931,213 | 2,067,627,443 (71.17%) | 35,058,531 | 26 | 121,623,092 | 1,764 | 268 | 14,867,391 | 3800 | 49,570 | 15,207 |
| **PB_canu + HiC_salsa (scaffolds)** | 1891 | 2,905,381,691 | 2,377,981,926 (81.84%) | 46,871,224 | 19 | 180,772,639 | 1,617 | 274 | 15,303,825 | 3892 | 49,495 | 15,177 |
| **PB_canu + 10X_arcs + HiC_salsa (scaffolds, HCC1395BL_ v1.0)** | 1645 | 2,905,196,510 | 2,691,295,119 (92.62%) | 69,970,292 | 14 | 181,209,810 | 1,406 | 244 | 14,104,388 | 3889 | 49,613 | 15,227 |

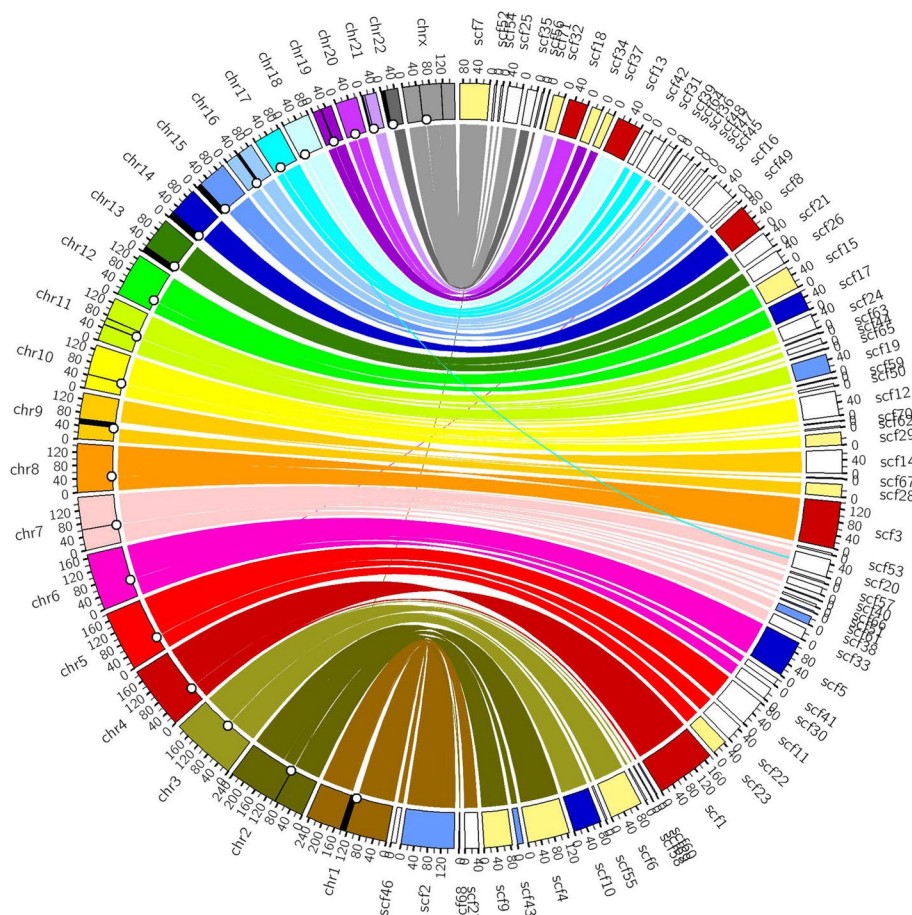

**Fig. 2** A Circos consistency plot of HCC1395BL_v1.0 (right side) against the GRCh38 reference (left side). Included are the 71 largest scaffolds with at least 2 Mbp, which accounted for 2,775,074,314 bp (95.51%). Shown here were alignments with coverage of at least 100kb and mapping quality of at least 60 on GRCh38 using minimap2. Centromeres are marked with circles on the inner circle of GRCh38 chromosomes. Black regions on the chromosomes represent GRCh38 gaps 100kb greater in size. Five chromosomes were almost completely covered by single scaffolds (Scaffold_1 for chr4, Scaffold_3 for chr8, Scaffold_8 for chr14, Scaffold_13 for chr18, and Scaffold_18 for chr20, and are colored red). Four chromosomes (chr2, chr3, chr12, and chr19) were broken only at centromere regions (covered almost completely by just two scaffolds). Centromere-crossing scaffolds are colored light blue. Scaffolds (Scaffold_5, Scaffold_10, and Scaffold_17) covering one arm are colored dark blue. Scaffolds with near full coverage of one arm are colored yellow

HCC1395BL assembly (Additional file 2: Figs. S1 and S2; Additional file 1: Table S2), as demonstrated by the cytogenetic analysis of HCC1395 and HCC1395BL cell lines [6]. The N50s of HCC1395 assemblies were significantly smaller than that of the corresponding HCC1395BL assemblies (Additional file 2: Fig. S1; Additional file 1: Table S2). Moreover, we identified smaller numbers of BUSCO genes (Additional file 2: Fig. S3), RefSeq protein-coding and non-protein-coding transcripts (Additional file 1: Table S2) on both the PacBio canu, and 10X Genomic Supernova assemblies of HCC1395 tumor cell line, as compared to the corresponding assemblies from the HCC1395BL normal cell line. Similarly, the PacBio assembly of the HCC1395 cell line was also shown to be of better quality in terms of gene content and completeness, as compared with that of 10X Genomics' Supernova assembly (Additional file 2: Fig. S3; Additional file 1: Table S2).

A previous study [23] showed that mitochondrial genome assemblies in recently published de novo assemblies have either been absent or highly fragmented. In this study, we confirmed that we completely assembled the mitochondrial genome into a single contig using PacBio data in both the normal and tumor cell lines. Direct comparison of these two mitochondrial genomes (see "Methods") revealed two nonsynonymous somatic mutations (T4813C and C4938A) in the MT-ND2 gene, and one nonsynonymous somatic mutation (G14249A) in the MT-ND6 gene (Additional file 2: Fig. S7).

### Personal genome assembly provides easily accessed sample-specific haplotypes for clinically relevant genomic regions

We aligned an NCBI RefSeq transcript set (excluding all pseudogenes and genes from chromosome Y) to HCC1395BL_v1.0. In total, 19,303 of 19,325 (99.89%) RefSeq protein-coding genes could be mapped onto HCC1395BL_v1.0 successfully, with minimum 95% alignment identity and 50% alignment coverage, while 19,164 of 19,303 (99.27%) of these genes aligned with at least 95% coverage (Fig. 3A; Additional file 1: Table S4). Among RefSeq non-protein-coding genes, 10,049 of 10,061 (99.88%) could be aligned to HCC1395BL_v1.0 successfully with minimum 95% identity and 50% coverage; 9958 of 10,049 (99.09%) of those genes were covered at more than 95% in length (Fig. 3A; Additional file 1: Table S4).

We next compared HLA gene family coverage in GRCh38 and HCC1395BL_v1.0, as the known variability in this region makes it likely that haplotypes found in this sample may differ from the haplotype represented in the chromosomes of the traditional reference genome GRCh38. HLA genes are located on the 6p region of chromosome 6, and previous cytogenetic analysis showed that this region from the HCC1395BL cell line was essentially haploid [6]. From the RefSeq gene set, 19 HLA protein-coding genes (25 protein-coding transcripts) are annotated on chromosome 6 of the GRCh38 primary assembly. We successfully identified all the HLA genes and corresponding transcripts in HCC1395BL_v1.0 and found that they are located on a single scaffold (Scaffold_30) aligning at minimum identity of 95% and the minimum alignment coverage of 95% with one exception, HLA-DQA1 (NM_002122.3) gene, aligned at 100% coverage to HCC1395BL_v1.0, but only 92.95% identity. No other mapped location was found for the HLA-DQA1 gene on HCC1395BL_v1.0. Notably, the order of the HLA genes on this scaffold was identical to that on GRCh38 (Fig. 3B). However, the haplotype of HLA-DRB genes between HLA-DRA and HLA-DQA1 is extremely divergent between HCC1395BL_v1.0 and GRCh38 [24]. The haplotype of HLA-DRB in the GRCh38 primary assembly is represented by the HLA-DRB1 and HLA-DRB5 genes (human HLA-DR51 haplotype group [25]), but the HLA-DRB haplotype in HCC1395BL_v1.0 consists of the HLA-DRB1 and HLA-DRB4 genes (human HLA-DR53 haplotype group), which is similar to the HLA-DRB haplotypes represented on the GRCh38.p13 ALT_REF_LOCI_4 scaffold NT_167246.2 and the GRCh38.p13 ALT_REF_LOCI_7 scaffold NT_167249.2. This demonstrated that the de novo assembly and scaffolding of HCC1395BL_v1.0 performed well on hypervariable/complex regions, such as those harboring the HLA genes.

We also evaluated other clinically relevant genes whose only representations in GRCh38 are on alternate locus scaffolds, which are included in the reference to capture population diversity. HCC1395BL_v1.0 included GSTT1 (Glutathione S-transferase

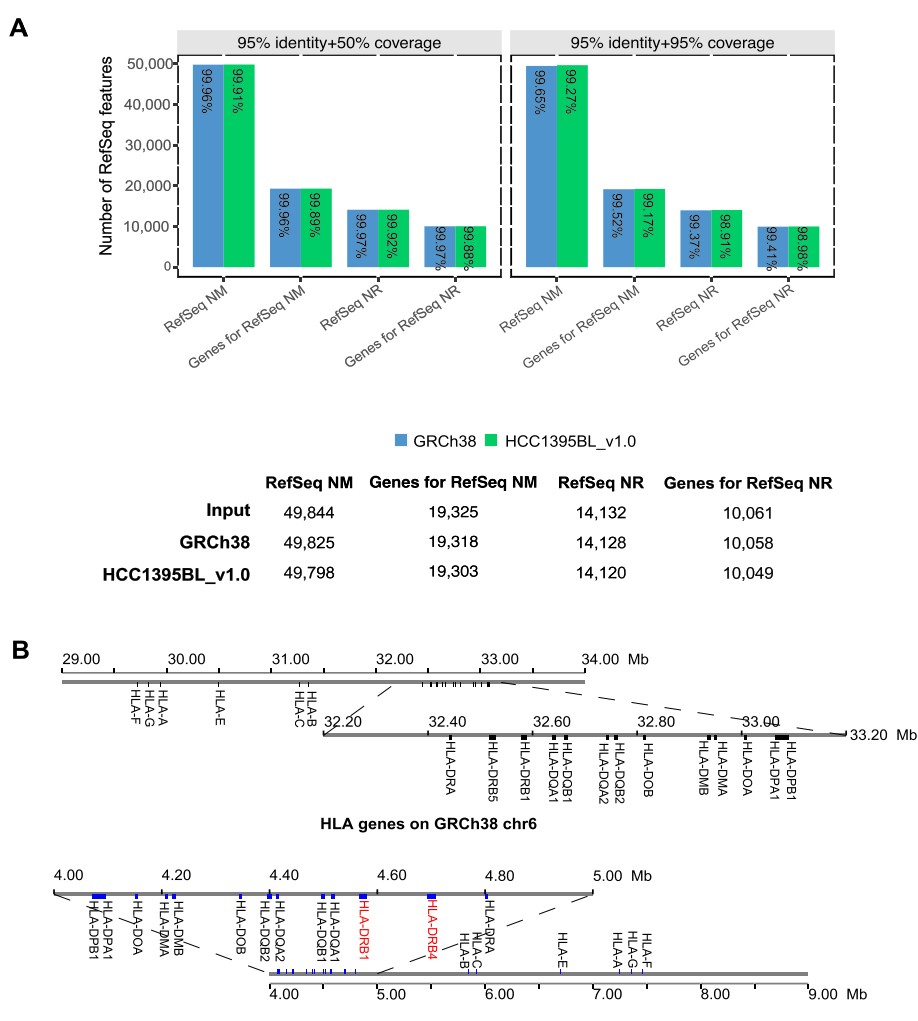

**Fig. 3 A** Summary of RefSeq genes/transcripts mapping on HCC1395BL_v1.0 and GRCh38 with cutoffs 95% identity + 50% coverage versus 95% identity + 95% coverage. The bottom table provides a summary of the mapping using 95% identity + 50% coverage as cutoff. **B** HLA coding genes on Scaffold_30 of HCC1395BL_ v1.0 in comparison to those on chromosome 6 of GRCh38 primary assembly. The haplotype of HLA-DRB (labels in red) in HCC1395BL_v1.0 consists of the HLA-DRB1 and HLA-DRB4 genes (human HLA-DR53 haplotype group), while the GRCh38 primary assembly contains HLA-DRB1 and HLA-DRB5 genes (human HLA-DR51 haplotype group). The human HLA-DR53 haplotype is represented only in GRCh38 ALT_REF_LOCI sequences. Although scaffold_30 maps onto GRCh38 entirely in reverse complement, the HLA gene order is preserved between GRCh38 and HCC1395BL_v1.0

theta 1) and the KIR2DL5A (killer cell immunoglobulin like receptor, which has two Ig domains and long cytoplasmic tail 5A); these two genes are not included in the haplotypes represented on the chromosomes of the GRCh38 primary assembly. GSTT1, a gene previously localized to Chromosome 22 of the GRCh37 primary assembly, is found only on the alternate locus scaffold **NT_187633.1** in GRCh38. Likewise, for the KIR2DL5 gene, the haplotype represented on the GRCh37 Chromosome 19 unlocalized scaffold NT_113949.1 included KIR2DL5A, but in the GRCh38 assembly, it is found only in the alternate locus, such as scaffold NT_113949.2. The changes in the localizations of these genes from GRCh37 to GRCh38 present analysis challenges when switching

between different versions of traditional reference genome. In addition, because these genes are represented only on alternate loci and patch scaffolds in GRCh38, and most existing tool chains do not handle those alternate locus scaffolds, they are consequently more difficult to study. Their exclusion from analysis thus presents a heightened risk for misinterpretation of results. In contrast, as the individual-specific haplotypes for these clinically relevant genes are represented in haplotypes of the personalized assembly, no special handling of alternate loci or patches would be needed to assess them in the tumor genome if HCC1395BL_v1.0 were to be used as reference as opposed to GRCh38.

### Use of a personalized genome as reference improved read mappings for both short and long reads

We compared HCC1395BL_v1.0 and GRCh38 as references for read mappings. While the mapping rates of short reads to the GRCh38 primary assembly (alternate loci scaffolds excluded) and HCC1395BL_v1.0 were very similar for all 12 WGS replicates from 6 sequencing centers for both HCC1395BL and HCC1395 cell lines, we observed overall improved read placements on HCC1395BL_v1.0 as opposed to GRCh38, with some variability in the extent of improvements across these replicates, possibly due to the differences in library preparations and sequencing coverages for paired normal and tumor samples when sequencing was performed in each of the sequencing centers (Fig. 4). For instance, only slightly higher percentages of properly paired reads for both normal and tumor samples were mapped onto HCC1395BL_v1.0 (Fig. 4A), but mappings for non-properly paired reads were reduced by as much as 41.4% for HCC1395BL, and up to 38.2% for HCC1395 (Fig. 4B). Notably, the mismatches for the mapped reads were decreased up to 18.2% for HCC1395BL and up to 16.6% for HCC1395 (Fig. 4C). In addition, read alignments with soft-clipping (without SA tags) were decreased up to 11.7% for HCC1395BL and up to 11.6% for HCC1395 (Additional file 2: Fig. S4A), while read alignments with hard-clipping were down by as much as 32.0% for HCC1395BL and as much as 28.7% for HCC1395 (Additional file 2: Fig. S4B). Moreover, read alignments with split reads (SA tags) were also reduced by up to 31.9% for HCC1395BL and up to 28.8% for HCC1395 (Fig. 4D). Interestingly, we observed that the library insert size standard deviations were 2.76 smaller on average for HCC1395BL and 2.83 smaller on average for HCC1395 when the personal genome HCC1395BL_v1.0 was used as reference (Additional file 2: Fig. S4C), indicating that paired reads were placed more consistently on HCC1395BL_v1.0 than GRCh38. In addition, we observed that the standard deviations of read coverages in alignments were much smaller on HCC1395BL_v1.0 than GRCh38 (Fig. 4E), demonstrating that reads were placed more uniformly on HCC1395BL_v1.0.

We evaluated PacBio long-read mapping onto GRCh38 and HCC1395BL_v1.0 references using minimap2. We observed that the mapping rates of PacBio long reads were slightly higher (1.65% for normal and 2.98% tumor sample) on HCC1395BL_v1.0 than on GRCh38 (Additional file 2: Fig. S4D). The mismatches were approximately 1% lower for both normal and tumor samples on HCC1395BL_v1.0 than on GRCh38 (Additional file 2: Fig. S4D). The non-primary alignments and supplementary alignments were significantly lower on HCC1395BL_v1.0 for both the normal (6.73%, 14.5%) and tumor samples (1.8%, 10.39%) (Additional file 2: Fig. S4D). Furthermore, we also observed that the standard deviations of read coverages in alignments were much smaller on

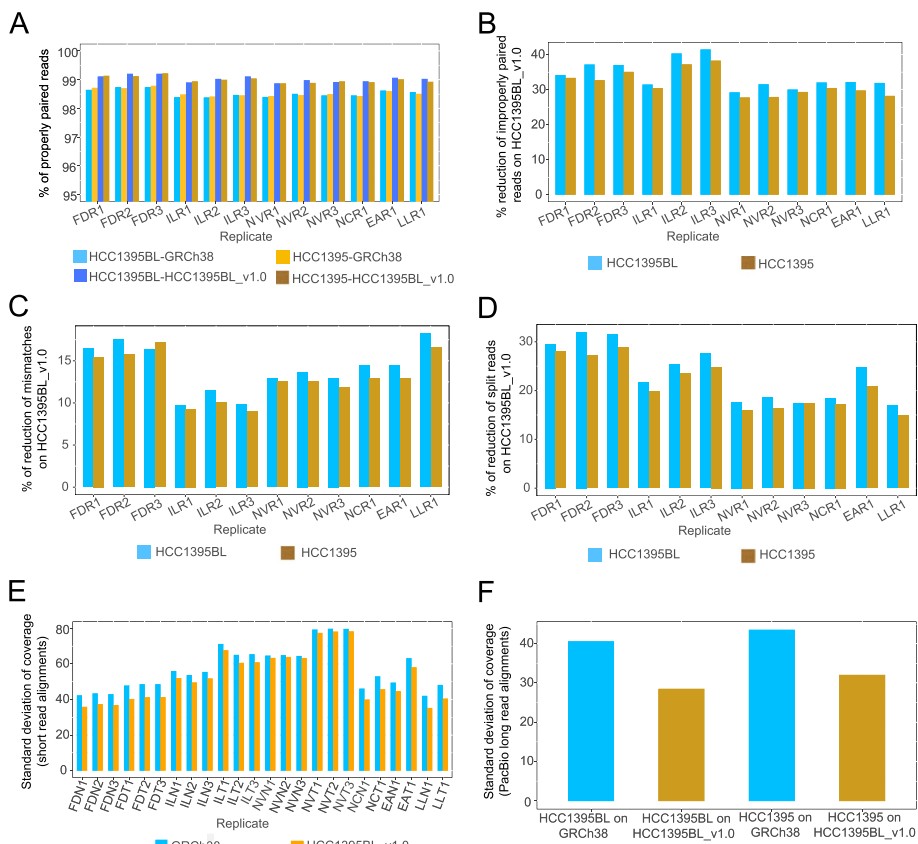

**Fig. 4** Improvements of Illumina short-read and PacBio long-read mappings with personalized genome HCC1395BL_v1.0 reference as compared to GRCh38. **A** Percentages of properly paired reads in alignments for tumor and normal samples. **B** Reductions of non-properly paired reads in alignments for tumor and normal samples with personalized genome HCC1395BL_v1.0 as compared to GRCh38. **C** Reductions of mismatches in alignments for tumor and normal samples with personalized genome HCC1395BL_v1.0 as compared to GRCh38. **D** Reductions of split reads in alignments for tumor and normal samples with personalized genome HCC1395BL_v1.0 as compared to GRCh38. **E** Standard deviations of read coverages in short-read alignments with personalized genome HCC1395BL_v1.0 as compared to GRCh38. **F** Standard deviations of read coverages in PacBio long-read alignments with personalized genome HCC1395BL_v1.0 as compared to GRCh38

HCC1395BL_v1.0 than on GRCh38 (Fig. 4F), indicating PacBio long reads were also placed more uniformly on HCC1395BL_v1.0 than GRCh38. Taken together, these ameliorations in both short-read and long-read mappings provide the important signals indicating that alignment-based somatic mutation discovery will be improved when a personal genome is used as the reference for a paired tumor sample.

## Detection of somatic SNV mutations with short reads using a personalized genome as reference as compared to GRCh38

Based on a previous study [26] and recent SEQC2 reports [5, 6], two commonly used somatic mutation callers, Strelka2 [27] and MuTect2 [28], were selected to generate reports of somatic SNVs and small indels with the same settings based on the same set of Illumina short-read data using HCC1395BL_v1.0 and GRCh38 (alternate loci excluded) as reference genomes, respectively. We initially analyzed a pair of pooled sequencing

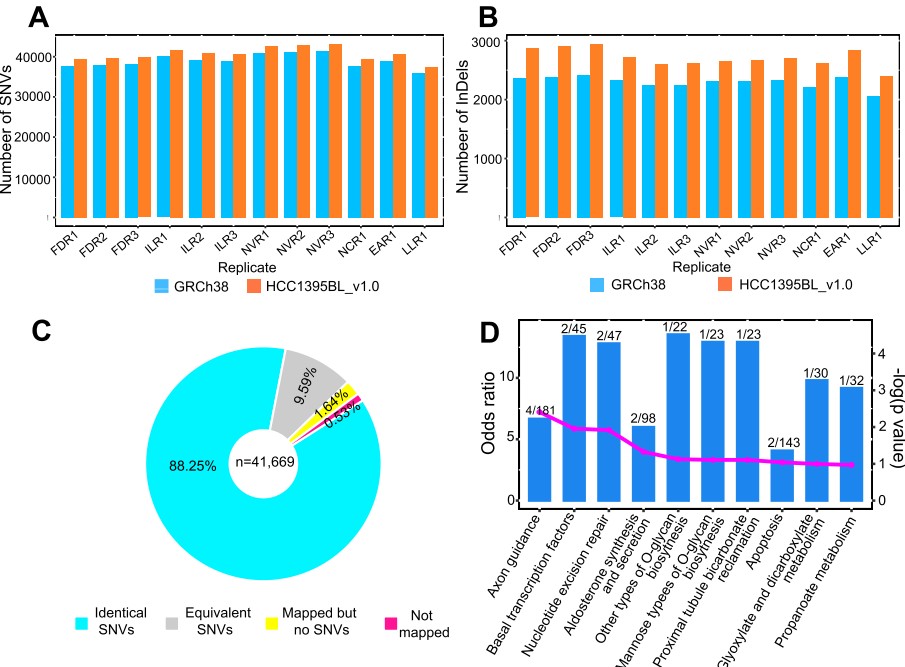

**Fig. 5** Somatic SNV detection using short reads on GRCh38 and HCC1395BL_v1.0 references. **A** Higher numbers of overlapping SNVs between MuTect2 and Strelka2 detected from 12 paired tumor-normal replicates on HCC1395BL_v1.0 as reference as opposed to GRCh38. **B** Higher number of overlapping INDELs between MuTect2 and Strelka2 detected from 12 paired tumor-normal replicates on HCC1395BL_v1.0 as reference as opposed to GRCh38. **C** 40,768 (97.83%) of 41,669 GRCh38-based somatic SNVs were considered mapped with HCC1395BL_v1.0-based SNVs, including 36,773 (88.25%) identical SNVs and 3995 (9.59%) equivalent SNVs. In total, 682 SNVs (1.64%) were able to map onto HCC1395BL_v1.0 but without overlapping Strelka2/MuTect2 calls. A total of 219 SNVs (0.53%) were considered as "not-mapped" onto HCC1395BL_v1.0 due to the stringent mapping criteria. **D** KEGG pathway enrichment analysis of 71 genes overlapped with the 1017 novel SNVs detected with HCC1395BL_v1.0 as a reference. Shown here are the top 10 enriched pathways with bar representing "odds ratio" (zScore) on the left side *y*-axis, dotted-line representing −log (*p*-value) on right side *y*-axis, and the numeric label showing counts of enriched gene versus the total genes in each pathway

data (FDN123 as normal and FDT123 as tumor, see "Methods") from one sequencing center and found that more overlapping calls (1983 more somatic SNVs/indels) between Strelka2 and MuTect2 could be detected on HCC1395BL_v1.0 than GRCh38 (Additional file 1: Table S5). This trend was retained when we expanded to analyze all 12 paired WGS replicates from 6 sequencing centers [5, 6]. On average, 1689 more overlapping somatic SNVs (Fig. 5A) and 415 more overlapping somatic indels were seen on HCC1395BL_v1.0 than GRCh38 (Fig. 5B), due to the various improvements in short-read mappings on HCC1395BL_v1.0 reference as we have already demonstrated.

Among 41,669 GRCh38-based somatic SNVs supported by both Strelka2 and MuTect2 callers (Additional file 1: Table S5), 40,768 SNVs (97.83%) were successfully mapped onto HCC1395BL_v1.0 with overlapping SNVs called by Strelka2/MuTect2 (Fig. 5C; Additional file 1: Table S6). An additional 682 SNVs (1.64%) were mapped on HCC1395BL_v1.0, but without overlapping Strelka2/MuTect2 calls, suggesting that these somatic SNVs might be questionable. Variant functional analysis using ANNOVAR [29] showed that 120 of these 682 SNVs were located within genes (Additional file 1: Table S6). Two hundred nineteen SNVs (0.53%) were considered as "not-mapped" on HCC1395BL_v1.0

due to the stringent mapping criteria we used (see "Methods"). Thus, inclusion of these questionable sites in mutation analysis would certainly cause misinterpretations. Moreover, 3995 SNVs (9.58%) were considered equivalent between GRCh38 and HCC1395BL_v1.0 (Additional file 1: Table S6), but with germline SNVs in their flanking sequences (Additional file 2: Fig. S5). For example, the same set of reads was found to align (with mapping quality 60) across corresponding intergenic SNV regions in HCC1395BL_v1.0 (scaffold_2:131886469-131886569 for SNV scaffold_2:131886519), and GRCh38 (chr1:177753949-177754049 for SNV chr1:177753999), but two additional homozygous germline SNVs were observed in flanking sequences in the latter (Additional file 2: Fig. S5A, B, G). Similar examples were found for an exonic SNV at scaffold_37: 17305121 or chr19:17555816 (causing amino acid change in gene COLGALT1) (Additional file 2: Fig. S5C, D) and an intronic SNV at scaffold_12:48083060 or chr10:114357477 (Additional file 2: Fig. S5E, F). Such discrepancies reflect the underlying genomic sequence differences between the personalized HCC1395BL_v1.0 and the common reference GRCh38, and illustrate the importance of using a personal genome for accurate somatic mutation discovery and subsequent analysis. For example, mismatches in allele-specific probes or primers would affect melting temperature and binding efficiency when they are used for validation of a SNP genotyping assay.

Among the 43,285 somatic SNVs supported by both Strelka2 and MuTect2 on HCC1395BL_v1.0 (Additional file 1: Table S5), 2790 SNVs were identified that lacked equivalent GRCh38-based SNVs. Among them, 1017 sites were well-supported by more than 10 alternate allele reads with the percentage of alternate allele read coverage at least 50%. By co-locating these SNVs with RefSeq genes and transcripts mapped onto HCC1395BL_v1.0, 522 of 2790 SNVs were found within genes, while 177 of well-supported 1017 SNV subset were located in 71 gene regions. KEGG pathway enrichment analysis suggested some of these 71 genes were involved with important pathways (Fig. 5D). For example, GTF2H2 (general transcription factor IIH subunit 2) encodes the subunit of RNA polymerase II transcription initiation factor IIH, which is involved in both basal transcription and nucleotide excision repair (Additional file 1: Table S7). PTPN13 (protein tyrosine phosphatase non-receptor type 13) encodes a signaling molecule that belongs to the protein tyrosine phosphatase (PTP) family, which regulates a variety of cellular processes such as cell growth, differentiation, mitotic cycle, and oncogenic transformation (Additional file 1: Table S7).

To demonstrate the validity of somatic mutations identified only on HCC1395BL_v1.0, we performed Sanger sequencing on a subset of SNVs in these 71 gene regions (177 SNVs). Eight of ten selected sites were confirmed as somatic SNVs (Additional file 2: Fig. S6; Additional file 1: Table S8), while one SNV site (scaffold_20:687304 with MAF=0.558) showed the mutation in both tumor and normal samples, and the other one (scaffold_19:2641776 with MAF=0.5) showed no point mutation (Additional file 1: Table S8). While a high validation rate (80%) was achieved by Sanger sequencing, the results may indicate that some artifacts might exist in novel SNVs identified only with HCC1395BL_v1.0.

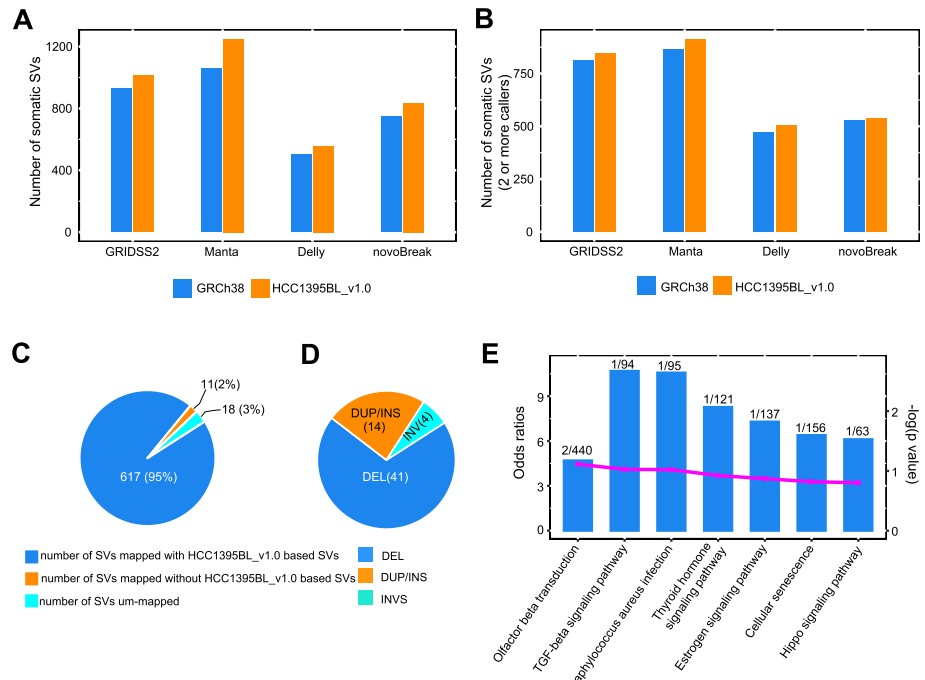

**Fig. 6** Summary of somatic SV detections using tumor-normal paired short-read WGS data with HCC1395BL_v1.0 reference as compared to GRCh38. **A** Somatic SV counts discovered by GRIDSS2, Manta, Delly, and novoBreak with HCC1395BL_v1.0 reference as compared to GRCh38. **B** Somatic SV counts discovered by two or more somatic SV callers with HCC1395BL_v1.0 reference as compared to GRCh38. **C** 617 of 646 GRCh38-based somatic SVs (TRA excluded) were mapped to HCC1395BL_v1.0-based SVs, while 18 of 646 SVs were "unmapped," and 11 of 646 SVs were mapped on HCC1395BL_v1.0, but no SVs in mapped locations on HCC1395BL_v1.0. **D** Somatic SVs supported by two or more callers were without mapped GRCh38-based SVs on HCC1395BL_v1.0. **E** KEGG pathway enrichment analysis of 17 genes overlapped with the 17 somatic SVs detected with HCC1395BL_v1.0 as a reference. Shown here are the top 7 enriched pathways with bar representing "odds ratio" (zScore) on the left side *y*-axis, dotted-line representing −log (*p*-value) on right side *y*-axis, and the numeric label showing counts of enriched gene versus the total genes in each pathway

## Detection of somatic SVs with short reads using a personalized genome as reference as compared to GRCh38

Due to differences in underlying algorithms for predicting somatic SVs, reported SV events called by different tools can vary widely in terms of event numbers, event types, and event sizes [30, 31]. Thus, we initially limited our somatic SV analysis for short-read sequencing to data generated by one sequencing center (FD) only (see "Methods"). For short-read WGS sequencing data, we selected four somatic SV callers, including GRIDSS2/GRIPSS [32], Manta [33], Delly [34], and novoBreak [35] as representatives for different detection algorithms, to evaluate their relative performances with HCC1395BL_v1.0 reference as compared to GRCh38. Both GRIDSS2 and Manta use split-read, read-pair, and breakpoint assembly approaches for somatic SV detection, whereas Delly uses read-pair first and then split-read information for SV detection and refinement, and novoBreak uses local assembly of associated read pairs with tumor-specific k-mers for somatic SV identification. As expected, all callers predicted various numbers of somatic SVs on both HCC1395BL_v1.0 reference and GRCh38, but in general, all four callers reported more somatic calls on HCC1395BL_v1.0 reference, whether translocation calls (TRA) were included (Fig. 6A; Additional file 1: Table S9) or excluded (Additional

file 2: Fig. S8A; Additional file 1: Table S9), as compared to GRCh38. The increases were observed in all SV types except DUP and INV calls by novoBreak (Additional file 2: Fig. S8B). We also observed that the total SV counts detected by both GRIDSS2 and Manta were higher than that by Delly and novoBreak, indicating that the callers with more sophisticated algorithms (such as GRIDSS2 and Manta that combined split-read, read-pair, and breakpoint assembly approaches) were likely more sensitive than relatively simpler callers (such as Delly and novoBreak) (Fig. 6A). Particularly, Delly reported far fewer DUP and TRA events than any of other callers, suggesting Delly may have lower sensitivity for detecting these two SV types (Additional file 2: Fig. S8B). Additionally, GRIDSS2 reported 2 insertions (scaffold_47:144105:INS:66bp and scaffold_49:9684521:INS:60bp called by GRIDSS/Manta/Delly, but not in the gene region), and Delly reported 3 insertions (scaffold_47:144105:INS:66bp and scaffold_49:9684521:INS:60bp called by GRIDSS/Manta/Delly, and scaffold_129:31804:INS:78bp called by Manta/Delly, not located in the gene region) on HCC1395BL_v1.0, but none on GRCh38, while novo-Break did not report any insertion event. Manta detected 20 insertions on HCC1395BL_v1.0 (one insertion scaffold_1:105582094:INS:52bp overlapping with a DUP scaffold_1:105582094-105582146 by GRIDSS/Manta, intronic region in LIN5 gene), but only 10 insertions on GRCh38 (Additional file 2: Fig. S8B), suggesting that the personalized HCC1395BL_v1.0 as reference may have slightly better sensitivity for somatic insertion detection with short-read data for this pair of samples due to the improvements in personalized genome reference and subsequent read mapping on HCC1395BL_v1.0.

We used a consensus approach so as to define a high-quality somatic SV callset, comprised only of somatic SV calls detected by two or more callers (see "Methods"). With this consensus callset, all four callers still reported more somatic SVs on HCC1395BL_v1.0 reference (Fig. 6B; Additional file 1: Table S9). Breaking down the consensus callset by SV types, 28 (7.25%) more deletions (DEL), 3 (7.89%) more inversions (INV), and 21 (9.29%) more translocations (TRA) were seen on HCC1395BL_v1.0 as compared to GRCh38, while the SV counts for the combination of DUP and INS were largely the same (Additional file 2: Fig. S8C). However, when we mapped those GRCh38-based SVs supported by two or more callers (TRA excluded) onto the HCC1395BL_v1.0 reference, 617 of 646 (95%) SVs (TRA excluded) were localized with HCC1395BL_v1.0-based SVs, but 18 SVs (3%), including 10 DEL, 7 DUP, and 1 INV, were considered "unmapped," while 11 SVs (2%), including 7 DEL and 4 DUP, were mapped but without matching SVs in the mapped locations on HCC1395BL_v1.0 reference (Fig. 6C). Inclusion of these 29 SVs (18 unmapped or 11 mapped-without-matching-SVs on HCC1395BL_v1.0) may cause misinterpretation in downstream analysis. Furthermore, we identified 59 HCC1395BL_v1.0-based SVs supported by two or more callers (including 41 DEL, 14 DUP/INS, and 4 INV) that lacked GRCh38-based SVs in corresponding locations on HCC1395BL_v1.0 with our current mapping criteria (Fig. 6D). By collocating these 59 SVs with RefSeq genes mapped onto HCC1395BL_v1.0, we found 17 SVs (11 DEL, 3 INV, and 3 DUP/INS) overlapped with 17 gene regions (including 7 exon-overlapping SVs). KEGG pathway enrichment analysis suggested that some of these 17 genes were involved in the pathways that may be related to tumor development, including TGF-beta signaling, cellular senescence, and Hippo signaling pathway (Fig. 6E). Manual inspection of read alignments in IGV revealed that 9 out of 11 somatic deletions discovered

in short-read sequencing data could be confirmed with in-read deletions from PacBio long reads generated from the tumor cell line, but not from the normal cell line (Additional file 1: Table S11). For instance, a 311 base pair *SINE/Alu* heterozygous deletion (scaffold_17:32976348-32976659), which overlaps with CCDC91 (coiled-coil domain containing 91), a gene enabling identical protein binding activity, was detected with the HCC1395BL_v1.0 reference, but not with GRCh38 when using the same set of reads, as it lacks this *Alu* sequence in GRCh38 (Additional file 2: Fig. S9A). A 57 base pair homozygous deletion (scaffold_6:44613899-44613956) overlapping with MED12L (mediator complex subunit 12L), a gene involved in transcriptional coactivation of nearly all RNA polymerase II-dependent genes, was detected with the HCC1395BL_v1.0 reference, but analysis on GRCh38 using the same set of the reads finds only a 40 base pair deletion, as the HCC1395BL_v1.0 reference has 17 more adenine (A) nucleotides in this region (Additional file 2: Fig. S9B). These two examples illustrate the importance of using a personalized genome as reference to accurately detect somatic SVs with short-read sequencing data when assessing matched tumor-normal cell lines.

We then extended our analysis to include all 12 tumor-normal paired WGS replicates from 6 sequencing centers [5, 6] so that we could look more deeply into how each of the four callers would be impacted by use of the HCC1395BL_v1.0 reference as compared to GRCh38. For this analysis, we required SVs to be called in at least two replicates for each of the four callers. Similar trends regarding the counts of the somatic calls were seen for each of four callers with GRCh38 and HCC1395BL_v1.0 as references (Fig. 7A; Additional file 1: Table S10; Additional file 2: Fig. S10). For example, the total counts of somatic SVs (particularly for DELs and TRAs) on HCC1395BL_v1.0 were all greater than that on GRCh38 for all callers, and Delly had the lowest counts of somatic SVs when compared with other callers (Fig. 7A; Additional file 2: Fig. S10). The inversion counts were largely similar on two references with just 1 or 2 more inversions on HCC1395BL_v1.0. For DUP, both GRIDSS2 and Manta report 10 (4.44%) and 11 (4.54%) more on HCC1395BL_v1.0, but the DUP counts detected by Delly were unchanged. With regard to insertions, we observed 2 novel ones by GRIDSS2 (scaffold_47:144105:INS:66bp and scaffold_49:9684521:INS:60bp called by GRIDSS/Manta/Delly, but neither located in gene regions), and 4 by Delly (scaffold_47:144105:INS:66bp and scaffold_49:9684521:INS:60bp by GRIDSS/Manta/Delly, and scaffold_129:31804:INS:78bp by Manta/Delly, scaffold_20:21223789:INS:50bp by Delly only, also not located in gene regions) on HCC1395BL_v1.0 reference only, but none by novoBreak. Manta reported 18 insertions (one insertion scaffold_1:105582094:INS:52bp overlapping with a DUP scaffold_1:105582094- 105582146 by GRIDSS/Manta, located in an intronic region in the LIN5 gene) on HCC1395BL_v1.0, but 14 insertions on GRCh38 (Additional file 2: Fig. S10). By mapping these GRCh38-based SVs having support from two or more replicates (TRA excluded) onto HCC1395BL_v1.0 reference, we found that 623, 673, 545, and 556 GRCh38-based SVs by GRIDSS2, Manta, Delly, and novoBreak, respectively, were mapped with HCC1395BL_v1.0-based SVs (Fig. 7B), while 24 SVs by GRIDSS2, 41 SVs by Manta, 43 SVs by Delly, and 28 SVs by novoBreak were considered as "unmapped" or "mapped but without matching SVs on HCC1395BL_v1.0" (Fig. 7C). Meanwhile, when using the HCC1395BL_v1.0 as reference, 61 SVs by GRIDSS2, 86 SVs by Manta, 61 SVs by Delly and 55 SVs by novoBreak with

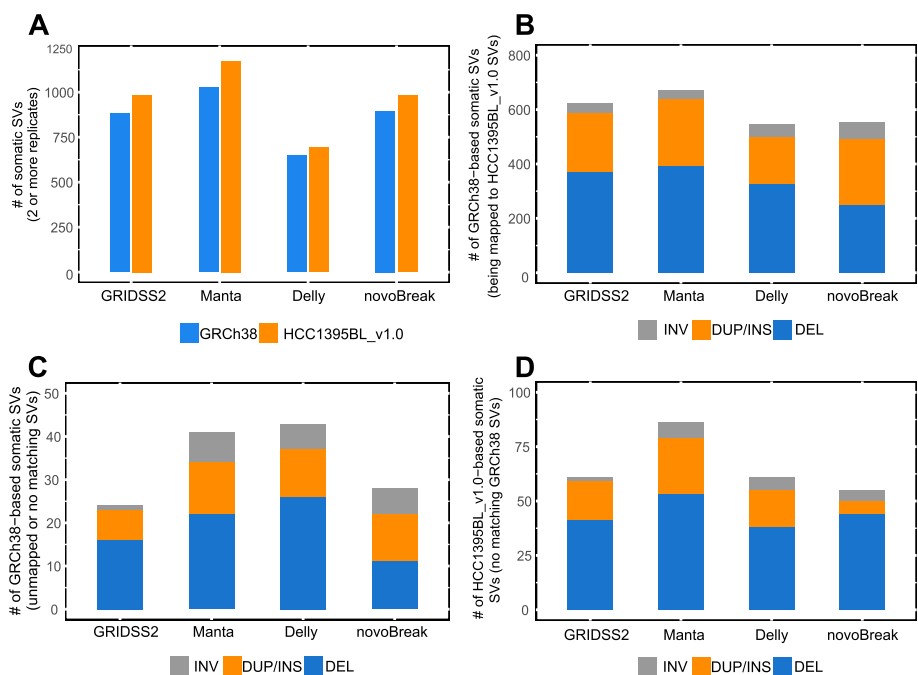

**Fig. 7** Summary of somatic SVs detected in two or more replicates by four short-read callers on HCC1395BL_ v1.0 as compared to GRCh38. **A** Counts of somatic SVs detected in two or more replicates by GRIDSS2, Manta, Delly, and novoBreak on HCC1395BL_v1.0 reference as opposed to GRCh38. **B** Counts of GRCh38-based somatic SVs with support from two or more replicates that were mapped to HCC1395BL_v1.0-based SVs. **C** Counts of GRCh38-based somatic SVs with support from two or more replicates that were unmapped or no matching SVs on the HCC1395BL_v1.0 reference (GRCh38 specific SVs). **D** Counts of HCC1395BL_ v1.0-based somatic SVs with support from two or more replicates that had no mapped GRCh38-based SVs in corresponding locations on HCC1395BL_v1.0 reference (personalized genome-specific SVs)

supports from two or more replicates lacked GRCh38-based SVs at corresponding loca-
tions on HCC1395BL_v1.0 reference under our mapping criteria (Fig. 7D).

### Detection of somatic SVs using long reads and assembled contigs with a personalized genome reference versus GRCh38

Due to the lack of a somatic SV caller that uses both PacBio long reads and assembled
contigs as inputs, somatic SVs were defined as SV calls in tumor cell line (HCC1395)
that were without overlapping germline SV calls in normal (HCC1395BL) cell line in SV
regions (see "Methods"). For each of the six calling methods, including four for PacBio
long reads and two for assembled contigs, we observed that the total counts of somatic
SVs were generally increased with the HCC1395BL_v1.0 reference as compared to
GRCh38 (Additional file 1: Table S12). The initial merged consensus somatic SVs that
had support from two or more calling methods contained an additional 194 SVs (138
DEL, 31 DUP/INS, 3 INV, and 22 TRA) by SV count when HCC1395BL_v1.0 was used
as reference as compared to GRCh38 (Fig. 8A). By mapping the initial GRCh38-based
consensus SVs having support from two or more calling methods (except TRA) to the
HCC1395BL_v1.0 reference, we found 1144 of 1318 SVs (86.8%) were able to localize on
HCC1395BL_v1.0, while 174 SVs (13.2%) were considered "unmapped" on HCC1395BL_
v1.0 with our mapping criteria. Among those that were mapped, 814 SVs had matched

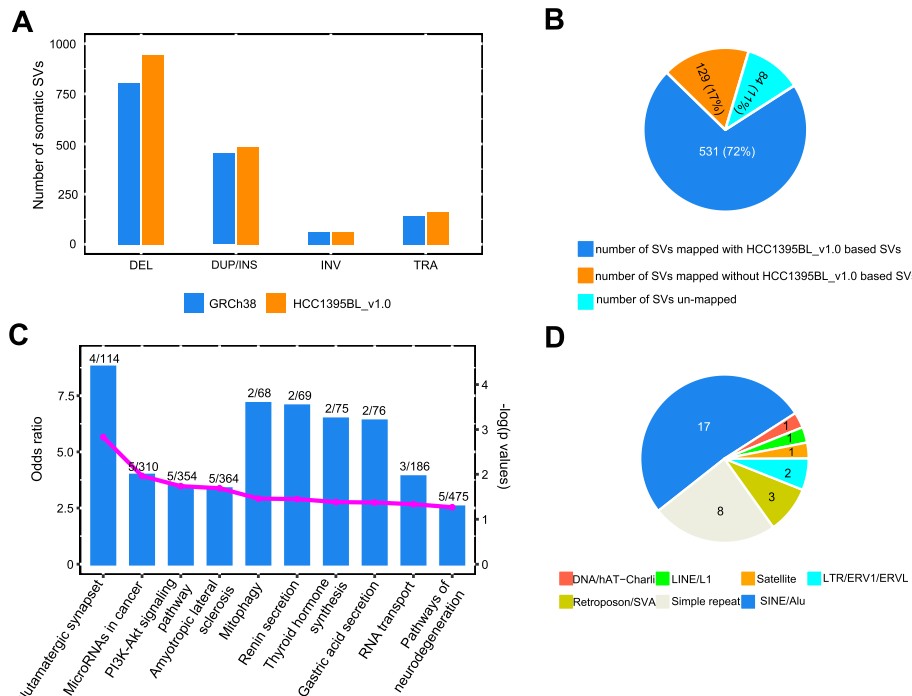

**Fig. 8** Summary of somatic SVs detected in PacBio long-read sequencing data and assembled contigs.
**A** Counts of somatic SVs with supports from two or more calling methods in tumor sample (HCC1395)
using PacBio long reads and assembled contigs on HCC1395BL_v1.0 as compared to GRCh38 references.
**B** Mapping 744 GRCh38-based somatic SVs that were supported by three or more calling methods onto
HCC1395BL_v1.0, 531 SVs were mapped with matched SVs on HCC1395BL_v1.0, and 129 SVs were mapped
but without matching SVs, whereas 84 SVs were considered "unmapped" on HCC1395BL_v1.0. **C** KEGG
pathway enrichment analysis for 86 genes overlapped with 91 novel SVs. Shown here are the top 10 enriched
pathways with "odds ratio" (zScore) and log (*p*-value) on the *y*-axis. The numeric labels are the enriched gene
counts versus the total genes in each pathway. **D** Repeat annotation for the sequences of 72 deletions from
91 SVs that overlapped 86 gene regions using RepeatMasker showed that 32 deletions overlapped 10 classes
of repeat families, among them 17 SINE/Alu, 8 simple repeats, and 3 retroposon/SVA

SVs on HCC1395BL_v1.0, but 330 SVs were mapped without HCC1395BL_v1.0-based
SVs. We observed that more than half of those "unmapped" and "mapped but without
matching SVs" had support from only two calling methods. Thus, to further reduce
potential noisy somatic SV calls from the consensus callset, we subsequently required
somatic SVs to be supported by three or more calling methods. Applying such criteria,
660 of 744 SVs (89%) were mapped onto HCC1395BL_v1.0, and among them, 531 SVs
were mapped with matched SVs on HCC1395BL_v1.0, whereas 129 SVs were mapped
but without matching SVs. Eighty four of 744 SVs (11%) were considered "unmapped"
on HCC1395BL_v1.0 (Fig. 8B). Functional analysis using ANNOVAR showed that 54 of
129 mapped but without matching SVs on HCC1395BL_v1.0 and 24 of 84 unmapped
SVs overlapped with gene regions on GRCh38 (Additional file 1: Table S13). Therefore,
inclusion of these questionable SVs in mutation analysis may cause misinterpretations.
Furthermore, we identified 279 SVs (including 217 DEL, 61 DUP/INS, and 1 INV) on
HCC1395BL_v1.0 lacking corresponding GRCh38-based SVs with our current map-
ping criteria. By collocating those SVs with RefSeq genes and transcripts mapped onto
HCC1395BL_v1.0, we found 91 SVs (72 DEL and 19 DUP/INS) were mapped onto 86
gene regions. KEGG pathway enrichment analysis suggested that some of these 86 genes

were involved with pathways related to cancer invasion and metastasis (e.g., CDH23, ST14), PI3K-Akt signaling pathway, and G protein-coupled receptor signaling pathway (e.g., GNG7) (Fig. 8C; Additional file 1: Table S14). An annotation of the sequences of these 72 deletions by RepeatMasker (https://www.repeatmasker.org/cgi-bin/WEBRe peatMasker) demonstrated that 32 deletions overlapped 10 classes of repeat families, among them 17 SINE/Alu, 8 simple repeats, and 3 retroposon/SVA (Fig. 8D).

We manually curated several of the gene-overlapping deletions (DELs) in IGV and confirmed that these deletions were detected only on HC1395BL_v1.0, and not on GRCh38 with in-read deletions from PacBio long reads (Additional file 2: Fig. S12), demonstrating the importance of using a personalized genome as reference to accurately detect somatic SVs in tumor-normal paired samples. For example, CDH23 (cadherin related 23) belongs to the cadherin superfamily that encodes calcium dependent cell-cell adhesion glycoproteins. This gene has been reported to play a role in early stages of tumor metastasis through regulation of cell-cell adhesion, and upregulation of CDH23 gene may be associated with breast cancer [36, 37]. A 327 bp homozygous deletion (scaffold_12:20762508-20762835), which overlaps CDH23 gene, was uncovered in tumor cell line when HCC1395BL_v1.0 was used as reference, which includes a copy of *SINE/AluY* (284 bp), but this *AluY* sequence is not present in GRCh38. Thus, mapping the same set of tumor reads to GRCh38 would not identify this deletion (Additional file 2: Fig. S12A). Similarly, a 128 base pair homozygous deletion (scaffold_24:40364012-40364140) that overlapped an *LTR/ERVL* repeat was uncovered in tumor cell line when using HCC1395BL_v1.0 reference, but not when using GRCh38. This deletion was located in an intronic region (exon1 and exon2) of ST14 (ST14 transmembrane serine protease matriptase) (Additional file 2: Fig. S12B). Studies have associated the expression of this protease with breast, colon, prostate, and ovarian tumors. Additionally, an intronic 289 base pair homozygous *AluY* deletion (scaffold_20:26715781-26716070, overlapping with ACE gene) (Additional file 2: Fig. S12C), a 538 base pair homozygous deletion (scaffold_37:2324216-2324754, overlapping with GNG7 gene) (Additional file 2: Fig. S12D), and a 1672 base pair homozygous deletion (scaffold_8:42055418-42057090, overlapping with JAG2 gene) (Additional file 2: Fig. S12E) were detected in the tumor cell line when using HCC1395BL_v1.0 reference, but not when using GRCh38.

Combining the aforementioned 6 somatic SV sets (generated by 6 calling methods using PacBio long-read data as well as assembled contigs from the tumor cell line) with 4 somatic SV sets (generated by 4 somatic SV callers using Illumina short-read data), we uncovered 1796 somatic SVs (TRAs excluded) supported by 2 or more calling methods when using the HCC1395BL_v1.0 as reference, which was 201 more somatic SVs detected than when using GRC38 reference (Additional file 2: Fig. S13A, B). Even when applying the stringent requirement that each of the somatic SVs is supported by 3 or more calling methods, 142 more somatic SVs (TRAs excluded) were detected with the HCC1395BL_v1.0 reference as compared to GRCh38 (Additional file 2: Fig. S13A, B). When requiring each of the somatic SVs to be supported by 2 or more calling methods and using the personalized assembly HCC1395BL_v1.0 as reference, 705 somatic SVs were supported by short-read sequencing data, 1381 somatic SVs were supported by PacBio long-read sequencing data, and 802 somatic SVs were supported by assembled contig sequences for the tumor cell line (Additional file 2: Fig. S13C). This observation

indicates that each sequencing technology or data source has its own unique advantages or limitations.

Together with all the evidence we have illustrated in this study, the benefits of using personalized genome assembly as reference are evident. The personalized genome not only comprises individual-specific haplotypes with better representations of the clinically important genes, but also enables better mappings for both short and long reads, and subsequently more accurate identification of somatic mutations in tumor-normal paired samples, when it is used as reference as compared to GRCh38 (Table 2).

## Discussion

In this study, we used a combination of multiple sequencing technologies, including sequencing data consisting of short reads, linked reads, and long reads, to construct the first de novo assemblies of a tumor-normal pair from the same individual with breast cancer. We subsequently used this well-assembled genome as a personal genome reference, in comparison to using the generic human reference GRCh38, for somatic variant detection and demonstrated the advantages of using a personalized genome as a reference.

Our analyses of existing data for HCC1395BL demonstrated that we generated a high-quality assembly in terms of contiguity and gene content, i.e., 99.9% of RefSeq protein-coding genes were successfully mapped onto the personal genome reference with minimum 95% alignment identity and 50% alignment coverage, while the vast majority (99.27%) of these genes were aligned with at least 95% coverage. Complex genomic regions were well-assembled, as evidenced by our demonstration that the complete HLA region, representing an individualized haplotype, is found in a single scaffold for HCC1395BL. Additionally, we found that some clinically relevant genes such as GSTT1 and KIR2DL5 (KIR2DL5A), which are not represented in the chromosomes of the GRCh38 primary assembly, were also captured in our de novo HCC1395BL assembly.

For the first time, we were able to identify cancer somatic mutations based on de novo assembly from the same person, instead of inferring them from the alignments to a mosaic standard reference benchmark such as GRCh38 [13]. Our analysis showed that the de novo assembly improved short-read mapping, resulting in a greater percentage of properly mapped mate-pair reads, reduced total numbers of mismatches and split reads, and many fewer reads with improper-pairing, soft-clipping, or hard-clipping, indicating that short-read mapping was improved with personalized reference genome. In particular, short reads were more uniformly placed on the personal genome reference than GRCh38 as shown by the smaller standard deviations for both read coverages and the library insert sizes. As a result, discovery of somatic SNVs and small indels by different calling algorithms with short reads was more consistent, and more overlapping calls between callers were observed with the personal reference.

Mapping analysis of GRCh38-based somatic SNVs set with flanking regions to the de novo assembled personal reference revealed that only 88.25% somatic SNVs were completely identical to the personal reference-based SNVs, and 9.59% somatic SNVs may have the same reference/alternate alleles on the de novo assembly as on the GRCh38 reference genome, but their flanking sequences may be slightly different with some germline SNVs, highlighting the critical importance of personal genome assembly for

**Table 2** Collective benefits of using the personalized assembly as reference for read mapping and somatic SNV/SV detection as compared to GRCh38

| | | Benefits of using personalized genome assembly as reference | Proofs |
|---|---|---|---|
| Personalized genome (PG) assembly | | • Individualized assembly with inclusion of the individual-specific haplotypes, and better representations for the clinically important genes; no need to deal with ALT loci in NGS secondary analysis | • Fig. 3B, Fig. S5A-G, Fig. S9A/B, Fig. S12A-E; HLA genes, GSTT1, KIR2DL5A |
| Read mapping | Illumina short reads | • More properly paired reads (HCC1395BL 0.5%, HCC1395 0.46%), fewer improperly paired reads (HCC1395BL 41.4%, HCC1395 38.2%)<br>• Fewer mismatches (HCC1395BL 18.2%, HCC1395 16.6%)<br>• Fewer soft-clipped reads (HCC1395BL 11.7%, HCC1395 11.6%), fewer hard-clipped reads (HCC1395BL 32.0%, HCC1395 28.7%)<br>• Fewer split reads (HCC1395BL 31.9%, HCC1395 28.8%)<br>• Better read placements with smaller standard deviations for library insert sizes (HCC1395BL 2.76, HCC1395 2.83)<br>• More uniformly read placements with smaller standard deviations for read coverages (HCC1395BL 4.31, HCC1395 4.92) | • Fig. 4A/B<br>• Fig. 4C<br>• Fig. S4A/B<br>• Fig. 4D<br>• Fig. S4C<br>• Fig. 4E |
| | PacBio long reads | • Higher numbers of reads being mapped (HCC1395BL 1.65%, HCC1395 2.98%)<br>• Fewer mismatches (HCC1395BL 1%, HCC1395 1%)<br>• Lower non-primary/supplementary alignments (HCC1395BL 6.73%/14.5%, HCC1395 1.8%/10.39%)<br>• More uniformly read placements with smaller standard deviations for read coverages (HCC1395BL 12.08, HCC1395 11.48) | • Fig. S4D<br>• Fig. S4D<br>• Fig. S4D<br>• Fig. 4F |
| Somatic SNV detection | Illumina short reads | • Total somatic SNV counts increased by, on average, 1689 SNPs and 415 InDels<br>• Novel SNVs discovered (1017), 177 overlapping with 71 genes, e.g., GTF2H2 and PTPN13, and some were confirmed by Sanger sequencing (8 out of 10 selected SNVs)<br>• Context sequences of somatic SNVs more accurate, some with germline SNVs (3995)<br>• Avoid GRCh38-only, non-personalized SNVs (901) | • Fig. 5A/B, Table S5<br>• Fig. 5D, Tables S6/S7/S8, Fig. S6<br>• Fig. 5C, Table S6, Fig. S5A-G<br>• Fig. 5C |

**Table 2** (continued)

|  |  | Benefits of using personalized genome assembly as reference | Proofs |
|---|---|---|---|
| Somatic SV detection | Illumina short reads | • Somatic SV counts increased by 82/GRIDSS2, 189/Manta, 54/Delly, and 86/novoBreak<br>• Novel SVs discovered (59), including 17 gene-overlapping SVs, e.g., CCDC91<br>• SV resolution more accurate, e.g., SV with MED12L gene<br>• Avoid GRCh38-only, non-personalized SVs (29) | • Fig. 6A, Tables S9/S10, Fig. 7A, Fig. S13A/B<br>• Fig. 6D/E, Table S11, Fig. S9A/B, Fig. 7D<br>• Fig. S9B<br>• Figs. 6C and 7C |
|  | PacBio long reads; assembled contigs | • Somatic SV counts increased by 194 (with supports by 2 or more calling methods)<br>• Novel SVs discovered (279), including 91 gene-overlapping SVs, e.g., CDH23, ST14, GNG7<br>• SV resolution more accurate, e.g., SV with MED12L gene<br>• Avoid GRCh38-only, non-personalized SVs (213) | • Fig. 8A, Fig. S10, Fig. S13A/B<br>• Fig. 8C/D, Table S14, Fig. S12A-E<br>• Fig. S9B<br>• Fig. 8B, Table S13 |

individualized medical research. A small percentage (1.64%) of GRCh38-based SNVs had good mapping locations on the personal genome but did not have corresponding SNV calls, suggesting potential false positives exist.

Our findings indicated that use of a personal genome as reference had impacts on SV discovery using short reads, but the extent of impact on SV calling depended on the SV callers, SV types, and the SV-calling algorithms. If the mappings of underlying supporting reads for potential SVs are improved with the personal reference, the respective SV calls should be improved, and such conspicuous improvements would be reflected in the SV results from these SV callers. Consistent with this assertion, we found with all tested callers that the somatic SV counts detected with use of the personal genome reference were generally higher than that when using traditional GRCh38 reference. In particular, even with short-read sequencing data, the personalized genome reference enabled us to identify additional somatic SVs that could not be detected on the GRCh38 reference due to the absence of certain personalized sequences (e.g., repeats) in the corresponding locations.

The personal genome reference also impacted long-read mapping and contig assembly-to-assembly mapping, as well as subsequent SV detections. Our analysis showed that more reads were mapped with fewer mismatches, and mapped reads were more uniformly placed on the personal genome reference as shown by the smaller standard deviations of read coverages. As a consequence, large SVs detected were more accurate. Most strikingly, germline insertions were significantly reduced using the personal genome as reference, possibly due to systematic collapse of repeats (thus leading to genome-wide deletion bias) in GRCh38 [38]. Such bias would have impacts on germline SV detections with a tendency to call more insertions [17, 39]. Ultimately, this would affect somatic SV calling as well. As demonstrated in our analysis, somatic SV counts were generally greater when the de novo personal assembly was

used as reference as compared to GRCh38. With PacBio long-read sequencing data and assembled contigs mapped onto the personalized genome reference, we uncovered additional somatic SVs only with HCC1395BL_v1.0 reference that could not be detected with GRCh38 reference due to the absence of the personalized sequences at these locations. Not surprisingly, some of these deletions could be easily confirmed with in-read deletions from PacBio reads using IGV and read mappings. Noticeably, some of these additional somatic SVs identified with the personalized genome reference using short- and long-read sequencing data were located in the regions of genes that are involved in pathways related to cancer development and metastasis.

Our approach using a personal assembly (HCC1395BL_v1.0) as reference identified many additional personalized somatic SNVs and SVs which were missed using GRCh38 as reference. These more personalized SNVs/SVs provide additional target choices for patients, researchers or clinicians, and physicians to look into further for personalized patient care. They may reduce the pursuit of incorrect treatment options based on GRCh38-specific or other non-personalized reference somatic SNVs/SVs, and missed opportunities for interventional target-specific treatment options.

As demonstrated in this study, use of a personalized genome as reference for somatic mutation calling in tumor-normal paired samples is promising, but the cost of creating such a personalized genome is still high as compared to using the generic reference. But the goal of scientific research is to find the truth, and this approach will assist in achieving that goal. Moreover, sequencing technologies have evolved rapidly in the last two decades, and the cost of sequencing has decreased substantially. Therefore, it is reasonable to expect that in the near future, the creation of a personalized genome for use as a reference will be cheaper than today.

As sequencing technology continues to advance, longer read length and lower per-base error rate offer great opportunity to tackle many difficult genomic regions, such as telomeres, centromeres, and regions with unplaced/unlocalized sequences, and unfinished gaps as indicated in GRCh38 [9]. Those regions which were previously impossible to assemble, and whose biology is consequently poorly understood, are now within reach [17]. Such new developments should encourage the scientific community to continue improving the quality of the de novo personal assembly for this tumor-normal pair by applying PacBio's HiFi reads and Oxford Nanopore's ultra-long reads in the near future. Such advancement in genome assembly will also provide a better path forward for improving somatic variant identification using a personalized genome as reference. Ultimately, it will lead to discovery of vital genetic markers for cancer diagnosis and therapeutic monitoring, as well as more insights into molecular understanding of tumorigenesis, so that design of personalized immunotherapies, detection of potential off-target effects of gene editing, and other aspects of drug development will be improved.

Recently, the Telomere-to-Telomere (T2T) consortium finished the first gapless telomere-to-telomere human genome assembly (T2T-CHM13) [40] and illustrated its advantages as a reference over GRCh38 for germline variant detection in population genetic analyses [41]. Theoretically, this new reference would improve somatic mutation detection as compared to GRCh38, but the extent of such improvements for somatic mutation discovery in tumor-normal samples has not yet been investigated. Some of the benefits we reported in this study may be impacted. For instance, read mappings to

T2T-CHM13 are anticipated to be better than those to GRCh38, but a personal genome (especially a complete T2T personal genome) reference would still probably outperform T2T-CHM13. Although HCC1395 (https://www.atcc.org/products/crl-2324) is from a Caucasian sample and CHM13 is mostly of European origin [40], there are likely some T2T-CHM13-specific somatic mutations that should be avoided, as well as some personal genome-specific somatic mutations that we would like to use as additional choices for personalized patient care and precision oncology medicine. If the ultimate goal of our patient care is individualized or personalized, then use of a personalized assembly rather than GRCh38 or T2T-CHM13 as reference to identify the full spectrum of somatic mutations in tumor-normal samples is advocated.

## Conclusions

We demonstrated that a personalized genome not only has individual-specific haplotypes that provide better representations of genomic regions in the sample, including clinically relevant genes, but it also enables better alignments for both short and long reads. Consequently, it allows for more accurate detection of somatic mutations, including somatic SNVs and SVs, in paired tumor-normal samples. In particular, novel somatic mutations (SNVs/SVs) were discovered only with a personalized genome as a reference, but not with traditional GRCh38. The unique resource we established in this study will be valuable to the development of precision oncology medicine not only for breast cancer, but also for other cancers.

## Methods

### Whole genome sequencing datasets

A matched tumor/normal pair of cell lines, derived from a TNBC breast cancer (HCC1395) and from normal B cells from the same donor (HCC1395BL), was selected for whole genome sequencing with multiple platforms [5, 6]. We included about 175-fold of Illumina short reads from 3 replicates (FDN1, FDN2, and FDN3) as FDN123 sequenced by Fudan University, 161-fold of 10X Genomics (10X) linked reads, 53-fold of Pacific Bioscience (PacBio) long reads (full-pass subreads, average length 9089 bp), 71-fold of Hi-C reads, and 15-fold Oxford Nanopore technologies (ONT) reads in development of the HCC1395BL_v1.0 assembly (Additional file 1: Table S1). For HCC1395, we used about 170-fold of Illumina short reads from 3 replicates (FDT1, FDT2, and FDT3) as FDT123 sequenced by Fudan University, 160-fold of 10X Genomics linked reads, and 46-fold of Pacific Bioscience (PacBio) long reads (full-pass subreads, average length 8146 bp). We also added all 12 WGS tumor-normal paired replicates (Illumina short reads) from 6 sequencing centers, including FDT1/FDN1 (FDR1), FDT2/FDN2 (FDR2), FDT3/FDN3 (FDR3), ILT1/ILN1 (ILR1), ILT2/ILN2 (ILR2), ILT3/ILN3 (ILR3), NVT1/NVN1 (NVR1), NVT2/NVN2 (NVR2), NVT3/NVN3 (NVR3), NCT1/NCN1 (NCR1), EAT1/EAN1 (EAR1), and LLT1/LLN1 (LLR1), for short-read mapping and SNV analysis [5, 6]. The median insert sizes were 377/367 for FDT1/FDN1, 375/371 for FDT2/FDN2, 371/368 for FDT3/FDN3, 417/419 for ILT1/ILN1, 395/393 for ILT2/ILN2, 401/402 for ILT3/ILN3, 404/400 for NVT1/NVN1, 394/390 for NVT2/NVN2, 389/395 for NVT3/NVN3, 408/417 for NCT1/NCN1, 422/412 for EAT1/EAN1, and 372/377 for

LLT1/LLN1 [5, 6]. Library preparations and sequencing for Illumina short reads, 10X Genomics linked reads, and PacBio long reads were described previously [5, 6].

Dovetail Hi-C library preparation and sequencing: The Dovetail Hi-C libraries were prepared as described previously (Erez Lieberman-Aiden et al., 2009). For each library, chromatin was fixed in place in the nucleus with a 1% formaldehyde solution and then extracted. Fixed chromatin was digested with DpnII, the 5′ overhangs were filled in with biotinylated nucleotides, and then free blunt ends were ligated. After ligation, crosslinks were reversed, and the DNA purified from protein. Purified DNA containing biotinylated free-ends was removed as it does not reflect proximity-ligated molecules. The DNA was then sheared to ~350 bp mean fragment size, and sequencing libraries were generated using NEBNext Ultra enzymes and Illumina-compatible adapters. Internal biotin-containing fragments were isolated using streptavidin beads before PCR enrichment of each library. The libraries were sequenced on an Illumina HiSeq X to a depth of ~200M read pairs per library.

Oxford Nanopore technologies (ONT) MinION sequencing data: Genomic DNA from the HCC1395BL cell line was extracted using the QIAGEN MagAttract HMW DNA Kit (QIAGEN, Hilden, Germany). One microgram of freshly isolated genomic DNA without fragmentation was used for library construction using the SQK-LSK109 ligation sequencing kit (ONT, Oxford, UK). Libraries were prepared following ONT standard protocol. Each library was sequenced on an individual MinION FLO-MIN106D R9.4 flowcell. Prior to sequencing, flowcell pore counts were measured using the MinKNOW Platform QC script (Oxford Nanopore Technologies, Oxford, UK). About 300 ng of completed libraries was loaded as instructed by ONT. Raw sequence reads were called in real time by the MinION operating software MinKNOW (Guppy version 2.1.3). Sequence data passing quality parameters (qmean > 7) were converted to fastq format. Only the reads passed the QC were included in further analyses.

### Assembly, polishing, scaffolding, and phasing

PacBio long reads data were first error-corrected and then assembled into primary contigs using the "canu" assembler (version 1.8) [20] with option by its developers. The contig sequences were then polished with Illumina paired-end reads using PILON (version 1.22) [42]. The polishing process was performed twice to achieve the best results. Scaffolding with linked reads was performed using ARCS (version 1.0.5) [43], while scaffolding with Hi-C data was completed using SALSA (https://github.com/marbl/SALSA) [44]. Linked reads from 10X Genomics were assembled using the "Supernova" assembler (version 2.0.0) [21] as instructed by its developer.

Contig assembly with PacBio long reads and polishing with Illumina short reads:

canu -p hcc1395bl -d hcc1395bl_out genomeSize=3.1g useGrid=false maxThreads=16 corConcurrency=4 corThreads=4 cormmapThreads=4 cormhapConcurrency=4 corovlConcurrency=4 maxMemory=360g correctedErrorRate=0.075 -pacbio-raw pacbio_reads.fa

java -Xmx300G -jar pilon-1.22.jar --genome pacbio_contig.fa --frags ILMN_read.bam --diploid --fix bases --outdir pilon --output pilon --changes

Scaffolding with 10X Genomics' linked reads using ARCS:

```
longranger-2.2.2/longranger   align   --id=FDN123   --fastqs=./lib1/,./lib2/,./lib3/,./
lib4/,... --sample=FDN123 --reference=./refdata-pacbio_contig_pilon --localmem=200
--localcores=8 --jobmode=local
```

```
arcs --file=./pacbio_contig_pilon.fasta --fofName=./aln_list.txt
```

```
python ./arcs/Examples/makeTSVfile.py ./pacbio_contig_pilon.fa.scaff_s98_c5_l0_d0_
e30000_r0.05_original.gv    hccbl_pilon.fasta.scaff_s98_c5_l0_d0_e30000_r0.05.tigpair_
checkpoint.tsv ./pacbio_contig_pilon.fa
```

```
links_v1.8.6/LINKS -f ./ pacbio_contig_pilon.fa -s empty.fof -k 20 -b pacbio_con-
tig_pilon.fa.scaff_s98_c5_l0_d0_e30000_r0.05 -l 5 -t 2 -a 0.3
```

```
mv   pacbio_contig_pilon.fa.scaff_s98_c5_l0_d0_e30000_r0.05.scaffolds.fa   pacbio_
contig_pilon_arcs.scaffolds.fa
```

Scaffolding with Hi-C reads using SALSA:

```
SALSA/run_pipeline.py -a pacbio_contig_pilon.fa -l pacbio_contig_pilon.fa.fai -b
reads.bam_k4sort.bed -e GATC -o scaffolds
```

Supernova assembly for 10X Genomics linked reads:

```
supernova-2.0.0/supernova   run   --id=FDN123   --fastqs=./lib1/,./lib2/,./lib3/,./
lib4/,... --sample =FDN123 --localmem 360
```

After scaffolding with ARCS and SALSA, we mapped the unitig sequences, which were produced with Illumina short reads using fermikit (version r188) [45], to the scaffold assembly using BWA [46], and then used bcftools (version 1.6, https://samtools.github.io/bcftools/bcftools.html) to generate the final consensus assembly (HCC1395BL_v1.0). Scaffolds smaller than 10 kb were excluded from further analysis.

```
fermi.kit/fermi2.pl unitig -s3g -t8 -l150 -p prefix "cat WGS_FDN123_R1.fq.gz
WGS_FDN123_R2.fq.gz" > prefix.mak
```

```
make -f prefix.mak
```

```
bwa mem -t 8 scaffolds_FINAL.fasta prefix.mag.gz | samtools sort -@ 8 -o unitig-
FDN123_sorted.bam
```

```
bwa index scaffolds_FINAL.fasta
```

```
samtools mpileup -uf scaffolds_FINAL.fasta unitigFDN123_sorted.bam | bcftools
call -mv -Oz -o calls.vcf.gz
```

```
tabix calls.vcf.gz
```

```
cat scaffolds_FINAL.fasta | bcftools consensus calls.vcf.gz > final_consensus.fa
```

The Illumina short reads were aligned onto the HCC1395BL_v1.0 genome using BWA mem [46], and duplicated reads were marked with Picard MarkDuplicates. High-confidence heterozygous sites (QUAL $\geq$ 30) were identified using GATK4 (version gatk-4.0.3.0) [47]. Calls on chrX, chr6p, and chr16q regions were excluded. Phasing was performed with the identified high-confidence heterozygous sites and long reads from PacBio and ONT using WhatsHap (version 0.18) phasing tool [48]. Statistics of phasing was generated using "whatshap stats." Two haplotypes of the assembly in FASTA format were also reconstructed with the phasing information. Assembly-based germline SNVs in diploid regions of autosomal chromosomes were called using dipcall (https://github.com/lh3/dipcall) with two haplotypes as inputs.

```
whatshap   phase   --reference final_consensus.fa   -o   phased.vcf   ILMN_gatk.vcf
pacbio_minimap2_sorted.bam --ignore-read-groups --sample=HCC1395BL
```

```
whatshap stats --gtf=phased.gtf phased.vcf 1>phased.vcf.gz_stats
bgzip phased.vcf
tabix phased.vcf.gz
bcftools consensus -H 1 -f final_consensus.fa phased.vcf.gz > haplotype1.fasta
bcftools consensus -H 2 -f final_consensus.fa phased.vcf.gz > haplotype2.fasta
```

### Assembly evaluation

QUAST (version 5.0.0) [49] and Benchmarking Universal Single-Copy Orthologue (BUSCO, version 3.0.0) [50] were used to assess the quality of each de novo assembly. BLAT (v36) was used for mapping all RefSeq mRNA transcripts (accession prefixed with NM_ and NR_) (https://ftp.ncbi.nlm.nih.gov/genomes/all/GCF/000/001/405/GCF_000001405.38_GRCh38.p12/GCF_000001405.38_GRCh38.p12_rna.fna.gz) that were previously annotated on the GRCh38 assembly to the new assembly with parameter minIdentity 92.

```
quast assembly1.fa assembly2.fa assembly3.fa …. -m 1000 --no-icarus
python run_BUSCO.py --in final_consensus.fa --out --lineage_path …lineage_files/
mammalia_odb9 --mode genome -sp human
```

For GRCh38 consistency analysis, each assembly was compared with the GRCh38 reference assembly (https://ftp-trace.ncbi.nlm.nih.gov/ReferenceSamples/seqc/Somatic_Mutation_WG/technical/reference_genome/GRCh38/GRCh38.d1.vd1.fa) using minimap2 [51]. Alignments with mapping quality 60 and alignment length 100Kb+ were considered as good links for the consistency plot by Circos (Krzywinski, M., et al., 2009).

We also introduced a new parameter "Top50," which is the summed length of the 50 longest scaffolds, to monitor the contiguity of a given assembly during the scaffolding process, as the long-read and Hi-C sequencing technologies could make it possible to have arm-scale or chromosomal scale assembly. For the human genome with a total of 48 chromosomal arms, Top50 might be a suitable indicator to reflect the contiguity of the scaffold assembly if each chromosomal arm forms a scaffold.

### Genome annotation

To better annotate the final assembly HCC1395BL_v1.0, BLAT (version 36) and AUGUSTUS (version 3.3.1) [52] were used to map the previously described RefSeq transcripts to the assembly (excluding all pseudogenes and genes from NC_000024 chromosome Y). Protein-coding transcripts with annotations containing "pseudogene" and non-protein-coding transcripts with annotations containing "pseudo=true" in their deflines were considered as "pseudogenes" in this analysis. For BLAT, the option "minIdentity" was set to 92. Transcripts with more than 95% alignment and 95% ungapped identity were considered mapped onto the HCC1395BL_v1.0 assembly. In case of multiple mapping locations, the best mapping location with the maximum number of matching bases for the transcript was selected.

```
blat -minIdentity=92 -q=rna -out=psl final_consensus.fa GCF_000001405.38_
GRCh38.p12_rna.fa cdna.out.psl
augustus --species=human --hintsfile=hints.E.gff –extrinsicCfgFile extrinsic.ME.cfg
--outfile=augustus.out final_consensus.fa
```

**Read mapping and somatic variant detection**

BWA [46] was used to align Illumina short reads from each of the 12 replicates onto the de novo (HCC1395BL_v1.0) and the GRCh38 primary assemblies, respectively. Duplicate reads were marked with Picard MarkDuplicates. For FDN123 and FDT123, 3 bams from 3 replicates of normal sample, and 3 bams from 3 replicates of tumor sample were merged separately using "samtools merge." Mapping statistics such as reads mapped, reads unmapped, reads mapped and paired, reads properly paired, and mismatches were collected using samtools (version 1.11) with the "stats" option (http://www.htslib.org/doc/samtools-stats.html). We defined the numbers of non-properly (or improperly) paired reads as the subtraction of properly paired reads from the mapped-and-paired reads. PacBio long reads were aligned onto two references using minimap2 [51]. Standard deviations of read coverages were based on read alignments with minimum mapping quality 10.

bwa mem -t 8 -R "@RG\tID:FDN1\tSM:HCC1395BL\tLB:FDN1\tPU:FDN1\tPL:illumina" GRCh38.d1.vd1.fa WGS_FDN1_R1.fq.gz WGS_FDN1_R2.fq.gz | samtools view -bS - > FDN1.bam

samtools sort -T . FDN1.bam > FDN1_sorted.bam

samtools index FDN1_sorted.bam

java -jar picard.jar MarkDuplicates INPUT=FDN1_sorted.bam OUTPUT=FDN1_sorted_dupmarked.bam METRICS_FILE=metrics.txt

samtools stats FDN1_sorted_dupmarked.bam > FDN1_sorted_dupmarked.bam_stats

For all SNV/SV variant analysis, variants in VCF files with "PASS" filter were included. The chrX, chr6p (coordinates below 58,500,000), and chr16q (coordinates above 38,400,000) regions were not included for variant comparison, for consistency with the reference somatic set from the SEQC2 Somatic Mutation Working Group [5, 6]. Variant calls from chrY and unlocalized/unplaced sequences (names with chrUn_, _random, _decoy, etc.) were also excluded. Strelka2 (version 2.9.2) [27] and MuTect2 (version gatk-4.0.3.0/gatk Mutect2) [28] were used to identify somatic SNVs and indels. MuTect2 VCF output was filtered using "gatk FilterMutectCalls." MuTect2 reported SNPs and InDels in a single VCF file, and Strelka2 reported SNVs and InDels in separate VCF files. SNPs and InDels from MuTect2 and Strelka2 calls were compared separately using "bcftools isec" followed with "rtg vcfeval" (https://github.com/RealTimeGenomics/rtg-tools; doi: https://doi.org/10.1101/023754) [53] to obtain common sites called by both callers.

gatk Mutect2 --native-pair-hmm-threads 8 -R GRCh38.d1.vd1.fa -I FDT1_sorted_dupmarked.bam -tumor FDT1 -I FDN1_sorted_dupmarked.bam -normal FDN1 -O mutect2_snvs_indels.vcf.gz

gatk FilterMutectCalls --variant mutect2_snvs_indels.vcf.gz --output mutect2_snvs_indels_filt.vcf

strelka-2.9.2.centos6_x86_64/bin/configureStrelkaSomaticWorkflow.py --normalBam FDN1_sorted_dupmarked.bam

--tumorBam FDT1_sorted_dupmarked.bam --referenceFasta GRCh38.d1.vd1.fa --runDir strelka2_fdt1

./runWorkflow.py -m local -j 8

bcftools isec Strelka2_vcf.gz MuTect2_vcf.gz -p isec_Streka2_MuTet2 --collapse all

rtg vcfeval --baseline 0000.vcf.gz --calls 0001.vcf.gz --template GRCh38/SDF --output rtg_results --sample HCC1395,HCC1395 --vcf-score-field INFO.TLOD --squash-ploidy

Somatic SVs from Illumina short reads (WGS) were discovered using GRIDSS2/GRIPSS [32], Manta [33], Delly [34], and novoBreak [35] as suggested by their developers. Merged tumor and normal (FDT123/FDN123) bams from 3 tumor-normal paired replicates (FDT1/FDN1, FDT2/FDN2, and FDT3/FDN3) from one sequencing center (FD) were used for initial somatic SV discovery. Later on, Illumin short-read sequencing data from all 12 paired replicates from 6 sequencing centers were analyzed using all four short-read somatic callers. SVs with SVLEN smaller than 50bp were removed. Intra-chromosomal BNDs (if reported) were filtered out for comparison. SVs with quality score below 20 from novoBreak calls were excluded.

gridss-2.13.2/gridss --jar gridss-2.13.2/gridss-2.13.2-gridss-jar-with-dependencies.jar —reference GRCh38.d1.vd1.fa --output gridss_output.vcf.gz --assembly assembly_n1t1.bam --thread 8 --workingdir ./gridss1 FDN1_sorted_dupmarked.bam FDT1_sorted_dupmarked.bam

java -jar gripss/gripss.jar -sample FD_T1 -reference FD_N1 -ref_genome GRCh38.d1.vd1.fa -pon_sgl_file gridss1/pondir/gridss_pon_single_breakend.bed_sort -pon_sv_file gridss1/pondir/gridss_pon_breakpoint.bedpe_sort -vcf FD_T1_gripss.vcf.gz -output_dir gridss1/pondir

R/4.1.2/bin/R --vanilla --slave < gridss/example/simple-event-annotation.R

manta-1.6.0.centos6_x86_64/bin/configManta.py --normalBam FDN1_sorted_dupmarked.bam --tumorBam FDT1_sorted_dupmarked.bam --referenceFasta GRCh38.d1.vd1.fa --runDir ./T1N1

./T1N1/runWorkflow.py

delly call -x human.hg38.excl.tsv -q 20 -s 15 -o fd_t1.bcf -g GRCh38.d1.vd1.fa FDT1_sorted_dupmarked.bam FDN1_sorted_dupmarked.bam

delly filter -f somatic -o fd_t1.pre.bcf -s ./samples.tsv fd_t1.bcf

delly call -g GRCh38.d1.vd1.fa -v fd_t1.pre.bcf -o fd_geno.bcf -x human.hg38.excl.tsv FDT1_sorted_dupmarked.bam FDN1_sorted_dupmarked.bam

delly filter -f somatic -o fd_t1.somatic.bcf -s samples.tsv fd_geno.bcf

novoBreak_distribution_v1.1.3rc/run_novoBreak.sh novoBreak_distribution_v1.1.3rc GRCh38.d1.vd1.fa FDT1_sorted_dupmarked.bam FDN1_sorted_dupmarked.bam 8 novobreak_out

Alignment-based structural variations from PacBio long-read data were identified using PBMM2/PBSV pipeline (version 2.4.0, https://github.com/PacificBiosciences/pbsv) and NGMLR/Sniffles2 pipeline [54] (version 2.0.5, https://github.com/fritzsedlazeck/Sniffles, https://github.com/philres/ngmlr). We noticed that there were certain differences in their alignments with two aligners (PBMM2 vs. NGMLR) using PacBio subreads, thus resulting SV calls (PBSV vs. Sniffles2) were different in some regions. Therefore, to minimize aligner and caller bias and maximize callers' concordance in subsequent merged callset, in our PacBio long-read SV analysis, Sniffles2 was also applied to PacBio pbmm2 bams after the bams were processed with "samtools calmd -u," and PBSV was applied to NGMLR bams. SVs were called jointly with both tumor and normal sample together. Therefore, for each assembly, four VCFs (PBMM2+PBSV, PBMM2+Sniffles2, NGMLR+PBSV, and NGMLR+Sniffles2) were generated for downstream analysis. Only calls with "PASS" were retained. Calls with "IMPRECISE,"

"SHADOWED," and SVLEN smaller than 50bp were removed. CNV calls and intra-chromosomal BND from PBSV were ignored. Two selection steps were applied for identifying somatic SVs based on the genotypes, alternate allele counts, and allele frequency in tumor and normal sample reported by PBSV and Sniffles2. Firstly, if the genotype of the site for normal sample was reported as "reference" in normal sample, the site was retained only if the genotype for tumor sample was reported as "1/1" or "0/1," with minimum alternate allele count 5 and minimum allele frequency 0.2. Secondly, if the genotype of the site for normal sample was reported as "0/1" in normal sample, the site was retained only if the genotype for tumor sample was reported as "1/1" with minimum alternate allele count 10, minimum allele frequency 0.85, and allele frequency difference between tumor and normal sample 0.45 and above.

pbsv discover -s Tumor HCC1395_pbmm2_merged.bam HCC1395_pacbio_B38.svsig.gz

pbsv discover -s Normal HCC1395BL_pbmm2_merged.bam HCC1395BL_pacbio_B38.svsig.gz

pbsv call -j 8 GRCh38.d1.vd1.fa HCC1395_pacbio_B38.svsig.gz HCC1395BL_pacbio_B38.svsig.gz HCC1395_pacbio_pbmm2_B38_pbsv_TN.vcf

Sniffles2.0/bin/sniffles --input HCC1395_ngmlr_B38.bam --vcf HCC1395_pacbio_ngmlr_sniffles2_B38.vcf.gz --snf HCC1395_pacbio_ngmlr_sniffles2_B38.snf --threads 8

Sniffles2.0/bin/sniffles --input HCC1395BL_ngmlr.bam --vcf HCC1395BL_pacbio_ngmlr_sniffles2_B38.vcf.gz --snf HCC1395BL_pacbio_pbmm2_sniffles2_B38.snf --threads 8

Sniffles2.0/bin/sniffles --input HCC1395_pacbio_ngmlr_sniffles2_B38.snf HCC1395BL_pacbio_ngmlr_sniffles2_B38.snf --vcf HCC1395_pacbio_ngmlr_sniffles2_B38_TN.vcf

Assembly-based SVs were generated from direct comparisons of the contigs of tumor cell line (HCC1395) with the HCC1395BL_v1.0 reference (as opposed to GRCh38 reference) using paftools [51] and Assemblytics [55] (https://github.com/MariaNattestad/assemblytics; http://assemblytics.com/) with procedures suggested by their developers. For Assemblytics, we prepared the delta input file by aligning contigs fasta to a reference using MUMmer/nucmer and run with the options of "10,000" for "Unique sequence length required," "20,000" for "Maximum variant size," and "50" for "Minimum variant size". For paftools, only SVs with SVLEN equal to or greater than 50bp were included for analysis. The paftools tool reports only deletions and insertions, and it identified 7475 large deletions and 5215 large insertions based on HCC1395 contigs on GRCh38 reference, but only 3154 (57.8% less) large deletions and 2425 (53.5% less) insertions on HCC1395BL_v1.0 (Additional file 2: Fig. S11A). Assemblytics reports repeat/tandem contractions and expansions in addition to deletions and insertions. We observed that, with the exception of repeat_contraction category, Assemblytics identified fewer SVs on HCC1395BL_v1.0 (25.53% fewer deletions, 63.86% fewer insertions, 35.08% fewer repeat expansions, 32.49% fewer tandem contractions, and 71.29% fewer tandem expansions) (Additional file 2: Fig. S11B). To circumvent the different SV notations that were used by paftools and Assemblytics (e.g., a 668 bp Deletion at chr3:159539232-159539900 called by paftools vs. a 668 bp Repeat_contraction at chr3:159539232-159543981 called by Assemblytics), we combined deletions with repeat/tandem contractions (as "DEL") and insertions with repeat/tandem expansions (as "INS") from Assemblytics calls for comparison purposes. Somatic SVs were retained by removing calls that

were overlapping with germline SV calls (requiring allele frequency 0.1 and above, and alternate allele count 5 or more) identified by Sniffles and PBSV in the normal sample with PacBio long reads. Due to the nature of such somatic SV set that were generated using similar approaches, for consensus somatic SVs analysis, we combined these 2 contig mapping-based somatic SV sets (by Assemblytics/paftools) along with 4 PacBio long-read-based somatic SV sets together (by Sniffles2/PBSV with PBMM2/ngmlr ) to evaluate how two genome references were affecting somatic SVs that were supported by at least two calling methods.

minimap2 -cx asm5 -t8 --cs GRCh38.d1.vd1.fa HCC1395_pacbio_contigs_polished.fasta > asm.paf

sort -k6,6 -k8,8n asm.paf > asm.srt.paf

paftools.js call asm.srt.paf > asm.var.txt

nucmer -maxmatch -l 100 -c 500 REFERENCE.fa contig.fa -prefix OUT

gzip OUT.delta

Consensus somatic SVs from multiple somatic SV callsets were generated using "merge" function of SURVIVOR (version: 1.0.7) with the parameters "max distance between breakpoints = 1000," "Minimum number of supporting caller = 2," and "Minimum size of SVs to be taken into account = 50."

SURVIVOR merge vcfs.list 1000 2 0 0 0 50 merged.vcf

### Mapping GRCh38-based SNVs and SVs to the de novo assembly

To find the locations on the de novo assembly corresponding to the GRCh38-based somatic SNVs we identified, we used a two-step mapping approach. We extracted both the reference and alternate alleles of each SNV with their 50bp flanking sequences from GRCh38 and created a fasta file before mapping using BLAST (blast 2.10.1). The first step was to map all SNVs with more stringent criteria so that SNVs with identity ≥99% and alignment length ≥101 bp were selected. The unselected SNVs from Step1 were then mapped in the second step with lower thresholds (95% identity and 95 bp alignment) to select the SNVs that mapped best despite some mismatches and small indels (Additional file 2: Fig. S5). For both steps, both alleles of each SNV were required to map onto the same locations, with identical start and end positions on the de novo assembly. In addition, to be considered an equivalent SNV call between GRCh38 and the de novo assembly, the alternate allele was required to be at the center position. Manual inspections on IGV for some SNVs were also performed. Unselected SNVs from Step2 were considered to be unmapped.

blastn -query query.fa -db blastdb/final_consensus -num_threads 8 -evalue 1e-10 -word_size 7 -num_alignments 10 -perc_identity 95

-qcov_hsp_perc 95 -dust no -soft_masking false -out blast.out -outfmt "6 qseqid sseqid qlen slen pident length mismatch gapopen qstart qend sstart send sstrand evalue bitscore"

GRCh38-based SVs were mapped to HCC1395BL_v1.0 assembly using a similar approach as for mapping SNVs. Sequences of 100 base pairs from each SV's flanking (for DEL/DUP/INS/INV only) were extracted as fasta format, and then were mapped using BLAST (blast 2.10.1). Only SV events with two flanking sequences being mapped in the

same locations with the minimum 98% identity and 90 bp alignment were considered as "Mapped" onto HCC1395BL_v1.0. Otherwise, the SVs were considered as "Unmapped."

### Variant annotation, pathway analysis, and repeat annotation

Variant function analysis was performed using ANNOVAR (version: de74a7d-59955d769c6cbb92a0d64d12c90c8eede, 2018-04-16) [29]. Pathway analysis was performed through the Enrichr web site (https://maayanlab.cloud/Enrichr/).

annovar/annotate_variation.pl -build hg38 var.input humandb/ -dbtype ensGene

Sequences of deletions (from predicted start to end) were extracted as fasta format, then were annotated using an online RepeatMasker tool (https://www.repeatmasker.org/cgi-bin/WEBRepeatMasker).

### Mitochondrial sequence analysis

Contigs from HCC1395BL assembly and HCC1395 assembly that fully covered the mitochondrial sequences from GRCh38 (16,569 bp; https://www.ncbi.nlm.nih.gov/nuccore/NC_012920.1) were selected based on minimap2 mapping results. Since the mitochondrial genome is circular, the full mitochondrial sequences were extracted from each of the selected contigs based on BLAST mapping results. CLUSTAL (v1.2.4) was used to generate multiple sequence alignments for variant analysis. The variants were annotated with the MITOMAP human mitochondrial genome database (http://www.mitomap.org, 2019) and dbSNP (v153).

### PCR validation using Sanger sequencing

We randomly selected 12 SNVs (MAF ranges from 0.5 to 1) from those 177 SNVs that were discovered only using HCC1395BL_v1.0 as reference and designed primers for PCR validation using Sanger sequencing. To confirm the specific point mutations, primers flanking the mutations were designed with an online software- Primer3. The point mutation flanking regions were then amplified using either control or tumor DNA samples as a template with Phusion flash High-Fidelity PCR master mix (Thermo Fisher Scientific, Waltham, MA.). The PCR conditions were 98 °C for 30 s, followed by 35 cycles of denaturing at 98 °C for 1 s, annealing at 64 °C for 5 s and extension at 72 °C for 15 s. The PCR products were then purified with GeneJET PCR purification kit (Thermo Fisher Scientific, Waltham, MA) and sequenced at GENEWIZ (Genewiz, South Plainfield, NJ). We were not able to design primers to cover either the SNV scaffold_1:3482191 (MAF=1) or the SNV scaffold_8:61936447 (MAF=0.557) without overlapping due to the technical limitations for Sanger sequencing, thus these two SNVs were left out for further assessment.

### Supplementary Information

Additional file 1. Included all the supplementary tables for this manuscript.

Additional file 2. Included all the supplementary figures for this manuscript.

Additional file 3. Review history.

**Acknowledgements**

Javkhlan Ganbat provided useful help regarding Hi-C data analysis in the early stage. Thanks to Malcolm Moos for the helpful comments on drafts of this manuscript.

**Peer review information**

**Review history**

The review history is available as Additional file 3.

**Disclaimer**

The content of this manuscript is solely the responsibility of the authors, and the views presented here do not necessarily reflect official policy of the US Food and Drug Administration or US National Institutes of Health. Any mention of commercial products or materials or tools is purely for clarification purposes and not intended as endorsement or discouragement.

**Authors' contributions**

CX, WX, and CW designed the study. CX wrote the manuscript draft. CP performed Hi-C sequencing and QA/QC. ZC, WC, TL, and CW performed ONT sequencing and QA/QC. WW and YY performed Sanger sequencing validation. CX, WX, WC, ZC, CW, CP, LF, and MC performed the analyses. CX, WX, WC, ZC, CW, and VS revised and improved the manuscript. CX, VS, and WX performed the data management. All authors read and approved the manuscript. ZC, WC, CW, and CX recreated the final main figures for publication. CX finalized and submitted the manuscript. CX managed the overall project.

**Funding**

 The genomic work carried out at the LLU Center for Genomics was funded in part by the National Institutes of Health (NIH) grant S10OD019960 (CW), the Ardmore Institute of Health (AIH) grant 2150141 (CW), Dr. Charles A. Sims' gift to LLU Center for Genomics. The study was also partially supported by the American Heart Association grant 18IPA34170301 (CW). Dr. Yibin Yang was also partially supported by NIH R01 CA251674 (Y.Y.). Chunlin Xiao and Valerie Schneider were supported by the Intramural Research Program of the National Library of Medicine, National Institutes of Health.

**Availability of data and materials**

All raw sequencing data used in this study are publicly available in NCBI SRA database (SRP162370 under NCBI BioProject PRJNA489865) [56]. The HCC1395BL_v1.0 genome assembly has been deposited at NCBI GenBank with accession GCA_021234545.1 [57]. The final de novo assembly fasta file for HCC1395BL_v1.0 is also accessible via NCBI ftp site (https://ftp-trace.ncbi.nlm.nih.gov/ReferenceSamples/seqc/Somatic_Mutation_WG/assembly).

All software or tools used for de novo genome assemblies, assembly evaluations, and variant calls were publicly available and listed in the "Methods" section.

## Declarations

**Ethics approval and consent to participate**

Not applicable.

**Competing interests**

CP was employed by Dovetail Genomics, LLC, and LF was employed by Roche Sequencing Solutions Inc during the course of this research. The other authors declare that they have no competing interests.

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

## 

