## [Additional file 3. Review history. · Genome Biology]

Review History

First round of review

Reviewer 1

Are you able to assess all statistics in the manuscript, including the appropriateness of statistical tests used? Yes, and I have assessed the statistics in my report.

Comments to author:

General comments:

Xiao et al present an evaluation of the impact having a personalised reference genome has on somatic variant calling. Surprisingly, they claim to be the first analysis to do this. Even more surprisingly, was that I was unable to find a prior publication on exactly this topic and kept ending up at their preprint. This approach has been discussed for years and holds great potential and an evaluation of exactly how good this approach actually is is long overdue.

As one might expect, genes not well represented in the reference genome and complex regions like the HLA region do very well with a personalised reference genome. Better SNV calling in regions with nearby variants also shows the value in using a personalised reference genome.

While the majority of the analysis appears sound, this manuscript leaves me with more questions than answers. The methods section is too brief for me to tell how much of their results are due to pipeline-specific artefacts, and how much should be generalisable.

With the data they have available, the authors are able to evaluate the impact of a personalised reference on SNV, indel, SV, and CNV calling for short, long and linked reads. It is somewhat disappointing that they've only evaluated short read SNV calling, and long read SV calling.

Whilst some of these would be difficult to evaluate (there aren't really any long read CNV callers, nor specialised paired tumour/normal long read SV callers, and paired tumour/normal analysis of linked reads is very niche), it would be nice to know things like:

- whether a personalised reference actually does anything for short read somatic SV/CNV calling
- whether one can get away with just running GATK and doing germline subtraction if a personalised reference existed
- whether this approach of creating a very good reference assembly meaningfully outperforms a hacky "GRCh38 with germline SNPs/indels" personal reference that one could create from short read sequencing of the normal.

Specific comments:

Figure 1: the assembly -> scaffolding -> phasing flow is a bit misleading.

Although it does represent the steps performed, it does not represent the actual flow of information. For example, the arrow between assembly and scaffolding seems to imply that both the PacBio_canu assembly and the 10x Chromium supernova assemblies were input to the scaffolding phase. This does not match page 5 line 10 which states "We then selected the PacBio contig assembly of HCC1395BL for further scaffolding.". Visual indicators of which assembly/scaffold/phased output was actually used in the next step would improve clarity. In

terms of actual results, it is just 1 assembly, 1 scaffold, and 1 phasing result that actually get used, the rest are experiments into alternative approaches for each step but are otherwise unused. Similarly, it appears that although you generate haplotype fasta files in your phasing stage, you don't actually use the downstream. Is this correct? This doesn't match what Figure 1 seems to imply. What steps are run off the haploid HCC1395BL_v1.0 scaffolded assembly, and what are run off earlier/later assemblies?

Page 8 line 6: "We successfully identified all the HLA genes and corresponding transcripts in HCC1395BL_v1.0"

Buried in the methods section (Page21 Line10-14) is the custom logic you needed for this full HLA identification to actually work. This should be mentioned in your results section. Notwithstanding this, direct HLA determination without having to run a specialised HLA genotyper is a good result.

Page 11 line 6-7. This isn't quite as simple as you seem to imply. Yes, it's definitely better that you have the gene in the assembly, but converting from the personalised reference into a form in which the cancer knowledgebases can be used is a non-trivial operation. In some respects, alt-only genes are actually easier to deal with than having to deal with every gene being on a different scaffold for every patient and having to reidentify genes every time.

Methods require more detail to be reproducible. Software versions are included which is good but the methods don't specify what arguments/parameters/options were used for each step. For example, what were the canu command-line parameters? What did you specify as the target genome size to canu? Were you using an options file? If so, what was in it? You don't need to specify setting such as canu's cluster queuing overrides as they're environment-specific but settings that impact the assembly output need to be specified if your analysis is to be considered reproducible.

One way to make this information available without making the actual methods section itself much larger is to make the actual scripts you used to perform the analysis available either on a publicly visible github, or as supplementary materials in the manuscript.

Without this level of methods detail is it difficult to evaluate whether the methods are appropriate.

For example, page22 line 4 says you use Assemblytics. You don't specify what you used as the input. The Assemblytics FAQ states: "Important: Use only contigs rather than scaffolds from the assembly. This will prevent false positives when the number of Ns in the scaffolded sequence does not match perfectly to the distance in the reference.". Did you use to scaffolded HCC1395BL_v1.0, or the raw contigs? This choice massively impacts your Fig 5F result but your methods section doesn't actually say which one you used which gives me little confidence in the Fig5F result as it's quite possibly an artifact of your pipeline.

This level of methods detail is essential as even seemingly minor differences in a bioinformatics pipeline can have a very large impact on the results.

Are your PacBio reads HiFi reads? I assume they're not but you should explicitly mention this somewhere.

Page12 Line11. Does the MinKNOW base-calling software use have a version? Nanopore base

calling is a very active area which significant improvements being made regularly (including recent improvements to Guppy).

One of the key questions regarding using a personalised reference genome is whether this obviates the need to actually perform tumour/normal variant calling and if calling variants directly from the personalised reference is sufficient to identify somatic mutations.

Do we still need to run tumour/normal variant callers such as Strelka2 or MuTect2, or can we get away with just running GATK and subtracting the germline heterozygous calls (or using a NovoBreak-style reference containing the ambiguous IUPAC nucleotide codes at heterozygous sites - e.g. germline A/T = W).

As you have already run germline GATK during the phase stage, it should be relatively straightforward to run GATK on the tumour, subtract the germline calls, and compare the resultant somatic call set to the Strelka2/MuTect2 calls. Do these GATK calls have so much noise so as to make them unusable, or are they a viable alternative to the specialised paired tumour/normal callers when using a personalised reference genome?

Similarly, the high quality reference genome you have created has high cost associated with its generation. How much better is this approach than a light-weight personalised germline one could create by just augmenting GRCh38 with the germline SNV/indels one could identify from short read sequencing of the normal? The biggest difference seems to be that the personalised reference does a better job of calling somatic variants when there are nearby germline variants. Does doing something like creating a heterozygous reference (het A/T=W) and aligning with a IUPAC-aware aligner such as NovoBreak perform almost as well as a good personalised reference genome?

No details on how SNVs were matched is included in your manuscript. Is it possible that some of the missing SNVs are not actually missing, but are reported in a format that your analysis pipeline does not account for? VCF allows variants to be represented in a multitude of ways.

Take the following example:

TAGTG ref

TTGAG sample

Can be represented as two SNVs:

2 A T

4 T A

Or could be represented as a single variant:

2 AGT TGA

If your analysis pipeline is just looking to see if there's a SNV call at position 4, you're going to incorrectly claim the variant is missed if it's represented in the latter notation.

Does your SNV analysis account for this? It's unclear as to what the difference between "Identical SNVs" and "Equivalent SNVs" are in Fig5C.

The SV analysis is relatively cursory and does not appear to account for many of the complexities encountered with SV calling:

Coordinate space. The methods state that SURIVIVOR was used for SV comparison but it is unclear how this was done without a common reference genome for the SVs. Were SVs translated to GRCh38 coordinate space? How?

Page 14 L14: This looks like a germline comparison. What matter for a somatic reference is whether the somatic SV call set is better with the reference? What is causing this bias between the ins/del length distribution of somatic calls (or is this a pseudo-germline comparison?)? If this bias is due to the GRCh38 deletion bias, shouldn't you find that the extra calls are overlap germline SVs w.r.t GRCh38? Is this the case?

Is the bias caused by microsatellites? Microsatellite instability is important in cancer genomics. This manuscript does not indicate how much microsatellite calling is improved by a personalised reference. Does a personalised reference allow microsatellite instability to be called by standard tools, or are MSI callers still needed? Are MSI calls better with the personalised reference? Does the fact that the personalised reference is haploid negate any potential MSI benefit?

P15L4: "assembly of the HCC1395 cancer cell line" what assembly of HCC1395 is this referring to? I might have missed it but I can't find anywhere in the methods where you state that you have performed assembly on the tumour cell line. Figure 1 lists "HCC1395 contigs" but you don't say how these were generated. Are they PacBio contigs? Chromium?

I'm not sure what you're trying to achieve with your comparison of HCC1385BL_v1.0 with HCC1385. If this is a somatic analysis, why do you find Alu and L1 repeat peaks? I'd expect to find those for comparisons of germline SV call sets but not for somatic. Even L1

One issue not addressed but critical to SV interpretation is the notational difference between SV call sets. The most common is notational differences between insertions and duplication. Take the following three VCF SV calls:

TACGTAGTG ref

TACGACGTGTG alt

1 T TACG

1 T <DUP> SVTYPE=DUP;SVLEN=3

1 T  SVTYPE=INS;SVLEN=3

All three VCF calls are representation of exactly the same event. Even more messy is that the calls don't have to be (and frequently aren't) left-aligned:

4 A ACGA

2 A  SVTYPE=INS;SVLEN=3

3 C  SVTYPE=INS;SVLEN=3

4 G  SVTYPE=INS;SVLEN=3

Can also be used in VCF to represent exactly the same event. SURVIVOR does not handle this.

How have you handled the potential for representational difference between your tools confounding your analysis? Can the imbalance between INS and DEL be explained by the callers reporting INS as DUP events?

What about complex rearrangements? Somatic SV that form part of complex rearrangements are extremely clustered (doi.org/10.1101/2020.07.09.196527). Does this impact somatic SV calling? There are at least 5 fusion genes in HC1395 (PMID 21808235)(or are there really?) so there's probably at least one instance of chromothripsis in there. How do the two references compare on terms of complex rearrangement resolution?

Overall, the SV results presented don't seem answer the key question: Does using a personalised reference improve somatic SV calling? There's no comparison of short read SV calling on HCC1395BL_v1.0 vs GRCh38. There's no evaluation of CNV caller on the two references. I can't seem to find any analysis of somatic SV calls that includes the critically important inter-chromosomal SVs.

You should split out the soft-clipping and hard-clipping into mutually counts of partially aligned reads (soft-clipped reads without a SA tag), and split-read alignments (reads with SA tag. Bwa uses soft-clipping for the primary alignment, and hard-clipping at all supplementary alignments). This will present double (or even triple) counting of chimerically aligned reads and better corresponds to what the aligner is actually doing. The SAM specifications use the term "chimeric" read alignment but "split read" alignment is the term in widespread usage outside the specifications document.

There are 4 scenarios for read pair alignments:

- 1) Aligned with orientation and distance expected by library fragment size distribution (aka concordantly aligned read pair; proper pair flag usually set by aligner)
- 2) Both aligned but not concordantly (aka, discordant read pair)
- 3) One read in pair aligned (aka OEA)
- 4) Both reads unaligned

It's unclear which of these categories you are grouping in each of your counts. Specially, it's unclear if 3)/OEA reads are considered "Reads with improper-pairing" or not. Do you just use the proper pair SAM flag as reported by bwa? To be technically correct you also need to account for when one or both of the reads in the read pair are split-read aligned (as will be the case when either read overlaps a true SV) but even most SV tool authors don't correctly account for this.

I'd also be interested in knowing whether/how much these results change at lower levels of coverage. Is 30x with a personalised reference as good as 60x against GRCh38? Answers to these questions impact how people think about the uptake of personalised reference genomes as it provides insights into the relative benefits which can be weighted against the costs.

I understand that fully addressing all of comment and performing all the experiment I've suggested is a lot of work. Whilst it would make for a paper that extremely comprehensively evaluates the impacts of having a personalised reference genome and I personally am interested in what it would find, only a subset of the experiments are likely to have significant impact and it is reasonable to concentrate only on the subset that are likely to have widespread interest (e.g. short read CNV calling is widespread whereas tumour/normal linked read indel analysis is very niche).

Reviewer 2

Are you able to assess all statistics in the manuscript, including the appropriateness of statistical tests used? There are no statistics in the manuscript.

Comments to author:

The paper explores the use of denovo assembled genomes for detection of somatic mutations in tumor-normal paired samples.

This work builds upon recent work (e.g.

genomebiology.biomedcentral.com/articles/10.1186/s13059-020-02047-7) demonstrating the use of a denovo assembled genome as opposed to using the reference human genome. The authors tackle an important question

but the conclusions of the paper are not well supported by the results that are presented. My main comments about the paper are

as follows:

1. Most of the results (Figure 2-3, Table 1) are for the normal genome and are applicable to any genome, i.e. for demonstrating the accuracy and quality of a de novo assembly of any human genome. Similarly, the results for the de novo assembly of the normal cell line are very similar to what was done

in a recent paper published in Genome Biology for the HG002/Ashkenazi Jewish genome (<https://genomebiology.biomedcentral.com/articles/10.1186/s13059-020-02047-7>). Since the goal of the paper is to demonstrate the utility of de novo assembly based analysis for tumor-normal sequencing. I would

have expected to see more results specifically focused on cancer relevant analysis. These would differentiate this work from previous published results.

2. Along the same lines as the previous comment, the authors looked specifically at the HLA locus. This makes sense since this is

one of the most diverge regions in the genome and can be resolved via de novo assembly. However, it would be beneficial

to demonstrate the value of assembly for genomic loci relevant to cancer. At the very end of the results, a whole section

is devoted to analyzing the mitochondrial genome. The mitochondrial genome is 16 kb in length and not difficult to assemble

using long reads. What additional value do the long reads provide compared to Illumina sequencing.

3. Very little space is devoted to the de novo assembly of the HCC1395 cancer cell line apart from one paragraph on page 6. The paper

would be strengthened if something valuable can be demonstrated from the assembly of the cancer cell line.

4. On page 14, the authors state that "suggesting that some meaningful somatic SNVs could be overlooked when using GRCh38 as the reference genome". Again

, if the authors could provide more details demonstrating that some of these somatic mutations identified only using the assembly based approach

are relevant for breast cancer or cancer, it would provide support to the conclusions of the paper.

Minor comments:

- In Figure 1, it is not mentioned whether PacBio sequencing was done using CLR or the HiFi technology.

Reviewer 3

Are you able to assess all statistics in the manuscript, including the appropriateness of statistical tests used? There are no statistics in the manuscript.

Comments to author:

Xiao et al. describe the use of a specifically assembled personal genome to compare it with a tumor genome of the same patient, both produced from cell lines. Use of personalized genome sounds promising, and it is interesting to know how results would compare to a standard reference. Hence it's disappointing that the authors did not demonstrate this advantage convincingly. Eg, the abstract only provides improvements in terms of reduction of "improper-pairings, soft-clippings and hard-clippings" [sic], which is a very indirect measure at best. Only the SNV analysis is performed but the SV work is very inadequate. There is also lack of clarity in multiple cases with the data, figures, tables.

WRITING: Text is often unclear. At the simplest level, it is problematic English, typos and awkward phrasing are quite frequent, here are some examples (there are at least twice as many across the whole paper):

caner

medical relevant genes

near-completely

High confident

The numbers of reads with improper-pairings, soft-clippings and hard-clippings were dropped 88.25% of GRCh38-based somatic SNVs were found identical to those identified on personalized genome, whereas 9.59% of GRCh38-based somatic SNVs were considered as equivalent to those somatic SNVs identified on personalized genome but with germline SNVs on their flanking.

somatic variations

have strong evidence supports

were covered with yellow. (and more errors in Figure 2 caption)

Based on our analysis, high quality assembly has been achieved based on our existing data for

For the first time to our knowledge, we were able to identify

A COUPLE OF MINOR POINTS:

Using % vs number in Abstract is poor style as it makes it harder to compare the numbers.

In Supplementary Tables, why are the captions above the tables? In the rest of the paper and Supplementary Figures, captions are below the figures and table.

MAJOR ISSUES:

There are multiple major concerns with this paper.

Two refs [46-47] are oddly inserted in the beginning without properly renumbering them. These

papers have not been published, yet they are used referring to the "well-studied reference samples by FDA-led SEQC-II consortium", the main datasets here, normal (BL) and cancer (95).

It's hard to guess what information is in those papers in press, and what is the overlap with this study. The authors need to make this really clear, to avoid concerns about the novelty of their statements. Eg, statements like "We first obtained massive whole genome sequencing data using multiple sequencing technologies" should clearly indicate that only a fraction such data (long range) was generated in this work, and the use of this fraction is extremely poorly described.

I found 2 biorxiv entries with presumably earlier versions of those 2 papers and I considered them during this review. I'll refer to them as 46 and 47.

[46] consider 21 pairs of replicates and [47] - 42 datasets (21x2, I suppose?) of short-read NGS from 6 seq centers for the same tumor-normal pair, using 3 aligners and different numbers of SNV callers, sometimes producing artificial mixtures of both genomes or adding artificial sequence data. Both have important results regarding comparisons of different protocols and bioinformatic pipelines, or a finding that WES was less reproducible than WGS.

[46] also focuses on SNVs and states "Larger structural variants and copy number analysis will be included in a separate manuscript that will discuss these findings in greater detail." I assume the current paper is that separate manuscript, as it touches upon structural variants. So the first question (Q1) is: how good is SV calling here, with all the heavy artillery (seq machinery) thrown at these genomes? I don't think an answer "it will be shown in yet another paper" will do here.

Further, in this m/s, long-range datasets have been added and the variants are called not with the standard GRCH38 but with a newly assembled personal genome. The arguments provided against the use of the standard are very reasonable and we are all intrigued how this approach performs. Thus the second question (Q2) is: how much better are the variant calls with personal genome versus the standard?

Q1. Indeed, the paper touches upon structural variants. However, it touches upon them in a rather superficial way and not in sufficient depth. And that's really disappointing: while SNV calling is relatively polished, SV detection could really benefit from the use of personal genome. In my view, they simply have not done the job here, in terms of both calling/analysis of SV and their interpretation. There are multiple concerns:

1. PacBio: why almost the same number of reads results in 14% drop in coverage depth in 95 sample, while the discrepancy is not as bad in short reads and 10x (sup tab1)? It also has consequences for downstream analyses and must be carefully explained.
2. The assembly of the 95 genome was >30% more fragmented, with the largest contigs losing >50% of their length. The authors say it is "mainly due to high level of chromosomal aneuploidy and structural variations in this cancer cell line", w/o providing any data. Confirmatory analyses based on SVs need to be presented here, to corroborate these statements and illustrate the real reasons.
3. The data generation is mostly focused on the BL, not cancer. While it is a bit strange in itself (complex rearrangements would require more, not less data), it's still important to determine if personal genome SVs (relative to GRCH38) are also found in the tumor sample. What is the

exact effect of "less data"? How would that affect a general strategy of variant calling using personal genome for future studies?

4. The structural variant analysis is totally inadequate. Fig 5A and 5C differ by 10-fold? A reader concludes that either Sniffles makes a lot of false positive SV calls or pbsv misses a lot of SVs. Both scenarios are bad but I don't see any evaluation of this result or comment.

5. Given the issues in p4 one would question the use of long reads only. With the enormous number of available short-read data, the authors should run SV analyses at least on par with the level of their effort in SNV/indel calling. A representative set of short-read SV callers needs to be run on these datasets and compared. Fuentes et al (DOI: gr.241240.118v1) would be a reasonable example to follow, it applied several SV callers including Pindel, DELLY, GROM, and Lumpy. A similar analysis of SVs is needed here.

6. The results that will be generated in p5 need to be compared with the SVs identified with long reads. Both p5 and p6 are critical analyses for the use of personal genome, and thus for the whole paper. Many points listed in Q2 below will also need to be addressed with SVs.

7. Would HLA change be detectable with short reads?

Q2. It's still unclear to me how much advantage does personal genome bring vs. the massive data generation involved.

1. Fig 4A - I am curious why BL with BL as a reference is not really better than with 95 and not MUCH better than with GRCH38? Sup Tab 5 - fewer variants for 95?

2. Analysis of mitogenome - were these variants found with GRCH38 reference? Eyeballing with IGV, done by the authors, provides perhaps the LOWEST degree of validation, after the calls were made with rather sophisticated SNV tools. If the authors claim novelty of these variants, experimental conformation is needed, which could be trivial with these cell lines. And also - what are sift/vep results?

3. Sup Tab 6 - why more variants are detected when genomes are closer? This needs to be explained and discussed. What share of detected events comes from alt-loci, what share reverts to GRCH38?

4. The text on alt-loci is relevant but GSTT1 and KIRDL5A - where are these located?

5. The authors bring up "balanced" numbers as an indication of improvement. Is that really the case? It needs to be discussed and argued for.

6. I am confused about the use of Dovetail and ONT data. The integration of these datasets needs to be detailed. These are the issues, currently:

Dovetail - usefulness not clear. Using it leads to increased numbers in the right half of Table 1. Interpretation? Specific advantages?

Oxford - usefulness not unclear. Is said to be used for phasing but "phasing" section does not even mention it. After the data is presented, one needs to see the same: Interpretation? Specific advantages?

7. The final paragraph in Discussion seems too focused on things like centromeres and telomeres, which were not relevant for tumor-normal findings here. Or were they relevant? - this is not explained.

8. Overall, what is the cost-benefit ratio of the approach involving personalized genome? How many relevant variants (unavailable with GRCH38) have been identified? Any clinical advantage of using personalized genome for this pair of samples? So far, we only see "SNVs were located in exonic or intronic regions, therefore including or excluding these sites would have an impact

on downstream mutation interpretations". Very specific examples and numbers should illustrate this vague statement. Otherwise, the advantages are inconclusive.

Authors Response

Point-by-point responses to the reviewers' comments:

Reviewer #1:

General comments:

Xiao et al present an evaluation of the impact having a personalised reference genome has on somatic variant calling. Surprisingly, they claim to be the first analysis to do this. **Even more surprisingly, was that I was unable to find a prior publication on exactly this topic and kept end up at their preprint.** This approach has been discussed for years and holds great potential and an evaluation of exactly how good this approach actually is long overdue.

As one might expect, **genes not well represented in the reference genome and complex regions like the HLA region do very well with a personalised reference genome. Better SNV calling in regions with nearby variants also shows the value in using a personalised reference genome.**

While the majority of the analysis appears sounds, this manuscript leaves me with more questions than answers. The methods section is too brief for me to tell guess how much of their results are due to pipeline-specific artefacts, and how much should be generalisable.

With the data they have available, the authors are able to evaluate the impact of a personalised reference on SNV, indel, SV, and CNV calling for short, long and linked reads. It is somewhat disappointing that they've only evaluated short read SNV calling, and long read SV calling. Whilst some of these would be difficult to evaluate (there aren't really any long read CNV callers, nor specialised paired tumour/normal long read SV callers, and paired tumour/normal analysis of linked reads is very niche), it would be nice to know things like:

- whether a personalised reference actually does anything for short read somatic SV/CNV calling
- whether one can get away with just running GATK and doing germline subtraction if a personalised reference existed
- whether this approach of creating a very good reference assembly meaningfully outperforms a hacky "GRCh38 with germline SNPs/indels" personal reference that one could create from short read sequencing of the normal.

[Response]

Thank you very much for your complimentary remarks ("**the majority of the analysis appears sounds**") and valuable suggestions regarding our manuscript. Since these questions also appear in the "Specific comments" section, they have been addressed individually there.

Specific comments:

Figure 1: the assembly -> scaffolding -> phasing flow is a bit misleading.

Although it does represent the steps performed, it does not represent the actual flow of information. For example, the arrow between assembly and scaffolding seems in imply that both the PacBio_canu assembly and the 10x Chromium supernova assemblies were input to the scaffolding phase. This does

not match page5line10 which states "We then selected the PacBio contig assembly of HCC1395BL for further scaffolding.". Visual indicators of which assembly/scaffold/phased output was actually used in the next step would improve clarity. In terms of actual results, it is just 1 assembly, 1 scaffold, and 1 phasing result that actually get used, the rest are experiments into alternative approaches for each step **but are otherwise unused.**

Similarly, it appears that although you generate haplotype fasta files in your phasing stage, you don't actually use the downstream. Is this correct? This doesn't match what Figure 1 seems to imply. What steps are run off the haploid HCC1395BL_v1.0 scaffolded assembly, and what are run off earlier/later assemblies?

[Response]

Thank you very much for the feedback and we apologize for any confusion caused by Figure 1, which is intended to show the overall study design and highlight the major tasks involved in this study. **Per your suggestion, we have made some revision to the figure.** As you noted, at each step in the assembly workflow, we performed comparative analysis to select the "best" product to feed into the next step. For example, in the initial assembly, we used "Supernova" to assemble 10X Genomics linked reads and used "canu" to assemble PacBio long reads. We evaluated the qualities of the two initial assemblies and concluded that the overall quality of PacBio canu assembly was better than that of the Supernova assembly, particularly when gene content and completeness were the primary concerns. Therefore, we selected PacBio contig assembly as the base for scaffolding in the next step, as **designated in a dotted box (newly added)**. Similarly, at scaffolding stage, we evaluated three methods, including the use of linked reads to help scaffolding with "ARCS" tool, use of Hi-C reads to assist scaffolding with "SALSA", and the combination of the previous two methods. We concluded that the approach that used linked-reads scaffolding (by ARCS), followed by PacBio long read scaffolding (by SALSA) produced best scaffolded assembly (**as designated in a dotted box now**), and this was selected for phasing in the next step. In the phasing stage, we tried to use PacBio long reads alone, and also tried combining them with Oxford Nanopore long reads (in bam format). We found that the latter approach had better phasing results and have now indicated this with **another dotted box**. Because the assemblies generated by these alternative approaches were used in the analyses leading to the selection the highest quality assembly, they **should NOT be treated as "unused"**, and we have chosen to leave them in this study showing the study design.

We respectfully submit that the reviewer's statement regarding the phased genomes in this study is incorrect. As noted on page 6 in the **Results** section, we generated two haplotypes in fasta format and used dipcall to generate germline SNVs from diploid region of this genome ("**Overall study design and construction of a reference-grade personal genome assembly**").

Finally, Figure 1 has also been updated to reflect the inclusion of newly added analyses comparing somatic SVs in the personal genome and GRCh38. These studies were added in response to other review comments.

Page 8 line 6: "We successfully identified all the HLA genes and corresponding transcripts in HCC1395BL_v1.0"

Buried in the methods section (Page21 Line10-14) is the custom logic you needed for this full HLA identification to actually work. This should be mentioned in your results section. **Notwithstanding this, direct HLA determination without having to run a specialised HLA genotyper is a good result.**

[Response]

We appreciate the reviewer's understanding HLA gene analysis ("**direct HLA determination without having to run a specialised HLA genotyper is a good result**"). As suggested, we have revised the HLA section accordingly in **Results** section.

Page 11 line 6-7. This isn't quite as simple as you seem to imply. Yes, it's definitely better that you have the gene in the assembly, but converting from the personalised reference into a form in which the cancer knowledgebases can be used is a non-trivial operation. In some respects, alt-only genes are actually easier to deal with than having to deal with every gene being on a different scaffold for every patient and having to reidentify genes every time.

[Response]

Thank you very much for the point. We also recognize that translation of results using a personalized genome to a format that can be consumed by cancer knowledgebases or used in a clinical setting is a non-trivial task.

However, our study represents only the first step in demonstrating the potential that personalized genomes have as references for somatic mutation identification. As sequencing technology and assembly algorithms continue to evolve, *de novo* assembly of a personal genome to chromosome level might one day become a routine diagnostic step for every patient. We believe that the automated genome annotation with improved algorithms that support every gene in a personal genome will be available in near future. Comparing the cost and time to sequence and assemble a genome today versus 20 years ago, such a prediction is not unreasonable. We do not need to wait until all the components become perfect before we start to demonstrate the great potential of personal genomes for clinical study.

Methods require more detail to be reproducible. Software versions are included which is good but the methods don't specify what arguments/parameters/options were used for each step. For example, what were the canu command-line parameters? What did you specify as the target genome size to canu? Were you using an options file? If so, what was in it? You don't need to specify setting such as canu's cluster queuing overrides as they're environment-specific but settings that impact the assembly output need to be specified if your analysis is to be considered reproducible.

One way to make this information available without making the actual methods section itself much larger is to make the actual scripts you used to perform the analysis available either on a publicly visible github, or as supplementary materials in the manuscript.

Without this level of methods detail it is difficult to evaluate whether the methods are appropriate. For example, page22 line 4 says you use Assemblytics. You don't specify what you used as the input. The Assemblytics FAQ states: "Important: Use only contigs rather than scaffolds from the assembly. This will prevent false positives when the number of Ns in the scaffolded sequence does not match perfectly to the distance in the reference.". Did you use to scaffolded HCC1395BL_v1.0, or the raw contigs? This choice massively impacts your Fig 5F result but your methods section doesn't actually say which one you used which gives me little confidence in the Fig5F result as it's quite possibly an artifact of your pipeline. This level of methods detail is essential as even seemingly minor differences in a bioinformatics pipeline can have a very large impact on the results.

[Response]

Thank you very much for these important comments regarding the level of detail included in the **Methods** section. As suggested, we **have now provided additional details** to this section for all steps involved in assembly, evaluation, read mapping, and variant calling.

For example, for the canu contig assembly, we have provided the software parameters used as shown below:

```
canu -p hcc1395bl -d hcc1395bl_out genomeSize=3.1g useGrid=false maxThreads=16 corConcurrency=4 corThreads=4  
cormmapThreads=4 cormhapConcurrency=4 corovlConcurrency=4 maxMemory=360g correctedErrorRate=0.075 -pacbio-raw  
pacbio_reads.fa
```

Regarding the reviewer’s concern about the use of Assemblytics, we were fully aware of the need to use contigs (in fasta format) as an input for this tool. As stated in the Methods section, “Assembly-based SVs were generated from direct assembly comparisons of HCC1395 cancer cell line contig assembly with the *de novo* HCC1395BL assembly using paftools [44] and Assemblytics (<https://github.com/MariaNattestad/assemblytics>)”. To further reduce confusion, we have added additional details to the **Methods** section as below:

Assembly-based SVs were generated from direct comparisons of the contigs of tumor cell line (HCC1395) with the HCC1395BL_v1.0 reference (as opposed to GRCh38 reference) using paftools [53] and Assemblytics [43] (<https://github.com/MariaNattestad/assemblytics>; <http://assemblytics.com/>) with procedures suggested by their developers. For Assemblytics, we prepared the delta input file by aligning contigs fasta to a reference using MUMmer/nucmer and run with the options of “10,000” for “Unique sequence length required”, “20,000” for “Maximum variant size”, and “50” for “Minimum variant size”.

Are your PacBio reads HiFi reads? I assume they're not but you should explicitly mention this somewhere.

[Response]

The PacBio reads we used were PacBio subreads, not HiFi reads, as specified in the manuscript, in “**Whole genome sequencing datasets**” and **Suppl. Table 1** in the **Methods** section.

Page12 Line11. Does the MinKNOW base-calling software use have a version? Nanopore base calling is a very active area which significant improvements being made regularly (including recent improvements to Guppy).

[Response]

We thank the reviewer for the question about the MinKNOW version. MinKNOW is the MinION controller and we used Guppy version 2.1.3. In the revised manuscript, we have added the information about the version in the **Methods** section as shown below:

Raw sequence reads were called in real time by the MinION operating software MinKNOW (Guppy version 2.1.3).

One of the key questions regarding using a personalised reference genome is whether this obviates the need to actually perform tumour/normal variant calling and if calling variants directly from the personalised reference is sufficient to identify somatic mutations.

Do we still need to run tumour/normal variant callers such as Strelka2 or MuTect2, or can we get away with just running GATK and subtracting the germline heterozygous calls (or using a NovoBreak-style reference containing the ambiguous IUPAC nucleotide codes at heterozygous sites - e.g. germline A/T = W).

As you have already run germline GATK during the phase stage, it should be relatively straight-forward to run GATK on the tumour, subtract the germline calls, and compare the resultant somatic call set to the Strelka2/MuTect2 calls. Do these GATK calls have so much noise so as to make them unusable, or are they a viable alternative to the specialised paired tumour/normal callers when using a personalised reference genome?

[Response]

We thank the reviewer for raising this important question.

In response to the first part of the question, we find that **somatic mutations cannot be sufficiently identified when calling variants directly, due to the inclusion of many germline variants in the callset**. In our study, we answered this question indirectly through SV analysis. Our PacBio long-read and assembly-based SV analyses demonstrated the need to filter tumor SV calls with normal SV calls even when a personal reference is used (**Supplementary Table 11 and Supplementary Table 12, Supplementary Figure 8, Supplementary Figure 9**).

In response to the questions regarding the accuracy of somatic SNV discovery with GATK, we respectfully suggest that simple subtraction of $GATK_{normal}$ from $GATK_{tumor}$ alone will not be sufficient as compared to the results from a somatic mutation caller such as MuTect2. In our analysis, we went even further to filter out additional germline SNVs using freebayes calls on the normal sample, and **we still observed greater numbers of unique calls in the $GATK_{(Tumor - Normal)}$ set than the MuTect2 set**, as shown in the table below.

	GATK $(Tumor - Normal)$	MuTect2
Common	40171	40171
Unique	46100	17190
Total	86271	57361

In short, to obtain accurate somatic SNV/SV calls, we find that the best approach is still to perform tumor/normal paired variant calling using a specialized somatic caller, even when a personalized genome is available. If a somatic caller is not available, variant callset from tumor sample need to be filtered, at a minimum, with variant callsets from the normal sample in order to remove as many false positives (or "noise") as possible.

Similarly, the high quality reference genome you have created has high cost associated with its generation. How much better is this approach than a light-weight personalised germline one could create by just augmenting GRCh38 with the germline SNV/indels one could identify from short read sequencing of the normal? The biggest difference seems to be that the personalised reference does a

better job of calling somatic variants when there are nearby germline variants. Does doing something like creating a heterozygous reference (het A/T=W) and aligning with a IUPAC-aware aligner such as NovoBreak perform almost as well as a good personalised reference genome?

[Response]

We thank the reviewer for raising this great question. While it is true that cost of creating a personalized reference genome is higher than use of general reference, but the goal of scientific research is to find the truth, and we suggest that the former approach will more likely lead to discovery of “truth. As we initially demonstrated (as well as with a **newly added SV analysis in the revised version**), using the personal genome as reference not only improved the read mapping and somatic SNVs calling, but it also had a greater impact on somatic SV calling, particularly in reducing false positive SVs.

Sequencing technologies have evolved rapidly in last two decades, and the cost of short read sequencing has dropped substantially. Although the cost of accurate long read sequencing (e.g., PacBio’s CCS/HiFi technology) is currently still high, it is getting cheaper and cheaper as time goes by. Therefore, we suggest that in the near future, creating a personalized reference genome will cost less than it does today.

In regard to the suggestion “creating a heterozygous reference (het A/T=W) and aligning with an IUPAC-aware aligner”, we respectfully respond that we feel this is not within the scope of our current study. Thus, we have not included a comparison of mutation calling using a *de novo* assembled personal genome reference with a modified GRCh38 (augmenting with germline SNVs/InDels) in this revision, but, recognizing the possible significance of such an analysis, would consider doing so in a future study.

We also provided additional discussions related to the cost of creating personalized reference for somatic mutation detection in tumor-normal paired samples in **Discussions** section.

No details on how SNVs were matched is included in your manuscript. Is it possible that some of the missing SNVs are not actually missing, but are reported in a format that your analysis pipeline does not account for? VCF allows variants to be represented in a multitude of ways. Take the following example:

TAGTG ref

TTGAG sample

Can be represented a two SNVs:

2 A T

4 T A

Or could be represented as a single variant:

2 AGT TGA

If your analysis pipeline is just looking to see if there's a SNV call at position 4, you're going to incorrectly claim the variant is missed if it's represented in the latter notation.

Does your SNV analysis account for this? It's unclear as to what the difference between "Identical SNVs" and "Equivalent SNVs" are in Fig5C.

[Response]

We addressed this question in two ways in our revised version. In the first, we evaluated how the two SNV callsets were matched and compared using the same reference; and the second, we looked at how the SNVs generated based on GRCh38 were compared with the SNVs based on the personalized reference.

To compare the SNVs based on the same reference (e.g., MuTect2 and Strelka2 callsets), we used “**bcftools isec**” and “**rtg vcfeval**” to obtain the maximum numbers of overlapping variants. The software “**rtg vcfeval**” can recognize the same variant with different allele representations. **In short, we looked at more than the SNV positions.**

To compare GRCh38-based SNVs with personal genome-based SNVs, we need to use a different approach since we could not compare them directly. First, we extracted both the reference and alternate alleles of each SNV with their 50bps flanking sequences from GRCh38 in a fasta format (101 bps in total for each allele). We then used “**blastn**” to map those allele sequences to the personal genome reference HCC1395BL_v1.0. If the two alleles of the same SNV were found to be identical on HCC1395BL_v1.0 at the same location, this SNV was counted as “Identical SNVs”. If the two alleles of the same SNV were mapped at the same location on HCC1395BL_v1.0, but with some mismatches or germline SNVs on the flanking as described **in Suppl. Figure 5**, then we counted this SNV as “Equivalent SNVs”. Those missing SNVs were below the thresholds.

These mapping and comparison approaches are described in the “**Methods**” section.

The SV analysis is relatively cursory and does not appear to account for many of the complexities encountered with SV calling:

Coordinate space. The methods state that SURVIVOR was used for SV comparison but it is unclear how this was done without a common reference genome for the SVs. Were SVs translated to GRCh38 coordinate space? How?

[Response]

We did not perform coordinate transformations between the GRCh38-based SVs and personal genome-based SVs in our analysis as no credible software or method to do such SV transformation appears to exist. As the reviewer mentions, “SURVIVOR” does not have the ability to do cross-reference coordinate lifting. In our original manuscript, we stated that “**SURVIVOR**” was used for merging SV callsets only. In the revised manuscript, we have emphasized analyses with multiple SV callers in order to establish whether using a personal genome as reference is better for somatic SV calling, and therefore did not use “SURVIVOR” to merge multiple SV callsets.

Page 14 L14: This looks like a germline comparison. **What matter for a somatic reference is whether the somatic SV call set is better with the reference?** What is causing this bias between the ins/del length distribution of somatic calls (or is this a pseudo-germline comparison?)? If this bias is due to the GRCh38 deletion bias, shouldn't you find that the extra calls are overlap germline SVs w.r.t GRCh38? Is this the case?

[Response]

Thank the reviewer for raising the question, and agreed that we should have emphasized whether the somatic SV calling is better with a personal reference.

Previous studies showed that **GRCh38 reference has a genome-wide deletion bias**, suggesting the **systematic collapse of repeats** during its initial cloning and assembly [M. J. P. Chaisson et al, 2015, Resolving the complexity of the human genome using single-molecule sequencing. *Nature*. 517, 608–611]. The impact of this bias on germline SV detections is to increase the tendency to **call more insertions** [Miga, K.H., et al., 2020, *Telomere-to-telomere assembly of a complete human X chromosome*. *Nature*. 585, 79-84. Extended Data Fig. 3. and Zook, J.M. et al. 2020. A robust benchmark for detection of germline large deletions and insertions. *Nat Biotechnol* 38, 1347–1355. Fig. 3]. Ultimately, this will also affect somatic SV calling.

For somatic SVs, even though we no longer used INS/DEL balance as an indicator of improvement for SV calling, we still observed many more INSERTIONS being called by paf tools and Assemblytics on GRCh38 as opposed to the personal reference (**Supplementary Figure 9**). Even when applying more stringent filtering criteria, we continued observing many more INS calls by both Sniffles and PBSV with PacBio long reads when GRCh38 was used as reference as opposed to the personal reference (**Supplementary Table 11**).

Regarding the reviewer's comment, "If this bias is due to the GRCh38 deletion bias, shouldn't you find that the extra calls are overlap germline SVs w.r.t GRCh38", we suggest that one might see more overlapping INS with germline on GRCh38, but that the overlapping may **not be proportional** when comparing with the personal genome as reference. Using the Sniffles results as an example: On GRCh38, Normal INS = 1743, Tumor INS = 2120 and Somatic = 929; but on the personal reference, Normal INS = 282, Tumor = 746, Somatic = 624 (**Supplementary Table 11 and Supplementary Table 12**). The observation clearly illustrates that most Tumor-INS calls on the personal reference were "somatic", but more than half of Tumor-INS calls on GRCh38 were filtered out.

We also provided additional discussions related to genome-wide deletion bias in GRCh38 in **Discussions** section.

Is the bias caused by microsatellites? Microsatellite instability is important in cancer genomics. This manuscript does not indicate how much microsatellite calling is improved by a personalised reference. Does a personalised reference allow microsatellite instability to be called by standard tools, or are MSI callers still needed? Are MSI calls better with the personalised reference? Does the fact that the personalised reference is haploid negate any potential MSI benefit?

[Response]

We agree with the reviewer that microsatellite instability (MSI) is important in cancer genomics. However, MSI calling was neither in the original scope nor the focus of this study. We did not add a microsatellite analysis in this revision, but may perform such analysis in a future study.

P15L4: "assembly of the HCC1395 cancer cell line" what assembly of HCC1395 is this referring to? I might have missed it but I can't find anywhere in the methods where you state that you have performed assembly on the tumour cell line. Figure 1 lists "HCC1395 contigs" but you don't say how these were generated. Are they PacBio contigs? Chromium?

[Response]

We performed an assembly of the tumor HCC13965 cell line using Supernova for 10X linked reads and canu for PacBio long reads, respectively, as described in both **Results** and **Methods** sections.

To call attention to this work, we have enhanced the description in **Results** section related to the assembly of HCC1395 cancer cell line:

(1) We have modified **Figure 1** to make clearer the work to generate assemblies of HCC1395 tumor cell lines using 10X linked reads and PacBio long reads. We have also provided evaluations of the HCC1395 assemblies in **Suppl. Figure 1, Suppl. Figure 2, Supple. Figure 3, and Suppl. Table 2**. Our evaluation demonstrated that the PacBio assembly (using canu) had the superior quality.

(2) We have also added a separate section discussing SV discovery by comparing HCC1395 assembly to HCC1395BL_v1.0 assembly (“**SV discovery using assembly-to-assembly mapping approach is improved using personalized genome reference**”).

I'm not sure what you're trying to achieve with your comparison of HCC1385BL_v1.0 with HCC1385. If this is a somatic analysis, why do you find Alu and L1 repeat peaks? I'd expect to find those for comparisons of germline SV call sets but not for somatic. Even L1

[Response]

We thank the reviewer for the opportunity to explain this analysis. Comparing the contig assembly of the tumor cell line HCC1395 with a reference genome (GRCh38 or HCC1395BL_v1.0) allows for detection of SVs using assembly-to-assembly based tools (paftools and Assemblytics). Such SV analysis have been used in many different studies for germline [Zook, J.M., et al., *A robust benchmark for detection of germline large deletions and insertions*. **Nat Biotechnol**, 2020. **38**(11): p. 1347-1355] and cancer cell line SV discovery [Nattestad, M., et al., *Complex rearrangements and oncogene amplifications revealed by long-read DNA and RNA sequencing of a breast cancer cell line*. **Genome Res**, 2018. **28**(8): p. 1126-1135]. In the revised manuscript, we used paftools and Assemblytics to generate initial SVs, then filtered with SVs from normal samples to obtain “somatic” SVs (see **Results** section “**SV discovery using assembly-to-assembly mapping approach were improved using personalized genome reference**”). However, we no longer used INS/DEL balance as an indicator of improvement due to the changes of filtering criteria for Sniffles and PBSV.

With respect to *Alu/L1*, *L1* and *Alu* are among the most prevalent mobile elements in human genome, constituting roughly 17% and 11% of the genome sequence respectively. *L1* and *Alu* mediated insertional mutagenesis and recombination are associated with genetic instability, one of the hallmarks of cancer. Traditionally, these mobile elements were believed to be active in germline, but largely dormant in somatic tissues. However, recent studies revealed increased activities of these elements in multiple cancer tissues. It has been shown that somatic retrotransposition of *L1* and *Alu* is possible and could potentially contribute to tumor genome dynamics. A recent study identified several *L1* and *Alu* insertion events occurring during the tumor progression in breast cancer patients. Therefore, we suggest that it is not surprising to see *Alu* and *L1* insertion/deletion events in the somatic analysis [Deininger, P. *Alu* elements: know the SINEs. **Genome Biol** **12**, 236 (2011); Ade C, Roy-Engel AM, Deininger PL. *Alu* elements: an intrinsic source of human genome instability. **Curr Opin Virol**. 2013 Dec;3(6):639-45; Scott EC, Devine SE. The Role of Somatic L1 Retrotransposition in Human Cancers. **Viruses**. 2017 May 31;9(6):131; Steely, C.J., Russell, K.L., Feusier, J.E. *et al*. Mobile element insertions and associated structural variants in longitudinal breast cancer samples. **Sci Rep** **11**, 13020 (2021).]

One issue not addressed but critical to SV interpretation is the notational difference between SV call sets. The most common is notational differences between insertions and duplication. Take the following three VCF SV calls:

```
TACGTAGTG ref
```

```
TACGACGTGTG alt
```

```
1 T TACG
```

```
1 T <DUP> SVTYPE=DUP;SVLEN=3
```

```
1 T  SVTYPE=INS;SVLEN=3
```

All three VCF calls are representation of exactly the same event. Even more messy is that the calls don't have to be (and frequently aren't) left-aligned:

```
4 A ACGA
```

```
2 A  SVTYPE=INS;SVLEN=3
```

```
3 C  SVTYPE=INS;SVLEN=3
```

```
4 G  SVTYPE=INS;SVLEN=3
```

Can also be used in VCF to represent exactly the same event. SURVIVOR does not handle this. How have you handled the potential for representational difference between your tools confounding your analysis? Can the imbalance between INS and DEL be explained by the callers reporting INS as DUP events?

[Response]

We thank the reviewer for raising this point. We were aware that a single SV event might have multiple representations in VCFs generated from different callers. In fact, we observed that paftools might call one SV event as "Deletion", but Assemblytics would call the same event as a repeat/tandem contraction, and that paftools would call another SV event as an "Insertion", but that Assemblytics would call the same event as a repeat/tandem expansion. Sniffles and PBSV sometimes called DUP as INS, or INS as DUP. Therefore, to account for these differences in calling, we treated "Insertion", "INS", "DUP", "Repeat_expansion", and "Tandem_expansion" as the same INS type, and treated "Deletion", "DEL", "Repeat_contraction", and "Tandem_contraction" as the same DEL type during the analysis of INS/DEL length distribution. Therefore, INS/DUP miscalling associated with aligner origin was not an issue for us.

However, in revising the manuscript in response to another reviewer, we have re-analyzed the PacBio data using more conservative filtering (e.g., excluding all SVs with "IMPRECISE" status for both Sniffles and PBSV, and many associated with INS) to minimize the differences between the Sniffles and PBSV calls, and not using INS/DEL balance as an indicator of the improvement for SV calling. This updated approach enables us to better address the question of whether using a personal genome reference improve SV calling.

As pointed out by the reviewer, SURVIVOR does not have ability to recognize the same SVs that are represented differently, but it can merge the SVs if using "NOT take the SV type into account" in the fourth argument (<https://github.com/fritzsedlazeck/SURVIVOR/wiki>). In our original analysis, we only used SURVIVOR to merge SVs of the same type (INS or DEL). In the revision, we did not use SURVIVOR to generate an integrated SVs set as we felt it to be out of scope for this study.

What about complex rearrangements? Somatic SV that form part of complex rearrangements are extremely clustered (doi.org/10.1101/2020.07.09.196527). Does this impact somatic SV calling? There are at least 5 fusion genes in HC1395 (PMID 21808235)(or are there really?) so there's probably at least one instance of chromothripsis in there. How do the two references compare on terms of complex rearrangement resolution?

[Response]

Thank the reviewer for raising this question. Complex rearrangements were not a focus of our study/evaluation in this manuscript, and we have not included their analysis in the revised manuscript either. Given the possible scale of such analysis, we submit that such work would be better suited to a more complete description in a separate manuscript. We may perform such an analysis in a future study.

Overall, the SV results presented don't seem answer the key question: Does using a personalised reference improve somatic SV calling? There's no comparison of short read SV calling on HCC1395BL_v1.0 vs GRCh38. There's no evaluation of CNV caller on the two references. I can't seem to find any analysis of somatic SV calls that includes the critically important inter-chromosomal SVs.

[Response]

We thank the reviewer for this question. In this revision, we have performed somatic SVs analysis with short reads using 4 SV callers, including **Pindel, BreakDancer, Delly, and novoBreak** (see **Results** section "Detections of somatic SVs using short reads were improved using personalized genome reference"). We also re-analyzed PacBio long reads for SV calling using Sniffles and PBSV with more stringent criteria (e.g., excluding all the calls with IMPRECISE tags) (see **Results** section "Detections of somatic SVs using long reads were improved using personalized genome reference"). Assembly-to-assembly based SVs were also re-analyzed using different filtering strategies for somatic SVs identification (see **Results** section "SV discovery using assembly-to-assembly mapping approach was improved using personalized genome reference"). **All the SV analyses demonstrated that the personal genome reference enabled more accurate SV calling, particularly reducing false positives. SV calls were more consistent between different callers.**

Regarding inter-chromosomal SV events, BreakDancer, Delly and novoBreak callers were all reporting inter-chromosomal translocations (CTX for BreakDancer in **Figure 6B**, TRA for novoBreak in **Supplementary Figure 7A**, BND for Delly in **Supplementary Figure 7B**); Pindel did not.

For somatic CNVs using short-reads, we first tested Illumina's DRAGEN somatic CNV caller (https://support.illumina.com/content/dam/illumina-support/help/Illumina_DRAGEN_Bio_IT_Platform_v3_7_1000000141465/Content/SW/Informatics/Dragen/SomaticCNVCalling_fDG.htm), but found that it was unable to use personal human genome references. We communicated with the DRAGEN development team regarding enabling CNV calling with personal genome reference, but such updates will not be ready in near future. Thus, it is currently impossible for us to perform such an evaluation.

We next selected the Control-freec CNV caller to call somatic CNV Gain and Loss using short reads, due to its flexibility to handle both standard (GRCh38) and non-GRC human genome references (HCC1395BL_v1.0). However, we discovered a major flaw in the Control-freec CNV results, namely that a fair amount of CNVs crossed known gap regions in GRCh38; the presence of these CNVs makes the

comparison less meaningful. We are working with the Control-freec team to solve this challenging problem, but the resolution is expected to take some time. Therefore, to maintain the accuracy and intactness of our analyses/results, we have chosen to exclude all CNV analyses in this manuscript. We will consider continuing such analyses in future studies.

Additionally, we tested another CNV caller, HATCHet, which we found, like DRAGEN, could not handle non-GRC human genome references such as our personal genome reference.

You should split out the soft-clipping and hard-clipping into mutually counts of partially aligned reads (soft-clipped reads without a SA tag), and split-read alignments (reads with SA tag. Bwa uses soft-clipping for the primary alignment, and hard-clipping at all supplementary alignments). This will prevent double (or even triple) counting of chimerically aligned reads and better corresponds to what the aligner is actually doing. The SAM specifications use the term "chimeric" read alignment but "split read" alignment is the term in widespread usage outside the specifications document.

There are 4 scenarios for read pair alignments:

- 1) Aligned with orientation and distance expected by library fragment size distribution (aka concordantly aligned read pair; proper pair flag usually set by aligner)
- 2) Both aligned but not concordantly (aka, discordant read pair)
- 3) One read in pair aligned (aka OEA)
- 4) Both reads unaligned

It's unclear which of these categories you are grouping in each of your counts. Specially, it's unclear if 3)/OEA reads are considered "Reads with improper-pairing" or not. Do you just use the proper pair SAM flag as reported by bwa? To be technically correct you also need to account for when one or both of the reads in the read pair are split-read aligned (as will be the case when either read overlaps a true SV) but even most SV tool authors don't correctly account for this.

[Response]

We appreciate the reviewer asking this question. We have expanded our short-read alignment analysis to cover all 12 paired tumor/normal WGS replicates, and the statistics include non-properly paired reads, mismatches, standard deviations of library insert sizes, read coverage standard deviations, split-reads statistics, soft-clipping with/without an SA-tags, so on and so forth. As suggested by the reviewer, we **added split-read statistics (Figure 4D: Reductions of split-reads in alignments for tumor and normal samples with personalized genome HCC1395BL_v1.0 as opposed to GRCh38)**, and soft-clipping reads without SA tags (**Supplementary Figure 4A: Percentages of reductions of soft-clip reads without SA tags on HCC1395BL_v1.0 as opposed to GRCh38**) in this revision.

As described in the **Methods**, for non-properly (or improperly) paired reads statistics, we relied on the aligner (**BWA-MEM**) to correctly set the bit for properly aligned reads, and the tool "**samtools stats**" to collect read mapping information from the aligned bam (by BWA-MEM), including "reads mapped", "reads unmapped", "reads mapped and paired" and "reads properly paired". In our analysis, the numbers of "non-properly paired reads" were the subtraction of "reads properly paired" from the "reads mapped and paired" as stated in **Methods (Figure 4B)**. In the case of split read alignments, one and only one alignment was the primary and soft clipped, per default with BWA-MEM; other lines were tagged with the 0x800 SAM flag (supplementary alignment) and hard clipped (<https://github.com/lh3/bwa/tree/mem>, FAQs #2). Since the bit (flag 0x2) for properly paired reads was

[Response]

We are grateful to the reviewer for their many excellent comments and questions, and appreciate their understanding that not all of the suggested analyses will have a significant impact on the findings, and that addressing all of them would substantially increase the length of the manuscript. As a consequence, we have only addressed the most relevant analyses in our resubmission. To summarize, **we have improved our manuscript in the following:**

- (1) Expanded short-read alignment analysis to cover all 12 paired tumor/normal WGS replicates, including improperly paired reads, mismatches, standard deviations of library insert sizes, read coverage standard deviations, split-reads statistics, soft-clipping with/without an SA-tags, so on and so forth. Split-reads and improperly paired reads were related to SV calling.
- (2) Expanded somatic SNVs analysis to cover all 12 paired tumor-normal WGS replicates.
- (3) Performed PCR validation using Sanger sequencing for a subset of somatic SNVs that were only discovered using a personal genome as reference.
- (4) Added somatic SV analysis with short reads using 4 SV callers, including Pindel, BreakDancer, Delly, and novoBreak.
- (5) Added long-read alignment statistics and comparison for both tumor and normal samples.
- (6) Re-analyzed PacBio long-read data for somatic SVs using Sniffles and PBSV with more stringent criteria, emphasizing the question of whether using a personal genome reference would make SV calling better.
- (7) Re-analyzed assembly-to-assembly based somatic SVs using Assemblytics and Paftools by comparing the tumor HCC1395 contig assembly to normal HCC1395BL_v1.0, focusing on the question of whether using a personal genome reference would make SV calling better, not relying on INS/DEL balance as an indicator of improving SV calling.
- (8) Performed somatic CNV analysis using short reads with DRAGEN, Control-freec, and HATCHet, but only included in the response to the reviewer, not in revised manuscript.
- (9) Performed $(GATK_{\text{tumor}} - GATK_{\text{normal}})$ using short-reads and compared to somatic SNV caller MuTect2, but only included in the response to the reviewer, not in the revised manuscript.
- (10) Enhanced the description for assembly of HCC1395 tumor cell line in **Results** section
- (11) Added more detailed descriptions in the **Methods** section for all the steps involved in assembly, evaluation, read mapping, and variant calling.
- (12) Revised the **Abstract**, **Discussions** sections accordingly in response to reviewers' suggestions.

We hope all these further analyses/improvements have satisfied **reviewer #1**.

Reviewer #2: The paper explores the use of denovo assembled genomes for detection of somatic mutations in tumor-normal paired samples.

This work builds upon recent work (e.g. genomebiology.biomedcentral.com/articles/10.1186/s13059-020-02047-7) demonstrating the use of a denovo assembled genome as opposed to using the reference human genome. The authors tackle an important question but the conclusions of the paper are not well supported by the results that are presented. My main comments about the paper are as follows:

1. Most of the results (Figure 2-3, Table 1) are for the normal genome and are applicable to any genome, i.e. for demonstrating the accuracy and quality of a de novo assembly of any human genome. Similarly, the results for the de novo assembly of the normal cell line are very similar to what was done in a recent paper published in Genome Biology for the HG002/Ashkenazi Jewish genome (<https://genomebiology.biomedcentral.com/articles/10.1186/s13059-020-02047-7>). Since the goal of the paper is to demonstrate the utility of de novo assembly based analysis for tumor-normal sequencing. I would have expected to see more results specifically focused on cancer relevant analysis. These would differentiate this work from previous published results.

[Response]

We thank the reviewer for the point. The first step in evaluating how a personal genome reference impacts somatic mutation calling, is the assembly of a high-quality personal genome. We submit that the methods and data used in assembly generation in our work differs substantially from that in the referenced HG002/Ash1 publication. The HG002/Ash1 paper (<https://genomebiology.biomedcentral.com/articles/10.1186/s13059-020-02047-7>) combined three different datasets, including Illumina, ONT, and the PacBio HiFi reads together and used MaSuRCA v3.3.4 to create an initial contig assembly, then used its own scaffolding and gap-filling strategies to create the final Ash1 assembly. However, in our study, we created two initial assemblies for each cell line (tumor and normal), including Supernova for 10X linked reads, and canu for PacBio long reads. Illumina short reads were used for polishing. We used linked reads (using ARCS tool) for initial scaffolding, followed by use of Hi-C reads (using SALSA tool) to create the final scaffolded assembly. We also created phased assemblies for the normal cell line (unlike Ash1, which was not phased). Also, ONT data was only used for better phasing in our study when combined with PacBio long reads.

Furthermore, our work extended beyond genome assembly, and differs from Ash1 in many aspects, including, but not limited to, the following:

- (1) The goal of our study was completely different from Ash1 paper, which focused primarily assembly and annotation. Although the Ash1 paper also included germline variant calling on Ash1, the analysis was geared more for population study or germline analysis. In contrast, in our study, we performed assembly and annotation for **BOTH normal and tumor** cell lines, and the **somatic mutation analysis** was a focus of our work.
- (2) We assembled the genomes of two cell lines (tumor-normal paired cell lines) that were derived from **the same individual**, but the Ash1 paper only presented the assembly of a single germline individual genome (HG002).
- (3) We evaluated somatic calling based on both GRCh38 and a personal genome using data generated from tumor-normal paired samples (same person), while the Ash1 paper tested an unrelated individual (66% Ashkenazi individual PGP17) on ash1 for germline variants.

- (4) Somatic mutation identification is a distinct analysis from germline variant detection, and we submit that it is much more challenging to accurately detect cancer somatic mutations.
- (5) The application of our study differs from the Ash1 work. Our analysis sought to demonstrate the use of a personalized genome as reference for personalized somatic mutation detection.
- (6) We assembled the genomes from both normal cell line and tumor cell line that were derived from the same person, and we also evaluated and compared these two assemblies and their genome annotations.
- (7) We evaluated the impact of both reference genomes for both short-read mapping and long-read mapping.
- (8) We evaluated the impact of both reference genomes on somatic SNVs and somatic SVs calling.
- (9) We also evaluated how tumor contig assembly, as compared to reference genomes, affected assembly-based SV identification.

Assembling the genomes of tumor and normal samples was only the starting point in our study, and served as the basis for the subsequent read mappings and variant analysis/comparison. We therefore respectfully argue that our study/manuscript is very different from the Ash1 paper.

In our revised manuscript, we have added new analyses (as well as re-analyses) to emphasize the question of whether using a personal genome reference would make somatic SNV/SV calling better. To summarize, **we have improved our manuscript in the following:**

- (1) Expanded short-read alignment analysis to cover all 12 paired tumor/normal WGS replicates, including improperly paired reads, mismatches, standard deviations of library insert sizes, read coverage standard deviations, split-reads statistics, soft-clipping with/without an SA-tags, so on and so forth. Split-reads and improperly paired reads were related to SV calling.
- (2) Expanded somatic SNVs analysis to cover all 12 paired tumor-normal WGS replicates.
- (3) Performed PCR validation using Sanger sequencing for a subset of somatic SNVs that were only discovered using a personal genome as reference.
- (4) Added somatic SV analysis with short reads using 4 SV callers, including Pindel, BreakDancer, Delly, and novoBreak.
- (5) Added long-read alignment statistics and comparison for both tumor and normal samples.
- (6) Re-analyzed PacBio long-read data for somatic SVs using Sniffles and PBSV with more stringent criteria, emphasizing the question of whether using a personal genome reference would make SV calling better.
- (7) Re-analyzed assembly-to-assembly based somatic SVs using Assemblytics and Paftools by comparing the tumor HCC1395 contig assembly to normal HCC1395BL_v1.0, focusing on the question of whether using a personal genome reference would make SV calling better.

All our somatic SNV/SV analyses indicated that personal genome reference enabled more accurate SNV/SV calling, particularly in reducing false positives. Somatic SNV/SV calls were found more consistent between different callers.

We hope all these cancer somatic SNV/SV results would satisfy **reviewer #2** regarding “I would have expected to see more results specifically focused on cancer relevant analysis”.

2. Along the same lines as the previous comment, the authors looked specifically at the) HLA locus. This makes sense since this is one of the most diverge regions in the genome and can be resolved via de novo assembly. However, it would be beneficial to demonstrate the value of assembly for genomic loci relevant to cancer. At the very end of the results, a whole section is devoted to analyzing the mitochondrial genome. The mitochondrial genome is 16 kb in length and not difficult to assemble using long reads. What additional value do the long reads provide compared to Illumina sequencing.

[Response]

We thank the reviewer for the question. Both reviewers of the manuscript recognized the importance of correctly assembling HLA locus, due to the extensive diversity it exhibits in human population. Our goal in presenting the HLA was to demonstrate that our assembly process performed well in complex genomic regions such as this. Although the induction and regulation of immune responses are the most important roles played by HLA-encoded molecules, *HLA* mutations have also been implicated as a mechanism of immune evasion during tumorigenesis (Castro A, Ozturk K, Pyke RM, Xian S, Zanetti M, Carter H. *Elevated neoantigen levels in tumors with somatic mutations in the HLA-A, HLA-B, HLA-C and B2M genes*. **BMC Med Genomics**. 2019 Jul 25;12(Suppl 6):107. PMID: 31345234). Also, HLA Class I gene mutations have been reported in many cancers, including the head and neck cancers, squamous cell lung cancer, and stomach cancers (Schaafsma E, Fugle CM, Wang X, Cheng C. *Pan-cancer association of HLA gene expression with cancer prognosis and immunotherapy efficacy*. **Br J Cancer**. 2021 Aug;125(3):422-432. PMID: 33981015). Thus, because **HLA genomic loci are also relevant to cancer**, we felt it important to include analysis of this locus in our manuscript.

Although mitochondrial genome assemblies may be considered relatively small in size, a previous report (Mai, Z., et al., *Misassembly of long reads undermines de novo-assembled ethnicity-specific genomes: validation in a Chinese Han population*. **Hum Genet**, 2019. **138**(7): p. 757-769) showed that such genomes may still be misassembled or fragmented, even when generated from long reads. In our study, we simply confirmed that we correctly assembled the mitochondrial genomes for the tumor and normal cell lines, and subsequently identified 3 somatic variants. In the revised manuscript, we have shortened the corresponding text and moved it to the end of **Results** section ("**Overall study design and construction of a reference-grade personal genome assembly**").

3. Very little space is devoted to the de novo assembly of the HCC1395 cancer cell line apart from one paragraph on page 6. The paper would be strengthened if something valuable can be demonstrated from the assembly of the cancer cell line.

[Response]

We thank the reviewer for the great suggestion regarding the assembly of the cancer cell line. We have enhanced the description in **Results** section related to the assembly of HCC1395 cancer cell line as follows:

(1) We have modified **Figure 1** to make clear our assembly of the HCC1395 tumor cell line using 10X linked reads and PacBio long reads. We also provide evaluations of the HCC1395 assembly in **Suppl. Figure 1, Suppl. Figure 2, Suppl. Figure 3, and Suppl. Table 2**.

(2) We have added a separate section regarding SV discovery involving comparison of the HCC1395 assembly to HCC1395BL_v1.0 assembly ("**SV discovery using assembly-to-assembly mapping approach**").

were improved using personalized genome reference”), using paftools and Assemblytics based on mapping HCC1395 contigs onto HCC1305BL_v1.0 as opposed to GRCh38. This work emphasizes the question of whether using a personal genome reference would make SV calling better.

4. On page 14, the authors state that "suggesting that some meaningful somatic SNVs could be overlooked when using GRCh38 as the reference genome". Again, if the authors could provide more details demonstrating that some of these somatic mutations identified only using the assembly based approach are relevant for breast cancer or cancer, it would provide support to the conclusions of the paper.

[Response]

We thank the reviewer for this suggestion. In our work (**Results** section “**Personalized genome reference allowed more accurate detection of somatic SNV mutations using short reads**”), we used KEGG pathway enrichment analysis (**Figure 5D**) to identify the pathways in which genes having SNVs identified using personal genome as reference (and without equivalent GRCh38-based SNVs), are involved. Several of these pathways are relevant to cancers. For example, GTF2H2 (General transcription factor IIH subunit 2) encodes the subunit of RNA polymerase II transcription initiation factor IIH, which is involved in both basal transcription and nucleotide excision repair (**Supplementary Table 7**). PTPN13 (Protein tyrosine phosphatase non-receptor type 13) encodes a signaling molecule that belongs to the protein tyrosine phosphatase (PTP) family, which regulates a variety of cellular processes such as cell growth, differentiation, mitotic cycle, and oncogenic transformation (**Supplementary Table 7**).

We also wish to emphasize that the main focus of our study was to evaluate the impacts of using personal genome reference, as opposed to GRCh38, on calling somatic mutations, not to identify any specific somatic mutations that associated to breast tumorigenesis. **As we demonstrated collectively in this study, using a personalized genome as reference not only improved short/long read mapping, but also improved somatic SNV/SV identification** (**Results** section “Personal genome assembly provides easily accessed sample-specific haplotypes for clinically relevant genomic regions”, “Use of a personalized genome as reference improved read mappings for both short reads and long reads”, “Use of a personalized genome reference allowed more accurate detection of somatic SNV mutations using short reads”, “Detection of somatic SVs using short reads was improved using a personalized genome as reference”, “Detection of somatic SVs using long reads was improved using a personalized genome reference”, and “SV discovery using an assembly-to-assembly mapping approach was improved using a personalized genome reference”).

Minor comments:

- In Figure 1, it is not mentioned whether PacBio sequencing was done using CLR or the HiFi technology.

[Response]

Our PacBio sequencing did not use HiFi technology, and the PacBio long reads were the PacBio’s “subreads”, as described in “Whole genome sequencing datasets” and **Suppl. Table 1** in the **Methods** section.

In summary, we wish to thank the reviewer for their excellent comments and questions that have helped us improve our manuscript.

Based on our analysis in this study, we find that the benefits of using personal genome reference vs. GRCh38 for tumor-normal paired somatic analysis included the following elements:

- (1) The personal assembly encompasses the individual specific haplotypes, and has better representations of the clinically important genes; and in particular, there is no need to deal with ALT loci in NGS secondary analysis.
- (2) Short-read mapping has been improved in terms of smaller standard deviations of both read coverages and library-insert sizes, fewer improperly paired reads, split-reads, soft-/hard-clipped reads, so on and so forth. Short reads were more uniformly placed on personal genome reference.
- (3) Long-read mapping has been improved with higher numbers of reads being mapped, fewer mismatches, and smaller standard deviations of read coverages. Long reads were more uniformly mapped with personal genome reference.
- (4) Most importantly, our analysis revealed that a personal genome reference improved somatic SNV and SV detection using short reads, particularly with respect to reducing false positives. Some of the somatic SNVs were only discovered using a personal genome as reference and validated, and some of them were related to biological pathways. Somatic SNV/SV callsets were found more consistent between different callers.
- (5) Use of a personal genome reference also improved long read SV analysis and assembly-based SV analysis, particularly in reducing the false positives. Somatic SV calls were found more consistent between different SV callers.

Together, the results we observed and presented in the manuscript, make us confident that use of a personal genome as reference for somatic mutation detection for tumor-normal paired samples prevailed over the use of GRCh38 as reference. We hope we have satisfied **reviewer #2**.

Reviewer #3: Xiao et al. describe the use of a specifically assembled personal genome to compare it with a tumor genome of the same patient, both produced from cell lines. Use of personalized genome sounds promising, and it is interesting to know how results would compare to a standard reference. Hence it's disappointing that the authors did not demonstrate this advantage convincingly. Eg, the abstract only provides improvements in terms of reduction of "improper-pairings, soft-clippings and hard-clippings" [sic], which is a very indirect measure at best. Only the SNV analysis is performed but the SV work is very inadequate. There is also lack of clarity in multiple cases with the data, figures, tables.

WRITING: Text is often unclear. At the simplest level, it is problematic English, typos and awkward phrasing are quite frequent, here are some examples (there are at least twice as many across the whole paper):

caner

medical relevant genes

near-completely

High confident

The numbers of reads with improper-pairings, soft-clippings and hard-clippings were dropped 88.25% of GRCh38-based somatic SNVs were found identical to those identified on personalized genome, whereas 9.59% of GRCh38-based somatic SNVs were considered as equivalent to those somatic SNVs identified on personalized genome but with germline SNVs on their flanking.

somatic variations

have strong evidence supports

were covered with yellow. (and more errors in Figure 2 caption) Based on our analysis, high quality assembly has been achieved based on our existing data for For the first time to our knowledge, we were able to identify

[Response]

We thank the reviewer for the comments regarding the writing. We have made the corrections and revised the manuscript accordingly.

A COUPLE OF MINOR POINTS:

Using % vs number in Abstract is poor style as it makes it harder to compare the numbers.

In Supplementary Tables, why are the captions above the tables? In the rest of the paper and Supplementary Figures, captions are below the figures and table.

[Response]

We appreciate the feedback on the abstract and table caption. We have revised the abstract accordingly, and the captions for Supplementary Tables are now below the tables.

MAJOR ISSUES:

There are multiple major concerns with this paper.

Two refs [46-47] are oddly inserted in the beginning without properly renumbering them. These papers have not been published, yet they are used referring to the "well-studied reference samples by FDA-led SEQC-II consortium", the main datasets here, normal (BL) and cancer (95).

It's hard to guess what information is in those papers in press, and what is the overlap with this study. The authors need to make this really clear, to avoid concerns about the novelty of their statements. Eg, statements like "We first obtained massive whole genome sequencing data using multiple sequencing technologies" should clearly indicate that only a fraction such data (long range) was generated in this work, and the use of this fraction is extremely poorly described.

[Response]

We apologize for any confusion caused by these references. The software we used to manage references (EndNote) did not support bioRxiv articles. Since the original submission of the manuscript, the two references (46 and 47) have been published in **Nature Biotechnology** (September 2021), and these two articles have now been correctly numbered in the revised manuscript (as ref5 and ref6).

Fang, L.T., et al., *Establishing community reference samples, data and call sets for benchmarking cancer mutation detection using whole-genome sequencing*. **Nat Biotechnol**, 2021. **39**(9): p. 1151-1160.

Xiao, W., et al., *Toward best practice in cancer mutation detection with whole-genome and whole-exome sequencing*. **Nat Biotechnol**, 2021. **39**(9): p. 1141-1150.

We have listed all data used in this study in the **Methods** section ("**Whole genome sequencing datasets**"). Indeed, all short-read data, 10X genomics linked-read data, and PacBio long-read data used in our analysis were part of the studies in ref5 and ref6. However, the Hi-C data and Nanopore data were newly generated for the analyses presented in this manuscript. The PacBio long-read data were used for *de novo* assembly (and SV analysis), while the short-read data were used for assembly polishing and variant analysis (**Supplementary Table 1**). In the revised manuscript, we expanded the short-read alignment analysis to cover all 12 paired tumor/normal WGS replicates, including improperly paired reads, mismatches, standard deviations of library insert sizes, read coverage standard deviations, split-reads statistics, soft-clipping with/without an SA-tags, so on and so forth. Split-reads and improperly paired reads were related to SV calling. We also expanded somatic SNVs analysis to cover all 12 paired tumor-normal WGS replicates.

I found 2 biorxiv entries with presumably earlier versions of those 2 papers and I considered them during this review. I'll refer to them as 46 and 47.

[46] consider 21 pairs of replicates and [47] - 42 datasets (21x2, I suppose?) of short-read NGS from 6 seq centers for the same tumor-normal pair, using 3 aligners and different numbers of SNV callers, sometimes producing artificial mixtures of both genomes or adding artificial sequence data. Both have important results regarding comparisons of different protocols and bioinformatic pipelines, or a finding that WES was less reproducible than WGS.

[46] also focuses on SNVs and **states "Larger structural variants and copy number analysis will be included in a separate manuscript that will discuss these findings in greater detail."** I assume the **current paper is that separate manuscript**, as it touches upon structural variants. So the first question (Q1) is: how good is SV calling here, with all the heavy artillery (seq machinery) thrown at these genomes? I don't think an answer "it will be shown in yet another paper" will do here.

[Response]

Indeed, there were separate papers focusing on the generation of high-confidence SV and CNV callsets for these paired cell lines, using integrated approaches. However, **the generation of such integrated callsets was considered out of scope for our study**, as it would have involved completely different methodologies/strategies to integrate all kinds of callsets based on several sequencing technologies.

While we also acknowledge that SV analysis is important, it is just one element contributing to the conclusion of this study. Other important elements included assembly, scaffolding, phasing, genome annotation, short-read mapping, long-read mapping, somatic SNVs calling and comparison, long-read SV calling and comparison, and assembly-based SVs calling and comparison.

It would be overwhelming to address all the important topics in a single manuscript. However, we do plan to expand our scope and to include more of the topics that were recommended by the reviewers using newer sequencing technologies in near future.

Further, in this m/s, long-range datasets have been added and the variants are called not with the standard GRCH38 but with a newly assembled personal genome. The arguments provided against the use of the standard are very reasonable and we are all intrigued how this approach performs. Thus the second question (Q2) is: how much better are the variant calls with personal genome versus the standard?

[Response]

We thank the reviewer for the comment and appreciate their acknowledgement of our argument regarding the use of a personal genome as opposed to the standard GRCh38 as reference being “**very reasonable**”.

To clarify further, in this manuscript, we called variants using BOTH the GRCh38 and newly assembled personal genome (HCC1395BL_v1.0) as references, and then compared the results.

Q1. Indeed, the paper touches upon structural variants. However, it touches upon them in a rather superficial way and not in sufficient depth. And that's really disappointing: **while SNV calling is relatively polished, SV detection could really benefit from the use of personal genome**. In my view, they simply have not done the job here, in terms of both calling/analysis of SV and their interpretation. There are multiple concerns:

[Response]

We thank the reviewer for this feedback and remark (“**SNV calling is relatively polished**”). In the section below, we address the various concerns surrounding SV calling.

1. PacBio: why almost the same number of reads results in 14% drop in coverage depth in 95 sample, while the discrepancy is not as bad in short reads and 10x (sup tab1)? It also has consequences for downstream analyses and must be carefully explained.

[Response]

In **Supplementary Table 1**, the mean coverage for normal sample (HCC1395BL) was 53.54x, and the mean coverage for tumor sample (HCC1395) was 46.88x. The read counts for HCC1395 were slightly greater than that for HCC1395BL, but the average length (8,146 bps) for HCC1395 was shorter than that for HCC1395BL (9,089 bps). Therefore, for HCC1395, the mean coverage was about: $(17,265,341 \text{ reads} \times 8,146 \text{ bps per read}) / 3,000,000,000$ (as genome size) = 46.88x. Similarly, for HCC1395BL, the mean coverage was about: $(17,671,082 \text{ reads} \times 9,090 \text{ bps per read}) / 3,000,000,000$ (as genome size) = 53.54x.

Dataset	Sequencing center	Usage	HCC1395		HCC1395BL	
			# Reads	Total base pairs (Depths)	# Reads	Total base pairs (Depths)
PacBio Sequel [ref 5-6]	CSHL	Primary Contig assembly, phasing, variant calling	17,265,341	140,649,790,223 (46.88x)	17,671,082	160,626,774,754 (53.54x)

We have revised the manuscript by adding the average lengths in the **Methods** section (“**Whole genome sequencing datasets**”).

2. The assembly of the 95 genome was >30% more fragmented, with the largest contigs losing >50% of their length. The authors say it is "mainly due to high level of chromosomal aneuploidy and structural variations in this cancer cell line", w/o providing any data. Confirmatory analyses based on SVs need to be presented here, to corroborate these statements and illustrate the real reasons.

[Response]

We thank the reviewer for this comment regarding fragmentation in tumor cell line assembly. Our assertion is based on the Extended Data Fig. 5: Karyotyping of HCC1395 and HCC1395BL (<https://www.nature.com/articles/s41587-021-00993-6/figures/9>) from the now named ref5 [Fang, L.T., et al., *Establishing community reference samples, data and call sets for benchmarking cancer mutation detection using whole-genome sequencing*. *Nat Biotechnol*, 2021. **39**(9): p. 1151-1160.]. The cytogenetic analysis in this paper showed that **the tumor HCC1395 cell line was hypertetraploid with chromosome counts ranging from 64-79 and a gain of 38-63 unidentifiable marker chromosomes**. Thus, it was **NOT surprising** that assembly of HCC1395 was highly fragmented.

- (a) Karyotype of HCC1395. Cytogenetic analysis was performed on ten G-Banded metaphase cells from HCC1395. Analysis pointed to a hypertetraploid line with chromosome counts ranging from 64-79 and gain of 38-63 unidentifiable marker chromosomes.
- (b) Karyotype of HCC1395BL. Cytogenetic analysis was performed on ten G-banded metaphase cells from HCC1395BL. All ten cells showed loss of a chrX and an unbalanced whole arm translocation between the long-arm of chr6 at band q10 and the short-arm of chr16 at band p10. This resulted in a net loss of one copy of the short-arm of chr6 and loss of one copy of the long-arm of chr16. The abnormal chromosome could be placed in either a chr6 or chr16 locus as we were unable to determine if the centromere belongs to chr6 or chr16 (inset figure).

To properly acknowledge the prior cytogenetic analysis of HCC1395, we have revised the relevant text as below:

“The resulting HCC1395 assembly was more fragmented than the HCC1395BL assembly (**Supplementary Figure 1, Supplementary Figure 2, and Supplementary Table 2**), as demonstrated by the cytogenetic analysis of HCC1395 and HCC1395BL cell lines [6].”

3. The data generation is mostly focused on the BL, not cancer. While it is a bit strange in itself (complex rearrangements would require more, not less data), it's still important to determine if personal genome SVs (relative to GRCH38) are also found in the tumor sample. What is the exact effect of "less data"? How would that affect a general strategy of variant calling using personal genome for future studies?

[Response]

We appreciate the opportunity to address this question. In this study, we added Hi-C data for scaffolding, and nanopore data for phasing for HCC1395BL normal cell line (**Table 1, supplementary Table 3**).

For the HCC1395 tumor cell line, we performed contig assembly using canu for PacBio data and Supernova for 10X linked read data (**Supplementary Table 2**), but **we did not perform scaffolding or phasing analysis**, because we felt that such an analysis for a tumor cell line would be substantially more challenging than that for normal cell line.

With regards to the role of SV analysis in this manuscript, we sought primarily to address the question of whether using a personal genome reference would improve SV calling better by SV callers. Thus, in the **Results** section of the revised manuscript, we have reported results obtained by performing more in-depth SV analyses with short-reads, long-reads, and contig assembly-mapping using both the personal genome and GRCh38 as reference (“**Detection of somatic SVs using short reads was improved using a personalized genome as reference**”, “**Detection of somatic SVs using long reads was improved using a personalized genome reference**”, and “**SV discovery using an assembly-to-assembly mapping approach was improved using a personalized genome reference**”).

We did not use the nanopore dataset for SV calling and comparison, since the coverage of nanopore data was low (Zhou A. et al Evaluating nanopore sequencing data processing pipelines for structural variation identification. **Genome Biol.** 2019; 20: 237. <https://www.ncbi.nlm.nih.gov/pmc/articles/PMC6857234/>).

4. The structural variant analysis is totally inadequate. Fig 5A and 5C differ by 10-fold? A reader concludes that either Sniffles makes a lot of false positive SV calls or pbsv misses a lot of SVs. Both scenarios are bad but I don't see any evaluation of this result or comment.

[Response]

We thank the reviewer for pointing out the differences between the two SV callers used. We acknowledge that the difference is striking, but our study was not designed to robustly compare the results of different callers to one another, but to compare how different the SV callsets generated on GRCh38 vs. personal genome references were to each other, when run using the same inputs, running parameters and post-filtering for a variety of callers. For example, the data in **Fig 5A** in the original submission compares results obtained from two different reference using PBSV results, while **Fig 5C** illustrates the differences between two references when using Sniffles. Since we have documented all the processes regarding the generations of the callsets in the **Methods** section, it should not lead readers to make any conclusion with respect to “which caller is better”, despite there being big differences, typically caused by the application of post-filtering criteria.

The differences in SV numbers between Sniffles and PBSV reflect differences in the callers themselves and the inclusion of the SV calls in the final SV callset for analysis. Despite Sniffles including automatic filter in its output, it still reports a large number of SV calls labeled with “IMPRECISE”, particularly for INS (Insertions). The Sniffles developers suggested that “the easiest is to ignore the IMPRECISE marked calls or you use the allele frequencies” [<https://github.com/fritzsedlazeck/Sniffles/issues/40> “How to filter the variants called by sniffles”, and <https://github.com/fritzsedlazeck/Sniffles/issues/255> “how to interpret the INFO marked with IMPRECISE”]. In the analyses reported in our original submission, we applied AF=0.9 filter to the Sniffles outputs, but this filtering still retained many calls labeled as “IMPRECISE”.

In this resubmission, we have **re-analyzed** the PacBio data using **more conservative filtering** (e.g., excluding all SVs with “IMPRECISE” status for both Sniffles and PBSV) to minimize the calling differences between Sniffles and PBSV, and to emphasize the question of whether using a personal genome reference would improve SV calling, and no longer use INS/DEL balance as an indicator of the

improvement. By using the new filtering, the difference in SV numbers between Sniffles and PBSV was much smaller (**Supplementary Table 11**).

5. Given the issues in p4 one would question the use of long reads only. With the enormous number of available short-read data, the authors should run SV analyses at least on par with the level of their effort in SNV/indel calling. A representative set of short-read SV callers needs to be run on these datasets and compared. Fuentes et al (DOI: gr.241240.118v1) would be a reasonable example to follow, it applied several SV callers including Pindel, DELLY, GROM, and Lumpy. A similar analysis of SVs is needed here.

[Response]

We appreciate the reviewer's comments. In our resubmission, we have selected 4 short-read somatic SV callers, including Pindel (split-read method), BreakDancer (read-pair method), Delly (split-read + read-pair method), and novoBreak (local assembly method) to evaluate how their performances were affected by underlying genome references (GRCh38 vs. personal genome reference). Findings are reported in **Results** section ("**Detection of somatic SVs using short reads was improved using a personalized genome as reference**").

6. The results that will be generated in p5 need to be compared with the SVs identified with long reads. Both p5 and p6 are critical analyses for the use of personal genome, and thus for the whole paper. Many points listed in Q2 below will also need to be addressed with SVs.

[Response]

As noted in our response above, the main goal of this study/manuscript is to compare the impacts of references (GRCh38 vs. a personal genome reference) on SV calling, not to integrate and compare all short-read SVs vs. long-read SVs called by various SV callers on different SV types. Such integration and comparison would need a totally different study design and strategy.

7. Would HLA change be detectable with short reads?

[Response]

We are sorry, but we do not understand this question, and thus how it pertains to the conclusions of our analyses.

Q2. It's still unclear to me how much advantage does personal genome bring vs. the massive data generation involved.

[Response]

We have addressed this question in our response to Q2.8 below.

1. Fig 4A - I am curious why BL with BL as a reference is not really better than with 95 and not MUCH better than with GRCh38? Sup Tab 5 - fewer variants for 95?

[Response]

For the same samples (tumor or normal), our analysis showed that short-reads mapped better when the personal genome was used as a reference as opposed to GRCh38, in terms of there being fewer improperly paired reads, fewer mismatches, fewer split-reads, fewer soft/hard-clipped reads, and fewer standard deviations of the read coverage and library insert-sizes. These improvements were significant as demonstrated in all 12 tumor-normal paired short-read replicates in the resubmission (**Figure 4**). In **Figure 4**, we also showed that the mapping improvement for HCC1395BL based on the personal genome was also much better than that for HCC1395. Our previous **Supplementary Table 5** was a summary of read mappings for the paired pool samples (FDN123 and FDT123), based on use of the personal genome reference as compared to GRCh38, and not related to the variants.

2. Analysis of mitogenome - were these variants found with GRCh38 reference? Eyeballing with IGV, done by the authors, provides perhaps the LOWEST degree of validation, after the calls were made with rather sophisticated SNV tools. If the authors claim novelty of these variants, experimental conformation is needed, which could be trivial with these cell lines. And also - what are sift/vep results?

[Response]

For mitochondrial genome assemblies, a previous report (Mai, Z., et al., *Misassembly of long reads undermines de novo-assembled ethnicity-specific genomes: validation in a Chinese Han population*. **Hum Genet**, 2019. **138**(7): p. 757-769) showed that mitochondrial assemblies may still be misassembled or fragmented (even using long reads), even though they are relatively small. As such, we felt it important to confirm that we had correctly assembled the mitochondrial genomes for the tumor and normal cell lines, and in doing so, subsequently identified 3 somatic variants (**Supplementary Figure 10**). However, we did not attempt extensive variant analyses of the mitochondrial genomes. Variants were annotated with the MITOMAP human mitochondrial genome database (<http://www.mitomap.org>) as described in **Methods**, but we did not use SIFT/VEP to determine the effect of those mitochondrial variants.

When GRCh38 was used as reference for read mapping, Strelka2 did not find all 3 variants. Rather, it reported only C4938A and G14249A as PASS, but T4813C was labeled as LowEVS (Somatic Empirical Variant Score was below threshold). MuTect2 reported 8 somatic SNVs, only 3 of which (T4813C, C4938A and G14249A) had an AF greater than 50%; the 5 other SNV sites had AF less than 10%.

IGV snapshots are commonly used in the field of genomics to verify computational results. However, in our manuscript, we have shortened the corresponding text and move it to the end of **Results** section “**Overall study design and construction of a reference-grade personal genome assembly**” as part of response to another reviewer.

3. Sup Tab 6 - why more variants are detected when genomes are closer? This needs to be explained and discussed. What share of detected events comes from alt-loci, what share reverts to GRCh38?

[Response]

The counts of somatic mutations are largely affected by the mutation types (SNVs vs. SVs), mutation callers, input read types (short reads vs long reads), read coverages, and filtering criteria. For example, the difference in raw MuTect2 calls between GRCh38 and the personal genome reference was much smaller (approximately 3000 SNVs or so) than that for raw Strelka2 calls (approximately 11k SNVs). This was because many low-quality calls such as “artifact_in_normal” and “clustered_events” included in MuTect2 somatic SNV calls were further filtered out by using “gatk FilterMutectCalls”, but we did not have a similar program to filter low quality Strelka2 calls. Thus, the Strelka2 SNV callset contained a fair number of low-quality calls, and these raw SNV calls in **Supplementary Table 5 (Supplementary Table 6** in our previous submission) should not be interpreted as the numbers of true variants. In our study, we intersected the two callsets so that we could obtain high-confident callsets for downstream analysis. Still, on average, 1,689 more overlapping somatic SNVs (**Figure 5A**) and 415 more overlapping somatic indels were seen when using HCC1395BL_v1.0 instead of GRCh38 (**Figure 5B**), because of various improvements in short read mapping (better read placements) using this reference as we have already demonstrated. We have added comments relevant to this point in the **Results** section.

However, the picture is completely different when it comes to the somatic SVs. With Pindel, we observed a clear trend towards reduction when using the personal genome as reference for all SV types detected from 3 paired replicates from a single sequencing center (**Figure 6A**), due to the reduction of split-reads in HCC1395BL_v1.0 based alignments relative to GRCh38-based alignments (**Figure 4D**). For BreakDancer, we observed a decrease in deletion calls, but insertion, inversion, ITX and CTX calls were all increased for each of the normal and tumor samples (**Figure 6B**), when HCC1395BL_v1.0 was used as reference. The observed increase in properly-paired reads (**Figure 4A**), as well as the concurrent decrease in non-properly-paired reads (**Figure 4B**), and the smaller standard deviations in library insert sizes (**Supplementary Figure 4C**) would all potentially affect the predicted deletions or insertions when the personal genome is used as reference. For long-read SVs produced by Sniffles and PBSV, the SV counts in all SV types, particularly for insertions, were all decreased in the tumor sample (HCC1395) based on HCC1395BL_v1.0 vs. GRCh38 as reference (**Figure 7A**).

When GRCh38 was used as reference, sequences from alt-loci were not included in our analysis as described in **Methods**, thus no variants were generated for these alt-loci. However, in our study, we did map those GRCh38 based somatic SNVs (41,669 supported by both MuTect2 and Strelka2) onto HCC1395BL_v1.0 personal reference (**Figure 5C**). We found that 40,768 (97.83%) of 41,669 SNVs were identical (36,773 SNVs, 88.25%) or equivalent (3,995 SNVs, 9.59%), and 682 SNVs (1.64%) were able to map onto HCC1395BL_v1.0 but not overlapped with any Strelka2/MuTect2 calls. 219 SNVs (0.53%) were considered as “not-mapped” onto HCC1395BL_v1.0.

We also provided additional discussions related to the improvements of somatic SNV/SV detection with personal genome reference in **Discussions** section.

4. The text on alt-loci is relevant but GSTT1 and KIRDL5A - where are these located?

[Response]

Both GSTT1 and KIRDL5A genes are found on alt-loci scaffolds, and are not represented in the haplotypes represented on the chromosomes of the GRCh38 primary assembly.

GSTT1 was localized to Chromosome 22 of the GRCh37 primary assembly; but is found only on the alternate locus scaffold (**NT_187633.1**) in GRCh38, due to a change in the underlying sequence at this location when the assembly updated.

KIR2DL5A was present in the haplotype represented on the GRCh37 Chromosome 19 un-localized genomic scaffold (NT_113949.1); but in the GRCh38 assembly, it is found only on alt_loci and novel patches (i.e. NT_113949.2).

As we stated in our manuscript, changes in gene representations from GRCh37 to GRCh38 (or the future versions of the reference) present analysis challenges when switching between different versions of standard reference genome. In addition, because these genes are represented only on alt_loci/patches in GRCh38, and most existing tool chains do not handle those alternate locus scaffolds, they are consequently more difficult to study. Their typical exclusion from analysis thus presents a heightened risk for misinterpretation of results. In contrast, as the individual specific haplotypes for these clinically relevant genes were correctly represented in the personalized assembly, no special handling of alt_loci/patches would be needed to assess them in the tumor genome if HCC1395BL_v1.0 were to be used as reference as opposed to GRCh38.

5. The authors bring up "balanced" numbers as an indication of improvement. Is that really the case? It needs to be discussed and argued for.

[Response]

This question is related to Q1.4. Again, in this revision, to minimize the calling differences between Sniffles and PBSV, we have re-analyzed the PacBio data using more conservative filtering (e.g., excluding all SVs with "IMPRECISE" status for both Sniffles and PBSV, and many related to INS) to emphasize the question of whether using a personal genome as reference would make the SV calling better, not using INS/DEL balance as an indicator for improvement.

However, previous studies showed that GRCh38 reference has a **genome-wide deletion bias**, suggesting the systematic collapse of repeats during its initial cloning and assembly [M. J. P. Chaisson et al, 2015, Resolving the complexity of the human genome using single-molecule sequencing. *Nature*. 517, 608–611]. Such bias would have impacts on germline SV detections with tendency to **call more insertions** [Miga, K.H., et al., 2020, *Telomere-to-telomere assembly of a complete human X chromosome*. *Nature*. 585, 79-84. Extended Data Fig. 3. Zook, J.M. et al. 2020. A robust benchmark for detection of germline large deletions and insertions. *Nat Biotechnol* 38, 1347–1355. Fig. 3]. Ultimately, this would affect somatic SV calling as well.

For somatic SVs, even when we no longer used INS/DEL balance as indicator of improvement for SV calling, we still could observe many more INSERTIONS being called by paf tools and Assemblytics on GRCh38 as opposed to the personal reference, as shown in **Supplementary Figure 9**. Also, with more stringent filtering criteria being used, we continued observing many more INS calls by both Sniffles and PBSV with PacBio long reads when GRCh38 was used as reference as opposed to the personal reference (**Supplementary Table 11**).

We also provided additional discussions related to genome-wide deletion bias in GRCh38 in **Discussions** section.

6. I am confused about the use of Dovetail and ONT data. The integration of these datasets needs to be detailed. These are the issues, currently:

Dovetail - usefulness not clear. Using it leads to increased numbers in the right half of Table 1.

Interpretation? Specific advantages?

Oxford - usefulness not unclear. Is said to be used for phasing but "phasing" section does not even mention it. After the data is presented, one needs to see the same: Interpretation? Specific advantages?

[Response]

Dovetail Hi-C data was used for evaluating scaffolding strategies for generating HCC1395BL_v1.0. Combining Hi-C data and 10X Genomics' linked reads produced a better scaffolded assembly, as demonstrated in **Table 1**.

ONT data (~15X) was used for phasing HCC1395BL_v1.0, as in the phasing results section, as well as in the **Methods** section. With the combination of PacBio and ONT data, the phasing result using WhatsHap, as described in Results section: "Approximately 15-fold coverage of Nanopore long reads was used to extend phasing further, to 2.54 Gb. The total number of phased blocks was subsequently decreased to 3,204 from 6,368 blocks, and the longest phased block was greatly improved, increasing from 6.37 Mb to 20.45 Mb (**Supplementary Table 3**)".

We also clearly stated the usages of these two data in the **Supplementary Table 1**.

7. The final paragraph in Discussion seems too focused on things like centromeres and telomeres, which were not relevant for tumor-normal findings here. Or were they relevant? - this is not explained.

[Response]

We thank the reviewer for this comment. Sequencing technology continues to evolve as time goes by. Using a combination of advanced assembly algorithms and latest long read sequencing technologies (PacBio HiFi, ONT etc.), we believe that it is possible to further improve our *de novo* assembly, possibly covering each of the whole chromosomes from telomere to telomere without gaps, and thus enable us to perform an even more comprehensive analysis to identify somatic mutations. Please note that approximately 8% of the human genome in GRCh38.p13 remains unfinished, including pericentromeric/subtelomeric regions, recent segmental duplications, ampliconic gene arrays, and rDNA arrays. The p-arms of five acrocentric chromosomes (Chr13, Chr14, Chr15, Chr21, Chr22) lack any ordered or oriented sequence. Thus, our discussion briefly touches on centromeres/telomeres/gaps etc. as an inspirational direction for improvement of the *de novo* assembly and future studies to explore somatic mutations in these unknown regions.

8. Overall, what is the cost-benefit ratio of the approach involving personalized genome? How many relevant variants (unavailable with GRCh38) have been identified? Any clinical advantage of using personalized genome for this pair of samples? So far, we only see "SNVs were located in exonic or intronic regions, therefore including or excluding these sites would have an impact on downstream mutation interpretations". Very specific examples and numbers should illustrate this vague statement. Otherwise, the advantages are inconclusive.

[Response]

We thank the reviewer for this comment. Based on our analysis in this study, the benefits of using personal genome reference as opposed to GRCh38 included multiple aspects:

- (1) The personal assembly encompasses the individual specific haplotypes, and has better representations for the clinically important genes; and in particular, there is no need to deal with ALT loci in NGS secondary analysis.
- (2) Short-read mapping has been improved in terms of smaller standard deviations of both read coverages and library-insert sizes, fewer improperly paired reads, split-reads, soft-/hard-clipped reads, so on and so forth. Short reads were more uniformly placed on personal genome reference.
- (3) Long-read mapping has been improved with higher numbers of reads being mapped, fewer mismatches, and smaller standard deviations of read coverages. Long reads were more uniformly mapped with personal genome reference.
- (4) Most importantly, our analysis revealed that use of a personal genome reference improved somatic SNV and SV detection using short reads, particularly in reducing false positives. Some of the somatic SNVs were only discovered using a personal genome as reference and validated, and some of them were related to biological pathways. Somatic SNV callsets were found more consistent between different SNV callers.
- (5) Use of a personal genome reference also improved long read SV analysis and assembly-based SV analysis, particularly in reducing the false positives. Somatic SV calls were found more consistent between different SV callers.

More specifically for somatic SNVs, as stated in **Results** section “**Personalized genome reference allowed more accurate detection of somatic SNV mutations using short reads**”, when we mapped GRCh38 based somatic SNVs (41,669 by both MuTect2 and Strelka2) onto HCC1395BL_v1.0 personal reference, we found that 40,768 (97.83%) of 41,669 SNVs were either **identical (36,773 SNVs, 88.25%)** or **equivalent (3,995 SNVs, 9.59%)**, and **682 SNVs (1.64%)** were able to map onto the HCC1395BL_v1.0 but without overlapping Strelka2/MuTect2 calls. **219 SNVs (0.53%)** were considered as “not-mapped” onto HCC1395BL_v1.0 (**Figure 5C**). We used KEGG pathway enrichment analysis (**Figure 5D**) to demonstrate the important pathways genes carrying those SNVs identified using personal genome as reference (and without equivalent GRCh38-based SNVs) are involved in. For example, GTF2H2 (General transcription factor IIH subunit 2) encodes the subunit of RNA polymerase II transcription initiation factor IIH, which is involved in both basal transcription and nucleotide excision repair (**Supplementary Table 7**). PTPN13 (Protein tyrosine phosphatase non-receptor type 13) encodes a signaling molecule that belongs to the protein tyrosine phosphatase (PTP) family, which regulates a variety of cellular processes such as cell growth, differentiation, mitotic cycle, and oncogenic transformation (**Supplementary Table 7**).

Finally, we wish to thank the reviewer for their many excellent comments and questions that have helped us improve our manuscript. In summary, we have revised our manuscript as highlighted below:

- (1) Expanded short-read alignment analysis to cover all 12 paired tumor/normal WGS replicates, including improperly paired reads, mismatches, standard deviations of library insert sizes, read coverage standard deviations, split-reads statistics, soft-clipping with/without an SA-tags, so on and so forth. Split-reads and improperly paired reads were related to SV calling.

- (2) Expanded somatic SNVs analysis to cover all 12 paired tumor-normal WGS replicates.
- (3) Performed PCR validation using Sanger sequencing for a subset of somatic SNVs that were only discovered using a personal genome as reference.
- (4) Added somatic SV analysis with short reads using 4 SV callers, including Pindel, BreakDancer, Delly, and novoBreak.
- (5) Added long-read alignment statistics and comparison for both tumor and normal samples.
- (6) Re-analyzed PacBio long-read data for somatic SVs using Sniffles and PBSV with more stringent criteria, emphasizing the question of whether use of a personal genome reference would improve SV calling.
- (7) Re-analyzed assembly-to-assembly based somatic SVs using Assemblytics and Paftools by comparing the tumor HCC1395 contig assembly to normal HCC1395BL_v1.0, emphasizing the question of whether use of a personal genome reference would improve SV calling, and no longer using INS/DEL balance as an indicator of SV call improvement.
- (8) Enhanced the description for assembly of HCC1395 tumor cell line in **Results** section
- (9) Added more detailed descriptions in the **Methods** section for all the steps involved in assembly, evaluation, read mapping, and variant calling.
- (10) Revised the **Abstract**, **Discussions** section accordingly in response to reviewers' suggestions.

With all the results we observed and have presented in the manuscript, we are confident that use of a personal genome as reference for somatic mutation detection for tumor-normal paired samples prevails over the use of GRCh38 as a reference. We hope all these further analyses/improvements have satisfied **reviewer #3**.

Second round of review

Reviewer 1

Firstly, an apology for referring to NovoBreak when I mean NovoAlign, my first round comments on aligner that handle SNV heterozygosity should make more sense in light of that.

Overall, the manuscript is much improved and, whilst you did (understandably) reject many of my suggestions, the new manuscript has a much clearer focus on somatic calling. Unfortunately, the SV analysis still needs more work.

Firstly, your choice of callers is problematic. Pindel and BreakDancer are massive outliers in terms of their somatic SV performance (doi.org/10.1093/bib/bbaa056 Fig1). BreakDancer is particularly bad as its sensitivity is as absolutely horrible on modern data sets where fragment size is not $\gg 2 * \text{read length}$ (<https://www.nature.com/articles/s41467-019-11146-4>). The reason everyone still benchmarks their SV caller against BreakDancer is that it'll make their caller look good no matter how bad it is because BreakDancer is only suitable for data sets with $2*50\text{bp}$ or shorter reads. Nobody should care how well BreakDancer performs at somatic SV calling because it is an inappropriate tool for the job and including it in any recent comparison of tumour data is actively dangerous and could literally get patients killed if used for actual patient analysis. This isn't an exaggeration. If frag size $> 2*\text{read length}$ then it's just a moderately bad caller but as soon as $\text{FS} < 2*\text{read length}$ then the results are complete garbage. It is entirely plausible that a clinical pipeline that uses BreakDancer does not hard QC-fail a sample with shorter than normal

fragment size (e.g. more degraded FFPE sample than usual) and as a result fails to detect a driver fusion and thus causing the oncologist to make a sub-optimal drug choice. Any mention of BreakDancer (other than to say it is unsuitable for somatic patient analysis) is irresponsible.

In your reviewer response, you mentioned that using a germline SNV caller for somatic SV detection is inappropriate. The same holds true for SV calling. It is unfortunate that you've only included one of the 6 somatic callers that have been used in large-scale cancer cohort SV analysis (PCAWG: SvABA, DELLY, BRASS, dRanger; Hartwig: manta, GRIDSS2(+GRIPSS)). You really need to include additional ones. I recommend adding manta (SR+RP+breakpoint assembly) and GRIDSS2(+GRIPSS for doing the somatic filtering)(SR+RP+breakend assembly) as they both explicitly support somatic calling and use different approaches to Delly (RP then SR) and NovoBreak (de novo somatic kmer identification). Benchmark-wise, Delly is an acceptable choice but, at least in the GRIDSS2 benchmark, novoBreak sensitivity is low (<https://genomebiology.biomedcentral.com/articles/10.1186/s13059-021-02423-x>) so you've really only got 1 somatic SV caller that I'd trust the results of in your comparison.

Overall, the SV analysis is lacking the following:

1) An evaluation of equivalence of calls between GRCh38 and HCC1395BL_v1.0

You've used the flanking sequence to define equivalence of SNV/indels between hg38 and BLv1.0 but you didn't do this for breakpoints. This information was critical to determining what the actually SNV/indel calling difference are between the references and is equally so for SVs. You're filtering the IMPRECISE calls (and shouldn't be using BreakDancer) so the same approach will work for breakpoints: two calls can be deemed equivalent if the breakpoint flanking sequences are sufficiently similar.

Once you have call matching code, extending this to look at how the short/long read callers are impacted and whether they are impacted in the same way.

Helpful tip: you can use the Bioconductor StructuralVariantAnnotation package to standardise calls from all callers into a consistent (breakpoint-centric) notation. It'll even handle the INS vs DUP issue you get matching long and short read calls and can report transitive calls. This brings me to my next concern

2) cis-phased adjacent SV handling

In your reply you dismissed this as hard/not relevant but I think it's two important to be dismissed so lightly. 22% of somatic SVs are within 1kbp of another somatic SV and this rises to 31% by 10kb and even on a relatively stable genome such as COLO829, this caused 5 of the 67 SV in truth set to be called differently between the short and long read callers (<https://genomebiology.biomedcentral.com/articles/10.1186/s13059-021-02423-x>). This is too big of an effect size to ignore completely. The simplest way to do this would be to consider cis/trans phasing in flanking sequence matching logic and match if any of the potential flanking sequences match.

3) Correct handling of event type.

- It's unclear if INV call are actual inversions or just ++ and -- oriented breakpoints. DELLY uses the CT INFO field (e.g. CT=3to3) to specify these breakpoints. The vast majority of your "inversions" are not actually inversions.

Furthermore, I expected to see more interchromosomal calls in the personalised genome as the scaffolding is less complete.

Proper classification of somatic breakpoints is hard (e.g. Figure 2 of

<https://www.biorxiv.org/content/10.1101/2020.12.03.410860v1>) so since it's too much work to do here, either a) just turn everything into breakpoints and compare breakpoints, or b) special case simple DEL/DUP events (although you're not going to be able to differentiate between them and DEL/DUP-like breakpoints that are part of complex rearrangements) if you find there's actually a difference there, or c) use a DEL/DUP/INV/BND classification and explicitly state that INV is any +/+ or -/- oriented intra-chromosomal breakpoint.

4) FP/FN estimation

You've made not attempt to determine the FNR/FDR of the calls from any of your callers. There's not even an analysis of the consistency across callers. If you don't like the idea of consistency across callers as your proxy for truth then you should consider using copy number consistency as a proxy. The GRIDSS2 authors used this to compare the caller on the PCAWG and Hartwig cohort and, although you've only got a single sample, it would be useful to at least estimate the FDR/FNR.

5) Analysis of the actual reference-induced differences

This is closely related to point 1. You haven't demonstrated why your reference is actually better for SV calling. Unlike germline SV, somatic SVs aren't enriched for low complexity sequence (except for LINE reactivation and MSI) so given you only have one sample, I don't expect a statistically significant difference between the reference genomes. There should only be a handful of somatic SVs that don't have matching calls in both references. This is clearly not the case from your results. Like you did with SNVs, you need to show that the few % of SVs that are different are different because of reference differences.

6) [Minor] Consistency of results

You have lots of replicates. Why not repeat the SV analysis across replicates to see how consistent the call are?

Specific points:

> We hypothesized that the numbers of the predicted SVs by Pindel would decrease due to the reduction of split-reads observed in HCC1395BL_v1.0- based alignments as compared to GRCh38 based alignments (Figure 4D)

Reduction in split-reads comes from two things: 1) fewer real SVs, 2) reduction of chimeric fragments generated during sequenced. The latter does not depend on the reference so the expected reduction must come from the former. For germline you would expect fewer calls but, as Figure 4D indicates, we're talking about somatic SVs and we expect either the same, or more (if the personalised reference is better) SVs. By what mechanism are you hypothesizing a reduction in the number of events? The only way I could explain this is if you expect most SVs in your call set to be false positives.

> undoubtedly demonstrated the benefits of using the personal genome HCC1395BLv1.0 in reducing potential false positives

You've provided no evidence that the reduction is caused by having fewer false positive and not more false negative. I interpret this result as the personalised reference being worse for Pindel somatic SV calling than hg38.

> BreakDancer

Please remove your BreakDancer results. BreakDancer's performance cliff is dangerous and your paper doesn't specify the real length and fragment size so I don't even know what side of the performance cliff your results are on.

> BreakDancer clearly identified 35.56% (on average) fewer supporting reads for normal sample,

and 32.34% (on average) fewer for tumor sample with the personal genome reference as compared with GRCh38 (Supplementary Table 9), which could be explained by the decrease of non-properly-paired reads (Figure 4B).

Unfortunately, it cannot. A personalised reference is supposed to reduce the discordant pair rate by including the SV in the reference. Real somatic SVs that remain should have exactly the same support. Supporting read counts for real somatic SVs that much lower indicates that the personalised reference is much worse for BreakDancer SV calling.

> Additionally, Delly reported 3 insertions on HCC1395BL_v1.0, but none on GRCh38, while novoBreak did not report any insertion event (Supplementary Figure 7A).

So what are those three calls? Are they microsatellites? Why are the present only on BLv1.0? These differences are the interesting one because they look like they could be genuine reference differences.

> Noticeably, the percentages of overlapping DELs (as well as DELs 50% or more by length) detected by both Delly and novoBreak on HCC1395BL_v1.0 were slightly higher (5% for Delly, 5.9% for novoBreak) than on GRCh38 (Figure 6D).

Overlapping DELs are almost certainly chromothripsis or BFB breakpoints and not actually overlapping DELs. Whether they overlap by 50% or more is not in actually meaningful. More overlapping DELs either means more FPs or more TPs. You need to show that these extra chromothripsis-like DELs (or are they fragile site/rigma?) are real (or at the very least, there is a change in CN at the breakpoint) and not just that the personalised reference results has a higher FDR or is misassembled thus causing more simple DELs to appear to be overlapping.

> Fig6D

Why do all the other this axes start at 0 and this one doesn't?

> Detection of somatic SVs using long reads was improved using a personalized genome reference

What's the concordance between PBSV and Sniffles?

> Most notably, somatic insertions were decreased significantly by 32.8% for Sniffles and 27.3% for PBSV

Doesn't this mean worse (LINE/microsatellite) sensitivity on BL_v1.0?

> overlapping somatic deletions between Sniffles and PBSV

Again, it's unclear what, if anything, this means.

If you have breakpoint classifications (e.g. from LINX -

<https://www.biorxiv.org/content/10.1101/2020.12.03.410860v1.full.pdf>) then you could say something about how efficient (or not) the recovery of complex events was but without it or a ground truth these result don't actually answer the question about which reference is better.

> SV discovery using an assembly-to-assembly mapping approach was improved using a personalized genome reference

The number of SVs reported in this section is so much high than the short read and long read results that I don't believe they're correct. Your assembly-to-assembly comparison has way more SVs than any of the samples in either PCAWG or Hartwig that I don't see how they can be correct. You need to sanity check your contig mapping and show that putative breakpoints are supported by your short and long read data. To me this result just says either a) your somatic assembly is wrong, or b) your somatic assembly has duplicate contigs, and/or c) your somatic assembly has split up amplified SVs into separate contigs thus resulted in inflated SV counts.

You can use the same breakpoint flanking matching logic to determine which of the 3 options is true or if you really do have an excessively rearranged sample.

Note that a perfect assembly of a breakage-fusion-bridge event will linearise the amplified region and an assembly-to-assembly comparison of such an assembly will have a much higher SV count than would be called by the equivalent perfect reference-based read alignment (e.g. a CN=20 breakpoint would be called as 1 reference-based breakpoint, and 20 assembly-based breakpoints). High counts are not necessary wrong, you just need to explain them.

In conclusion, the results presented does not support your conclusion that the personalised reference improves SV calling. More/different analysis is required.

>We included about 175-fold of Illumina short reads that were pooled from 3 replicates (FDN1, FDN2, and FDN3)

You pooled 3 different libraries together? You can't do that unless they have the same fragment size distribution (which they won't). Since fragment size is so important for SV calling, callers typically require you to specify them as separate BAM files. If you're lucky your caller will support read groups but very few do.

Please include the median fragment size for each library.

>Pindel Somatic SVs were filtered on the VCF file with no supporting reads in the matched normal sample [30].

>BreakDancer somatic SVs were filtered with no supporting reads in the matched normal sample [30].

That won't filter germline SVs that have don't have any germline coverage at all.

I recommend using actual somatic SV callers, as it avoids annoying review comments complaining about how your custom filtering is not representative of actual usage.

[Minor issues]

> Since the bit (flag 0x2) for properly paired reads was set by the aligner (BWA-MEM), there was no reason to believe that BWA-MEM would not set the "proper paired" bit for a split-read alignments if the orientation and insert size were satisfied, based on the post (<https://www.biostars.org/p/79611/#81436>)

It's a minor point but you should be aware that bwa sets the proper pair flag based on the primary alignment. If it's the supplementary alignment that's concordantly aligned, then bwa won't set the flag. Here is an example from simulated data:

```
variant.chr12-art4601880 161 chr12 1527056 60 100M = 1528347 1347
ACTAAGTGCCAAACACTAAGTTGTAAGTTTATAATCACATTGTAAGGATGTTTTTCAT
TGAAACCAATATAACAATCCTAAGAGATAGATACCATTTTTTCAT
5??A=BBBBDDD@D-DGCGGEGH/FAFFIFI/@FIFHIA+GHHHFH-
@HH*5GF=IHGEHIEH+HHFEFGEHAHFGEFH<DHAH/EHHHIBHDHHHE/=H M
C:Z:44S56M MD:Z:46C53 RG:Z:1 NM:i:1 MQ:i:60 AS:i:95 XS:i:21
variant.chr12-art4601880 2129 chr12 1527279 60 49M51H = 1527056 -272
TACCTCTGAAGGATACATAGCTAGAAGGATTTTTTAAAGACGTTATTAT
```

HHHHHHHHI7BHFFFHBFHIIHIAHIIIIGIHDEHDIHHHIFHDHHH SA:Z:chr12,1528347,-
,44S56M,60,0; MD:Z:49 RG:Z:1 NM:i:0 AS:i:49 XS:i:21
variant.chr12-art4601880 81 chr12 1528347 60 44S56M = 1527056 -
1347

TACCTCTGAAGGATACATAGCTAGAAAGGATTTTTTAAAGACGTTATTATAAATAAGA
CTAACAATAAACTCCTTTTTAAATCTGAGTAGAAAAGGACCCC
HHHHHHHHI7BHFFFHBFHIIHIAHIIIIGIHDEHDIHHHIFHDHHHIGEIIFAHIHIGFHIIHIGI
HHFIIFIIDF8FGGDDDDDDDD?@<??A5A SA:Z:chr12,1527279,-,49M51S,60,0; MC:Z:100M
MD:Z:56 RG:Z:1 NM:i:0 MQ:i:60 AS:i:56 XS:i:20

The phrasing of your abstract could be improved:

> cancer-normal

The usual phrase is "tumor-normal"

> individual specific haplotypes for complex regions and medically relevant genes

The meaning of "haplotype" is problematic here. It could mean maternal/paternal or it could mean you actually reconstructed all derivative chromosomes.

> but also ameliorated the detection accuracy of somatic SNVs and SVs

"improved" would be better as you haven't established that hg38 results are "bad", merely that the personalised ones are better

> Our findings demonstrated that use of a personalized genome with individual specific haplotypes was essential for accurate detection of the full-spectrum somatic mutations in paired tumor-normal samples

Missing an "of"; you haven't demonstrated that it's essential. AFAIK, the current state-of-the-art in short read somatic calling is the GRIDSS2+GRIPSS+PURPLE+LINX Hartwig pipeline which claims a sub-5% FNR/FDR at 100x coverage

(<https://genomebiology.biomedcentral.com/articles/10.1186/s13059-021-02423-x>).

> The N50s of HCC1395 assemblies were significantly smaller than that of the corresponding HCC1395BL

Why? Are the contigs breaking at somatic SVs and are otherwise intact or is assembler failing because the genome does not have uniform coverage? HCC1395 probably has chromothripsis so either more genes are broken, or the assembly is worse (probably a combination of both).

Figure 2 could be improved by changing the colour of the discordantly aligned contig fragment to black and plotting them over the top of the others. Eg. the small scf2 is mostly chr1 but there's a bit that goes somewhere else and I can tell where it is. If you also highlight where the centromeric repeats are in the light blue scaffolds that'd answer whether that is chr8 alpha satellite repeat or not.

I'm still a bit confused about the HLA typing result. Does HCC1395BL_v1.0 contain just the HLA-DR51 haplotype, or did you assemble two haplotypes?

Figure 4: it would be good to indicate the legend labels that HCC1395BL/HCC1395 refer to the aligned reads from that replicate, not the reference. Even something as simple as having "Reads from" would reduce the cognitive burden of understanding what's being plotted.

Page 18L33: please add a sentence making it clear when you're talking about somatic SVs, and when you're talking about germline.

[out of scope] Why are there more GATK calls? I would have expected fewer since GATK will filter calls that it can't genotype and the presence of aneuploidy massively messes up SNV genotyping.

> Pindel RPL

One could argue that these aren't really SVs at all (and I've seen RPL-like sequencing artifacts) and could be just a bunch of adjacent SNVs/indels.

Reviewer 2

The revised manuscript addresses the concerns in the review and is substantially improved.

Reviewer 3

The authors have reworked the m/s substantially and added some missing analyses. The presentation is definitely improved but the story is still rather lackluster overall. So much effort and money spent on this personal reference - but the real outcome is that SD is a bit lower and read mapping has improved. This had been expected from the start, hadn't it? With the personal reference the readers will expect results that are less abstract and closer to a patient, otherwise they will be disappointed. I think there is a way to make this paper exciting, but it requires more work. Given that the authors rebutted some comments as being outside the scope I should say this IS the scope, what's the point of the personal genome otherwise?

My current concerns (occurring from incomplete analyses or unaddressed) include:

1. Benefits of a personal genome are now presented in greater detail. However, the points highlighting those are currently dispersed all over the text as written and a reader must dig for it. A succinct summary in a tabular form (column 1 - benefit; column 2 - short proof, maybe some numbers) would greatly benefit the paper in such consideration. Authors' answers to my Q2.8, with a bit of granularity would work fine there.
2. One of the major concerns I had - would such personal genome improve SV detection in tumor with short reads? It is addressed to some extent but not completely and the confident tone is over the top. Eg: "Our findings indicated that use of a personal genome as reference had great impacts on SV discovery..." I don't think the effects shown (like in Fig. 6) are particularly GREAT impacts, I'd say there is SOME impact, even in the ever so numerous calls of BreakDancer. As DEL is the easiest SV, the fact that "the numbers of somatic deletion calls were similar" for BreakDancer in both references argues against so called "great impacts". "Small but consistent improvements" would be a more fitting expression. If such personalization would allow for finding variants that could not be found otherwise or clarifying something of the kind - that would be a demonstrable impact. Especially, if we're talking about clinical use (and personal reference makes me imagine a patient) - imagine talking to cancer patients about SD of their mapped reads?
3. Equally troubling is the statement "undoubtedly demonstrated the benefits of using the personal genome HCC1395BLv1.0 in reducing potential false positives" - how could you say this w/o even knowing the false positives? Any examples, maybe illustrated or best - validated?

And, hopefully, not with the IGV. Strictly speaking, manual inspection in IGV (unless mapping is really poor) is not different from reading the contents of VCF files, except a graphical viewer is used instead of a text editor.

4. One possibility to address points 2 and 3 and add flavor to the story could be: take the SV calling results and show how they may incorrectly predict a potentially relevant deletion with a standard reference but that such result could be avoided with HCC1395BLv1.0, thus potentially avoiding a wrong treatment option. An IGV illustration would work fine here, actually.

5. Further, the authors have improved the presentation of the long-read results but refuse to compare them with short reads. Such comparison is important, as it also would address a potential recommendations (or just thoughts) - should one use short or long reads tech in such cases? Or both? And next - what variant calling tools would be advisable? And possibly - what to watch for, what pitfalls to avoid? All this can be expressed rather succinctly and based on the current observations with the variant callers used here.

6. Figures would strain the readers. I had a hard time reading the axis labels of Fig. 6. for example. Fonts are often so puny, they are unreadable, legends aren't very descriptive. Fig. 1/Table 1 legends are particularly laconic, and more details would be great.

7. They also didn't address Q1.7. Let me clarify. The authors underscore the importance of the correct personal assembly in complex regions. They also mention in response to Reviewer 2 the relevance of HLA mutations to cancer. Was anything detected in this region in the tumor sample with short reads? A measure of improvement compared to using the standard reference would be illustrative.

Reviewer reports and point-by-point responses for GBIO-D-21-00720R1

Reviewer #1:

Firstly, an apology for referring to NovoBreak when I mean NovoAlign, my first round comments on aligner that handle SNV heterozygosity should make more sense in light of that.

Overall, the manuscript is much improved and, whilst you did (understandably) reject many of my suggestions, the new manuscript has a much clearer focus on somatic calling. Unfortunately, the SV analysis still needs more work.

Firstly, your choice of callers is problematic. Pindel and BreakDancer are massive outliers in terms of their somatic SV performance (doi.org/10.1093/bib/bba056 Fig1). BreakDancer is particularly bad as its sensitivity is as absolutely horrible on modern data sets where fragment size is not $\gg 2 * \text{read length}$ (<https://www.nature.com/articles/s41467-019-11146-4>). The reason everyone still benchmarks their SV caller against BreakDancer is that it'll make their caller look good no matter how bad it is because BreakDancer is only suitable for data sets with $2*50\text{bp}$ or shorter reads. Nobody should care how well BreakDancer performs at somatic SV calling because it is an inappropriate tool for the job and including it in any recent comparison of tumour data is actively dangerous and could literally get patients killed if used for actual patient analysis. This isn't an exaggeration. If frag size $> 2*\text{read length}$ then it's just a not so great tool but as soon as $\text{FS} < 2*\text{read length}$ then the results are complete garbage. It is entirely plausible that a clinical pipeline that uses BreakDancer does not hard QC-fail a sample with shorter than normal fragment size (e.g. more degraded FFPE sample than usual) and as a result fails to detect a driver fusion and thus causing the oncologist to make a sub-optimal drug choice. Any mention of BreakDancer (other than to say it is unsuitable for somatic patient analysis) is irresponsible.

In your reviewer response, you mentioned that using a germline SNV caller for somatic SV detection is inappropriate. The same holds true for SV calling. It is unfortunate that you've only included one of the 6 somatic callers that have been used in large-scale cancer cohort SV analysis (PCAWG: SvABA, DELLY, BRASS, dRanger; Hartwig: manta, GRIDSS2(+GRIPSS)). You really need to include additional ones. I recommend adding manta (SR+RP+breakpoint assembly) and GRIDSS2(+GRIPSS for doing the somatic filtering)(SR+RP+breakend assembly) as they both explicitly support somatic calling and use different approaches to Delly (RP then SR) and NovoBreak (de novo somatic kmer identification). Benchmark-wise, Delly is an acceptable choice but, at least in the GRIDSS2 benchmark, novoBreak sensitivity is low (<https://genomebiology.biomedcentral.com/articles/10.1186/s13059-021-02423-x>) so you've really only got 1 somatic SV caller that I'd trust the results of in your comparison.

[Response]

We thank the reviewer for their additional feedback. We have added two more modern SOMATIC callers, **GRIDSS2/GRIPSS and Manta** as suggested, to replace 2 germline SV callers (Pindel and BreakDancer) in our latest revision. We included Pindel (split-read method) and BreakDancer (read-pair method) in our previous version of manuscript solely because they were single algorithm callers for SV detection that could be used for assessing how two reference assemblies affect germline SV calling based on the same set of short reads. If underlying read mappings were improved or changed (e.g., reduction of split-reads, and non-properly-paired reads) on the personalized assembly, then the germline SV calls would be impacted accordingly as we've demonstrated before.

In response to the reviewer’s concerns about BreakDancer, we also recognized the limitation of the tool. Please note that previous version of our manuscript noted that BreakDancer is less sensitive to smaller SVs, particularly SVs that are smaller than the library insert size (p. 19: “Read-pair SV calling relies on discordant alignment signatures, such as mapping intervals that are smaller or larger than expected library insert sizes, or abnormal mapping orientations. It is less sensitive to smaller SVs, particularly those that are smaller than the library insert sizes”). This was true in our study. For example, we compared 121 somatic deletions detected by BreakDancer (requirement: no germline SVs in the regions and with 5 or more reads supporting the deletion) with that (275 deletions) by GRIDSS2 (P2P-Table 1). We noticed that more than half of GRIDSS2’s deletions were smaller than 300bps, but only one from BreakDancer was smaller than 300 bps. Moreover, 109 of 121 deletions (90.08%) were matching GRIDSS2’s deletions, and even some of the BreakDancer-unique deletions (not detected by GRIDSS2) were supported by PacBio in-read deletions (P2P-Figure1A/1B).

	# Somatic DELs	# Equivalent DELs	# Uniq DELs	Note
BreakDancer	121 (1 DEL < 300bps; 120 DELs >= 300bps)	109	12	
GRIDSS2	275 (150 DELs < 300bps; 125 DELs >= 300bps)	102	173	150 of 173 DELs < 300bps

P2P-Table 1. Summary of somatic deletions detected by BreakDancer and GRIDSS2.

P2P-Figure 1A. IGV snapshot for Chr8:113921981-113922454:291bps-DEL detected by BreakDancer (but not by GRIDSS2 with a very large 51Mbps DEL chr8:71344802-122827322), supported by PacBio in-read deletions (top 4 alignment tracks: pbmm2-tumor, pbmm2-normal, ngmlr-tumor, ngmlr-normal; bottom 2 alignment tracks: bwa-tumor, bwa-normal).

P2P-Figure 1B. IGV snapshot for Chr13:30022410-30022709:326bps-DEL detected by BreakDancer (but not by GRIDSS2) that is supported by PacBio in-read deletions (top 4 alignment tracks: pbmm2-tumor, pbmm2-normal, ngmlr-tumor, ngmlr-normal; bottom 2 alignment tracks: bwa-tumor, bwa-normal).

Additionally, we would like to point out that reviewer's comments about BreakDancer's performance with respect to library insert sizes and read lengths were **NOT applicable to the case in our study**. Firstly, the sequencing data we used for BreakDancer analysis were from 3 tumor/normal paired replicates from a single sequencing center, with read length of 150bps, and median insert sizes of 377/367 for replicate1, 375/371 for replicate2, and 371/368 for replicate3. The average insert size for 3 tumor/normal replicates is about **371**, which is much greater than $2 * 150 \text{ read length} = 300 \text{ bps}$. In fact, all our 12 tumor/normal paired replicates had median insert sizes greater than $2 * 150\text{bps}$ read length. Thus, this should address the reviewer's concern, as they themselves noted that it is when median insert size is *less than* ($2 * \text{read length}$) that BreakDancer results cannot be trusted. As that is not the case in our analyses, the SVs called by BreakDancer should be still valid. Secondly, we wish to emphasize that we did not use, nor are we recommending, BreakDancer callsets to develop somatic SVs for clinical usage. Rather, we used it in our analysis to assess how calls made by BreakDancer would be impacted by use of a personalized genome reference as compared to use of GRCh38. Lastly, we feel that it would be irresponsible for an oncologist to rely on SV results from any single SV caller without verifying results from other callers. After all, we feel that there is **NO single perfect SV caller** that would detect all SVs based on our analysis and curation experience.

Nonetheless, we have removed BreakDancer results from the manuscript, as the corresponding data from 4 other SOMATIC SV callers is sufficient to demonstrate the benefits of using personalized assembly as reference for somatic SV detection (without confusion of germline SVs with somatic SVs).

Overall, the SV analysis is lacking the following:

- 1) An evaluation of equivalence of calls between GRCh38 and HCC1395BL_v1.0 You've used the flanking sequence to define equivalence of SNV/indels between hg38 and BLv1.0 but you didn't do this for

breakpoints. This information was critical to determining what the actually SNV/indel calling difference are between the references and is equally so for SVs. You're filtering the IMPRECISE calls (and shouldn't be using BreakDancer) so the same approach will work for breakpoints: two calls can be deemed equivalent if the breakpoint flanking sequences are sufficiently similar.

Once you have call matching code, extending this to look at how the short/long read callers are impacted and whether they are impacted in the same way.

Helpful tip: you can use the Bioconductor StructuralVariantAnnotation package to standardise calls from all callers into a consistent (breakpoint-centric) notation. It'll even handle the INS vs DUP issue you get matching long and short read calls and can report transitive calls. This brings me to my next concern

[Response]

Thank you for your suggestions. In the new revision, we indeed mapped the flanking sequences of GRCh38-based SVs to HCC1395BL_v1.0, and generated equivalent SV calls with two references, as well as un-mapped SVs, and HCC1395BL_v1.0 specific SV calls (**Figure 6C/6D, Figure 7B/7C/7D for short-read SVs, and Figure 8B for long-read SVs**).

Also, the use of a non-standard genome reference (HCC1395BL_v1.0) precluded us from adopting the reviewer's recommendation to use Bioconductor StructuralVariantAnnotation package. This was similar to the situation when using the PURPLE/LINX. StructuralVariantAnnotation appears to have dependency on annotation from a standard reference genome (hg38 or hg19 for human) (e.g.,

VariantAnnotation::readVcf (vcf.file, "hg19"), and we are not sure how non-GRCh37/38, like HCC1395BL_v1.0, would fit into this package. In our analysis, we used "**SURVIVOR**" to merge and generate consensus SVs as those authors performed in their respective papers (GRIDSS2 paper:

<https://genomebiology.biomedcentral.com/articles/10.1186/s13059-021-02423-x> ; "Truth" SV set paper:

<https://www.biorxiv.org/content/10.1101/2020.10.15.340497v1>). In the consensus callsets from different callers, we

combined the counts of DUP and INS to uncover the differences between HCC1395BL_v1.0 and GRCh38, which was sufficient for this study.

2) cis-phased adjacent SV handling

In your reply you dismissed this as hard/not relevant but I think it's two important to be dismissed so lightly. 22% of somatic SVs are within 1kbp of another somatic SV and this rises to 31% by 10kb and even on a relatively stable genome such as COLO829, this caused 5 of the 67 SV in truth set to be called differently between the short and long read callers

(<https://genomebiology.biomedcentral.com/articles/10.1186/s13059-021-02423-x>). This is too big of an effect size to ignore completely. The simplest way to do this would be to considered cis/trans phasing in flanking sequence matching logic and match if any of the potential flanking sequences match.

[Response]

Thank you for your comment. Per your recommendation, we have added the GRIDSS2/GRIPSS caller as part of our short-read somatic SV analysis. According to GRIDSS2 paper

(<https://genomebiology.biomedcentral.com/articles/10.1186/s13059-021-02423-x>), GRIDSS2 is a single-breakend based somatic SV caller with **the ability to phase somatic structural variants, particularly for complex rearrangements** (*"GRIDSS2 simplifies complex rearrangement interpretation through phasing of structural variants with 16% of somatic calls phasable using paired-end sequencing"*). As a result, we do not need to reinvent the wheel for specifically handling cis-phased adjacent SVs as GRIDSS2 does (for demonstrating its advantages over other callers). Specifically, we evaluated how somatic SV calling using GRIDSS2 (along with other 3 callers) differed between use of a personalized genome reference (HCC1395BL_v1.0) and the standard

GRCh38 when using short-read data from all 12 paired tumor-normal replicates (**Figure 7**). With GRIDSS2 (somatic SV phasing handling), we demonstrated that generally more somatic SVs were detected based on HCC1395BL_v1.0 compared to GRCh38, and we were able to detect a set of novel somatic SVs specific to HCC1395BL_v1.0, not to GRCh38 (**Figure 7**). By requiring somatic SV calls to be supported by 2 or more callers (including GRIDSS2/GRIPSS, Manta, Delly, and novoBreak), each of four callers still reported more somatic SVs with HCC1395BL_v1.0 reference as compared to GRCh38 with additional somatic SVs detected only when HCC1395BL_v1.0 was used as reference (**Figure 6, Supplementary 9A/9B**).

3) Correct handling of event type.

- It's unclear if INV call are actual inversions or just +/- and -/- oriented breakpoints. DELLY uses the CT INFO field (e.g. CT=3to3) to specify these breakpoints. **The vast majority of your "inversions" are not actually inversions.**

Furthermore, I expected to see more interchromosomal calls in the personalised genome as the scaffolding is less complete.

Proper classification of somatic breakpoints is hard (e.g. Figure 2 of <https://www.biorxiv.org/content/10.1101/2020.12.03.410860v1>) so since it's too much work to do here, either a) just turn everything into breakpoints and compare breakpoints, or b) special case simple DEL/DUP events (although you're not going to be able to differentiate between them and DEL/DUP-like breakpoints that are part of complex rearrangements) if you find there's actually a difference there, or c) use a DEL/DUP/INV/BND classification and explicitly **state that INV is any +/- or -/- oriented intra-chromosomal breakpoint.**

[Response]

We would like to first answer your question regarding inter-chromosomal calls (translocations, TRA) in your comment "*expected to see more interchromosomal calls in the personalized genome as the scaffolding is less complete*". In fact, we observed that trend; all SV callers reported more translocations based on HCC1395BL_v1.0 reference as compared to GRCh38 (**Supplementary Figure 8B/8C**).

Next, we address the comment regarding INV and SV classification here. In our short-read SV analysis, we simply took what each caller reported in terms of SV types before we compared callsets from different callers. Each caller has its own mechanism to classify SV events with respective SV types, and we tried to avoid reinventing the wheel to redefine or reclassify SV-types from each caller's output, nor did we try to compare SV callers to figure out which caller defines SV type more correctly. Our major goal is to evaluate how each caller's results differ based on use of the personalized genome reference as compared to standard GRCh38, using the same set of reads.

As you stated here, **proper classification of somatic SV types is hard, and needs "too much work to do"**, and in fact this is completely out of our scope for this study. Most importantly, the classification tool **PURPLE/LINK**, which you recommended, and we have explored, only worked for standard reference genomes such as GRCh38, and not for a personalized genomes like HCC1395BL_v1.0. To accommodate this, a number of reference genome-specific assumptions in PURPLE and the input components (such as **AMBER and COBALT**) need to be modified or regenerated. Based on our recent communications with the developer, these modifications are not currently available at this time. We also noticed that like Delly's notation for inversion (INV), **the function "simpleEventType" in Bioconductor StructuralVariantAnnotation package would classify those breakpoints with both ++ and -- "as inversions regardless of whether both breakpoint that constitute an actual inversion exists or not"**

(<https://www.bioconductor.org/packages/release/bioc/manuals/StructuralVariantAnnotation/man/StructuralVariantAnnotation.pdf>, Page 21). Presumably, GRIDSS2/GRIPSS's classification program (**simple-event-annotation.R**) follows this same logic.

Based on our short-read somatic SV analysis using 4 callers, we found in merged consensus SV sets, vast majority of SVs had consistent SV types in SV regions, while very few (just a handful or less) SVs overlapped with 2 different SV types (some are DUP/INS). Specifically for inversions as you pointed out, we compared the inversion calls from GRIDSS2 (recommended by reviewer, breakpoint/BND-based calling initially, then classifying into SV types using **simple-event-annotation.R**), and found INV calls were mostly consistent. Because these INVs were classified by GRIDSS2/GRIPSS, and were consistent with Delly's INV calls, we respectively submit that the reviewer's opinion "*The vast majority of your 'inversions' are not actually inversions*" is not sufficiently supported (P2P-Table 2).

Please note that we were just users of somatic SV callers, and developing a better algorithm for SV calling was never our purpose for this study.

	# INVs	# INVs matching GRIDSS2
GRIDSS2	38	38
Manta	45	37
Delly	21	20
novoBreak	37	34
Total (merged)	49	

P2P-Table 2. Summary of inversions (INVs) detected by four callers and their matching INVs as compared to that from GRIDSS2.

4) FP/FN estimation

You've made not attempt to determine the FNR/FDR of the calls from any of your callers. There's not even an analysis of the consistency across callers. If you don't like the idea of consistency across callers as your proxy for truth then you should consider using copy number consistency as a proxy. The GRIDSS2 authors used this to compare the caller on the PCAWG and Hartwig cohort and, although you've only got a single sample, it would be useful to at least estimate the FDR/FNR.

[Response]

Rather than establishing a ground truth set of accurate SV calls, our analysis with short-read somatic SV calling in this manuscript was really focusing on how somatic SV calls were impacted based on these two genome references (GRCh38 vs HCC1395BL_v1.0) using representative callers that have different SV detecting algorithms. Unlike a manuscript introducing a new somatic SV caller, we are not attempting to compare the callers to one another for demonstrating the superiority of the new caller (<https://genomebiology.biomedcentral.com/articles/10.1186/s13059-021-02423-x>). It seems that the reviewer was asking us to follow GRIDSS2 authors' approaches to perform a similar analysis for our manuscript, but the objective of these two manuscripts are completely different. While we could provide some figure like the one (P2P-Figure 2) shown below (sensitivity vs precision for 4 callers), we think estimating these callers' FDR/FNR is not relevant to the primary objective of our manuscript.

P2P-Figure 2. Plot of sensitivity and precision of four short-read somatic callers. The truth set was defined by SV calls supported by two or more callers. (1) GRIDSS2: yellow circle; (2) Manta: blue circle; (3) Delly: green circle; and (4) novoBreak: red circle.

5) Analysis of the actual reference-induced differences This is closely related to point 1. You haven't demonstrated why your reference is actually better for SV calling. Unlike germline SV, somatic SVs aren't enriched for low complexity sequence (except for LINE reactivation and MSI) so given you only have one sample, I don't expect a statistically significant difference between the reference genomes. There should only be a handful of somatic SVs that don't have matching calls in both references. This is clearly not the case from your results. Like you did with SNVs, you need to show that the few % of SVs that are different are different because of reference differences.

[Response]

As you mentioned, this question is related to your Point 1 above. In this revised manuscript, we mapped the flanking sequences of GRCh38-based SVs (using both short-reads and long-reads) to HCC1395BL_v1.0 reference and generated equivalent SV calls between the two references. In the meantime, we also identified a set of somatic SVs that could only be discovered (with the same pipeline) using HCC1395BL_v1.0 as reference, not using GRCh38 (**Figure 6C/6D**, **Figure 7B/7C/7D** for short-read SVs, and **Figure 8B** for long-read SVs), because certain personalized sequences were uniquely presented on HCC1395BL_v1.0 reference (e.g., some *Alu*, or simple repeat sequences), but were missed on GRCh38, as we have demonstrated in several examples using IGV and read-mappings (**Supplementary Figure 9A/9B** from short-read DELs, **Supplementary Figure 12A-12E** from long-read DELs). All these results demonstrated the importance of using a personalized genome as reference for accurately detecting somatic SVs with short-read sequencing data in tumor-normal cell lines.

6) [Minor] Consistency of results

You have lots of replicates. Why not repeat the SV analysis across replicates to see how consistent the call are?

[Response]

In this revision, we extended our analysis to cover all 12 tumor/normal paired replicates using GRIDSS2, Manta, Delly, and novoBreak somatic callers. SV calls were required in two or more replicates to be

included for further comparison for each of four callers. Similarly, somatic SV counts were generally higher when using HCC1395BL_v1.0 as reference as compared to GRCh38 for all callers, and we identified equivalent SV sets between two references, as well as HCC1395BL_v1.0 specific SV sets for each of the four callers (Figure 7A/7B/7C/7D).

Specific points:

- > We hypothesized that the numbers of the predicted SVs by Pindel would
- > decrease due to the reduction of split-reads observed in
- > HCC1395BL_v1.0- based alignments as compared to GRCh38 based
- > alignments (Figure 4D)

Reduction in split-reads comes from two things: 1) fewer real SVs, 2) reduction of chimeric fragments generated during sequenced. The latter does not depend on the reference so the expected reduction must come from the former. For germline you would expect fewer calls but, as Figure 4D indicates, we're talking about somatic SVs and we expect either the same, or more (if the personalised reference is better) SVs. By what mechanism are you hypothesizing a reduction in the number of events? The only way I could explain this is if you expect most SVs in your call set to be false positives.

- > undoubtedly demonstrated the benefits of using the personal genome
- > HCC1395BLv1.0 in reducing potential false positives

You've provided no evidence that the reduction is caused by having fewer false positive and not more false negative. I interpret this result as the personalised reference being worse for Pindel somatic SV calling than hg38.

[Response]

As mentioned in this Point-to-Point Response, in the previous version of manuscript, the purpose of using Pindel (split-read method) was to evaluate how the SV calls (germline and somatic) would be different using personalized genome reference versus standard GRCh38. Reduction of split-reads with HCC1395BL_v1.0 reference (Figure 4D) would have impacts on germline SVs as we demonstrated the total SVs were reduced with HCC1395BL_v1.0 reference in our initial calls. This also suggested that there would have **more noisy or false positive germline calls in GRCh38-based SVs.**

Regarding somatic SVs calls by Pindel based on the filtering using something like “Somatic SV calls = $(SV_Calls_{Tumor_Sample} - SV_Calls_{Normal_Sample})$ ”, we observed that “somatic” SV calls were generally reduced for Pindel in our analysis as shown in previous version (Figure 6A). We agreed that such custom filtering may not be ideal for obtaining somatic SVs, and more strict filtering may be needed to improve the accuracy, but we tend to believe there were more noisy or false-positive calls in the somatic SV calls by using “Somatic SV calls = $(SV_Calls_{Tumor_Sample} - SV_Calls_{Normal_Sample})$ ”, given that we applied same set of rules for the filtering using the same set of short-reads, and the only difference was the reference genome. That is why we need to use personalized assembly as reference for both germline and somatic SV calling, given that all the improved statistics for the personalized assembly and short-/long-read mappings (reduction of mismatches, split-reads, discordant reads, and increased uniformity of read coverage, smaller standard deviations in library insert sizes etc.) have been illustrated in Figure 4.

Nonetheless, we have removed results from two germline callers (Pindel and BreakDancer) from our revised manuscript. With two modern short-read SV callers (GRIDSS2/GRIPSS and Manta) being included in this revision, we believe there is no need to deal with those custom filtering anymore based on the results from germline callers.

> BreakDancer

Please remove your BreakDancer results. BreakDancer's performance cliff is dangerous and your paper doesn't specify the real length and fragment size so I don't even know what side of the performance cliff your results are on.

> BreakDancer clearly identified 35.56% (on average) fewer supporting reads for normal sample, and 32.34% (on average) fewer for tumor sample with the personal genome reference as compared with GRCh38 (Supplementary Table 9), which could be explained by the decrease of non-properly-paired reads (Figure 4B).

Unfortunately, it cannot. A personalised reference is supposed to reduce the discordant pair rate by including the SV in the reference. Real somatic SVs that remain should have exactly the same support. Supporting read counts for real somatic SVs that much lower indicates that the personalised reference is much worse for BreakDancer SV calling.

[Response]

We have removed results from two germline callers (Pindel and BreakDancer) from our revised manuscript. With two modern short-read SV callers (GRIDSS2/GRIPSS and Manta) being included in this revision, we really do not need to deal with those custom filtering anymore based on the results from germline callers.

However, we would like to point out that the statistic you referred to above for BreakDancer was for "germline" SVs only, not "somatic" SVs in the previous version of our manuscript. Thus, the subsequent comment would not be valid. Your comment "**Real somatic SVs that remain should have exactly the same support**" may not be entirely true, which may be applicable to the unique regions of the genome. But once we get into non-unique regions, such as repeats, segdups, the short-read mapping would be different. That's why we have that many different short-read SV callers (germline or somatic) with varying concordance and validation rates (as compared to SNVs). And that's why we need long-read data to resolve SVs in those non-unique regions (with repeats, segmental duplication etc.). You may also discount the novel sequences that presents in personalized HCC1395BL_v1.0 reference only, but not in GRCh38 as we have demonstrated in this revision (**Supplementary Figure 9A/9B** from short-read DELs, **Supplementary Figure 12A/12B/12C/12D/12E** from long-read DELs). Somatic DELs would not be able to detect when GRCh38 was used as reference. Those somatic SVs are without counterparts on GRCh38, but they would not have exactly the same read support.

> Additionally, Delly reported 3 insertions on HCC1395BL_v1.0, but none on GRCh38, while novoBreak did not report any insertion event (Supplementary Figure 7A).

So what are those three calls? Are they microsatellites? Why are the present only on BLv1.0? These differences are the interesting one because they look like they could be genuine reference differences.

[Response]

Thank the reviewer for the comment. Of course, these two genome references are not the same. The 3 insertions on HCC1395BL_v1.0 reported by Delly (and other callers as well) were:

- (1) scaffold_47:144105:INS:**66bps**
(TGCTGCTAGTTGGGTAGCAGGTAACGGTGTCCATGGTCCCTGACCCAGCAGTCTCCTGGCTTCTGCC) called by GRIDSS/Manta/Delly;

- (2) scaffold_49:9684519:INS:60bps
(GGTTAAAGCACTGTACAGTGCTGGATGGAATGTGAATTAGTCCAAGCACTGTGGAAAGCA) called by GRIDSS/Manta/Delly;
- (3) scaffold_129:31804:INS:78bps
(GTAAAATGCAGTCATCACTTTCAACACTTTGTCATGCATGCATGTAATAGGACAGAGGTGAGCATTGTGGGTAATCTC) called by Manta/Delly.

But none of them was detected with GRCh38 reference. It seems that those 3 insertions were not classified as microsatellites (<https://www.repeatmasker.org/cgi-bin/WEBRepeatMasker>). We also checked that all these 3 insertions were not located in the gene region on HCC1395BL_v1.0 reference. However, one novel insertion detected by Manta (scaffold_1:105582094:INS:52bps, TTTCTGTCTAACTTTTTTCCCTGCCTCCCTCCCTCCCTCCCTCCCTCC, simple-repeat) overlaps with intronic region of LIN5 gene.

Clearly, these two references (GRCh38 and HCC1395BL_v1.0) have some differences, resulting in different read mapping, and thereafter different SV calls, like those insertions here, and many other deletions we have inspected that were detected only on HCC1395BL_v1.0, but not on GRCh38 due to the missed sequences, sometimes *Alu* repeat etc. on GRCh38 (**Supplementary Figure 9A/9B, 12A/12B/12C/12D/12E**).

In this revision, we summarized these insertion analysis in **Results** section (Page 19) as below:

“Additionally, GRIDSS2 reported 2 insertions (scaffold_47:144105:INS:66bps and scaffold_49:9684521:INS:60bps called by GRIDSS/Manta/Delly, but not in gene region), and Delly reported 3 insertions (scaffold_47:144105:INS:66bps and scaffold_49:9684521:INS:60bps called by GRIDSS/Manta/Delly, and scaffold_129:31804:INS:78bps called by Manta/Delly, not located in the gene region) on HCC1395BL_v1.0 reference, but none on GRCh38, while novoBreak did not report any insertion event. Manta detected 20 insertions on HCC1395BL_v1.0 (one insertion scaffold_1:105582094:INS:52bps overlapping with a DUP scaffold_1:105582094-105582146 by GRIDSS/Manta, intronic region in LIN5 gene), but only 10 insertions on GRCh38 (**Supplementary Figure 8B**), suggesting that the personalized HCC1395BL_v1.0 as reference may have better sensitivity for somatic insertion detection with short-read data for this pair of samples, due to the improvements in personalized genome reference and read mapping on HCC1395BL_v1.0.”

>Noticeably, the percentages of overlapping DELs (as well as DELs 50% or more by length) detected by both Delly and novoBreak on HCC1395BL_v1.0 were slightly higher (5% for Delly, 5.9% for novoBreak) than on GRCh38 (Figure 6D).

Overlapping DELs are almost certainly chromothripsis or BFB breakpoints and not actually overlapping DELs. Whether they overlap by 50% or more is not in actually meaningful. More overlapping DELs either means more FPs or more TPs. You need to show that these extra chromothripsis-like DELs (or are they fragile site/rigma?) are real (or at the very least, there is a change in CN at the breakpoint) and not just that the personalised reference results has a higher FDR or is misassembled thus causing more simple DELs to appear to be overlapping.

[Response]

We are sorry that the reviewer apparently misunderstood our statement regarding “*the percentages of overlapping DELs (as well as DELs 50% or more by length) detected by both Delly and novoBreak on HCC1395BL_v1.0 were slightly higher (5% for Delly, 5.9% for novoBreak) than on GRCh38 (Figure 6D)*” in the previous version of manuscript, and we feel sorry for not making this more clear.

The point we were trying to make in Figure 6D (previous version of manuscript) was that the somatic deletion results of two callers were more consistent on HCC1395BL_v1.0 because both callers had more deletions in common on HCC1395BL_v1.0 as compared to GRCh38, indicating the personal genome reference (HCC1395BL_v1.0) is better. **There was nothing to do with adjacent deletions that were overlapping in the same locus, nor with chromothripsis.**

However, in the revised manuscript, we have added two more modern somatic SV callers (GRIDSS2/GRIPSS and Manta), replacing two germline SV callers (Pindel and BreakDancer), and evaluated how the two genome references are impacting somatic SV calls that were supported by at least 2 SV callers, showing that more such somatic SVs were discovered with HCC1395BL_v1.0 reference than that with GRCh38 (**Figure 6B**), and some of HCC1395BL_v1.0 reference specific SVs were detected only when using personal reference, not using GRCh38 due to the missed sequences (**Supplementary 9A/9B**), and some of those novel SVs were related to important pathways (**Figure 6E**).

>Fig6D

Why do all the other this axes start at 0 and this one doesn't?

[Response]

Thank you for pointing out this, and this would be an easy fix. But this figure (regarding overlapping deletions between Delly and novoBreak) is no longer included in revised manuscript, as we have performed more comprehensive analysis based on four somatic SV callers, e.g., requiring the SV calls to be supported by at least two of the four callers and evaluating how different they are with HCC1395BL_v1.0 reference as compared to GRCh38.

> Detection of somatic SVs using long reads was improved using a

> personalized genome reference

What's the concordance between PBSV and Sniffles?

[Response]

Thank you for the question. Based on merged consensus somatic SV calls from 4 different calling methods (combination of 2 aligners and 2 callers) with requirement of SVs in 2 or more methods, the concordance rate for the somatic SVs with personalized genome reference between PBSV and Sniffles2 could reach 87.55% for PBSV and 87.78% for Sniffles2.

>Most notably, somatic insertions were decreased significantly by 32.8%

>for Sniffles and 27.3% for PBSV

Doesn't this mean worse (LINE/microsatellite) sensitivity on BL_v1.0?

[Response]

Thank you for the question. Firstly, whether the counts of insertions were particularly correlated with LINE or microsatellite is not known, and there seems to be no reports that the insertions must be LINE or microsatellite sequences. Thus, it would be hard to link insertion detection with something related to LINE/microsatellite sensitivity. Secondly, our somatic SV calls from PBSV/Sniffles in the previous version of manuscript still contained some amount of germline SVs due to insufficient filtering (thus we

reanalyzed PacBio long read with more stringent criteria, see **Methods**), therefore, the relevant statistics would be still more or less applied to germline SVs. We have seen that the germline INS counts would be much fewer with HCC1395BL_v1.0 reference (whether using short-reads or long-reads) as compared to GRCh38, which could be explained by that GRCh38 contains systematic collapse of repeats (eg. un-resolved repeats) during the construction of GRCh38 as reported by others (M. J. P. Chaisson et al, 2015, Resolving the complexity of the human genome using single-molecule sequencing. *Nature*. 517, 608–611]. Such bias would have impacts on germline SV detections with tendency to **call more insertions** [Miga, K.H., et al., 2020, *Telomere-to-telomere assembly of a complete human X chromosome*. *Nature*. 585, 79-84. Extended Data Fig. 3. Zook, J.M. et al. 2020. A robust benchmark for detection of germline large deletions and insertions. *Nat Biotechnol* 38, 1347–1355. Fig. 3].

> overlapping somatic deletions between Sniffles and PBSV

Again, it’s unclear what, if anything, this means.

If you have breakpoint classifications (e.g. from LINX -

<https://www.biorxiv.org/content/10.1101/2020.12.03.410860v1.full.pdf>) then you could say something about how efficient (or not) the recovery of complex events was but without it or a ground truth these result don’t actually answer the question about which reference is better.

[Response]

This question was similar to the one related to the overlapping deletions between Delly and novoBreak. We feel sorry for not making the statement more clear.

Again, the point we were trying to make in **Figure 7B** in the previous version of manuscript was that, the somatic deletion results of two callers (Sniffles and PBSV) were more consistent with HCC1395BL_v1.0 reference because both callers had more deletion in common when using HCC1395BL_v1.0 as compared to GRCh38, indicating the personal genome reference (HCC1395BL_v1.0) is better.

As mentioned earlier, due to the limitations of the **PURPLE/LINX** pipeline with non-GRCh37/38 standard reference genome, we were not able to perform PURPLE/LINX with HCC1395BL_v1.0-based SVs.

However, in the revised manuscript, we have re-analyzed the PacBio long-read data using 4 calling methods (2 callers: PBSV/Sniffles with 2 aligners: pbmm2/ngmlr) with more stringent filtering criteria for somatic calls, and demonstrated that more somatic SVs (particularly for DELs, DUP/INS, and INV) were detected when using HCC1395BL_v1.0 as compared to using GRCh38 (**Figure 8A, Supplementary Table 10**), while most are equivalent, we did find 84 GRCh38-based SVs were not mapped on HCC1395BL_v1.0, and 119 SVs were mapped but without matching SVs on HCC1395BL_v1.0. But we identified 279 HCC1395BL_v1.0 specific SVs (including 217 DEL, 61 DUP/INS, and 1 INV) that were without matching GRCh38-based SVs on HCC1395BL_v1.0. Among them, 91 SVs (72 DEL and 19 DUP/INS) were mapped to 86 gene regions. KEGG pathway enrichment analysis suggested that some of these 86 genes involved in pathways related to cancer invasion and metastasis (e.g., CDH23 gene, ST14 gene), PI3K-Akt signaling pathway and G protein-coupled receptor signaling pathway (e.g., GNG7 gene) (**Figure 8C, and Supplementary Table 12**). We manually checked several somatic DELs using IGV, and found that those deletion examples were all valid somatic events with HCC1395BL_v1.0 reference, but not detected with GRCh38 (**Supplementary Figure 12A-12E**) with confirmation of mapping extracted reads, indicating that the personalized genome reference has unique advantages over GRCh38 in detecting these somatic SVs.

>SV discovery using an assembly-to-assembly mapping approach was
>improved using a personalized genome reference

The number of SVs reported in this section is so much high than the short read and long read results that I don't believe they're correct. Your assembly-to-assembly comparison has way more SVs than any of the samples in either PCAWG or Hartwig that I don't see how they can be correct. You need to sanity check your contig mapping and show that putative breakpoints are supported by your short and long read data. To me this result just says either a) your somatic assembly is wrong, or b) your somatic assembly has duplicate contigs, and/or c) your somatic assembly has split up amplified SVs into separate contigs thus resulted in inflated SV counts. You can use the same breakpoint flanking matching logic to determine which of the 3 options is true or if you really do have an excessively rearranged sample. Note that a perfect assembly of a breakage-fusion-bridge event will linearise the amplified region and an assembly-to-assembly comparison of such an assembly will have a much higher SV count than would be called by the equivalent perfect reference-based read alignment (e.g. a CN=20 breakpoint would be called as 1 reference-based breakpoint, and 20 assembly-based breakpoints). High counts are not necessary wrong, you just need to explain them.

In conclusion, the results presented does not support your conclusion that the personalised reference improves SV calling. More/different analysis is required.

[Response]

Thank you for your question regarding the SV counts based on contigs mapping onto HCC1395BL_v1.0 and GRCh38. We went back to check our analysis and found that some amounts of germline SV calls were still not being filtered out from the contig callsets, due to the input SVs containing only AF0.5+ for normal sample by PBSV/Sniffles.

In this revision, we have re-analyzed somatic SVs based on contig mapping onto HCC1395BL_v1.0 and GRCh38 using more stringent filtering as described in **Methods** section. Now the counts of somatic SVs identified with contig mapping were much smaller and comparable to the counts identified by other PacBio long-read calling methods (**Supplementary Table 10**). Like the analysis with PacBio long-read somatic SV calls, we were also able to find some GRCh38-based SVs that were un-mapped or mapped but without matching SVs with HCC1395BL_v1.0 reference. We were also able to identify some HCC1395BL_v1.0-specific somatic SVs that were not matching GRCh38-based SVs. Due to the nature of such somatic SV set that were generated using similar approaches, for consensus somatic SVs, we combined these 2 contig mapping based somatic SV sets (by Assemblytics/paftools) along with 4 PacBio long-reads based somatic SVs together (by Sniffles2/PBSV with pBMM2/ngmlr) to evaluate how two genome references were affecting somatic SVs that were supported by at least two calling methods (**Figure 8**).

>We included about 175-fold of Illumina short reads that were pooled
>from 3 replicates (FDN1, FDN2, and FDN3)

You pooled 3 different libraries together? You can't do that unless they have the same fragment size distribution (which they won't). Since fragment size is so important for SV calling, callers typically require you to specify them as separate BAM files. If you're lucky your caller will support read groups but very few do.

Please include the median fragment size for each library.

[Response]

Thank you for your comments. The sentence “that were pooled from 3 replicates (FDN1, FDN2, and FDN3)” was not exactly accurate to describe what we did for the relevant data. Actually, we were merging the three individual bams together (as FDN123) from three replicates (FDN1, FDN2, and FDN3) to increase the total coverage so as to detect SVs more sensibly (doi.org/10.1093/bib/bbaa056) as they have very close **median** insert sizes. We have made appropriate changes to reflect this fact in this revision (**Methods** section, Page 31, Page 35).

Please note that initial bam was generated with individual replicate (library), thus the library information was retained in each replicate bam, and such library information would be carried over when three bams (FDN1, FDN2, and FDN3) from the normal sample (FD center) were merged (as FDN123), so that the caller would take advantage of. Similarly, three bams (FDT1, FDT2, and FDT3) from the tumor sample were merged (as FDT123). This was the ONLY pair of merged bams (FDT123/FDN123) for our initial SV evaluation for Delly and novoBreak in the previous version of manuscript, and now for GRIDSS2/GRIPSS and Manta in this revision.

In addition, all three replicates were done in a single laboratory with identical protocol for gDNA fragmentation with intended fragment size between 300-400 bps. The insert fragment size derived from 5’ forward strand read and 3’ reverse strand read, as shown in the table below (**P2P-Table 3**), were in line with the intended fragment size, and the differences were extremely small.

Most importantly, all four somatic SV callers we selected, including Delly, novoBreak, GRIDSS2, Manta, do not need to specify insert size for SV calling (**Methods** section, Page 37). Only the Pindel SV caller needs to specify insert size in its three-column bam_config file (bam_file, insert_size, Sample_Name). But in the previous version of the manuscript, we only analyzed 3 replicates (from FD sequencing center) individually, not on merged bams. In this revision, we removed Pindel results completely.

	Normal	Tumor
FD-Replicate1	367	377
FD-Replicate2	371	375
FD-Replicate3	368	371

P2P-Table 3. Median library insert sizes for 3 FD replicates.

>Pindel Somatic SVs were filtered on the VCF file with no supporting reads in the matched normal sample [30].

>BreakDancer somatic SVs were filtered with no supporting reads in the matched normal sample [30]. That won’t filter germline SVs that have don’t have any germline coverage at all.

I recommend using actual somatic SV callers, as it avoids annoying review comments complaining about how your custom filtering is not representative of actual usage.

[Response]

Thank you for your suggestion regarding Pindel and BreakDancer, both of which are germline callers. With two modern short-read SV callers (GRIDSS2/GRIPSS and Manta) being included in this revision, we really do not need to deal with those custom filtering anymore based on results from germline callers using something like “Somatic SV calls = (SV_Calls_{Tumor_Sample} – SV_Calls_{Normal_Sample})”.

[Minor issues]

- > Since the bit (flag 0x2) for properly paired reads was set by the
- > aligner (BWA-MEM), there was no reason to believe that BWA-MEM would
- > not set the “proper paired” bit for a split-read alignments if the
- > orientation and insert size were satisfied, based on the post
- > (<https://www.biostars.org/p/79611/#81436>)

It's a minor point but you should be aware that bwa sets the proper pair flag based on the primary alignment. **If it's the supplementary alignment that's concordantly aligned, then bwa won't set the flag.** Here is an example from simulated data:

```
variant.chr12-art4601880 161 chr12 1527056 60 100M = 1528347 1347
ACTAAGTGCCAAACACTAAGTTGTAAGTTTATAATCACATTGTAAGGATGTTTTATTGAAACCAATATAACAATCC
TAAGAGATAGATACCATTTTCAT 5??A=BBBDDDD@D-DGCGGEGH/FAFFIFI/@FIFHIA+GHHHFFH-
@HH*5GF=IHGEHIEH+HHFEFGEHAHFGEFH<DHAH/EHHHIBHDHHE/=H M
C:Z:44S56M MD:Z:46C53 RG:Z:1 NM:i:1 MQ:i:60 AS:i:95 XS:i:21
variant.chr12-art4601880 2129 chr12 1527279 60 49M51H = 1527056 -272
TACCTCTGAAGGATACATAGCTAGAAGGATTTTTAAAGACGTTATTAT
HIHIIHHIHI7BHFFFHBFIIIHIAHIIIIGIHDEHDIHHIIFHDHSH SA:Z:chr12,1528347,-,44S56M,60,0;
MD:Z:49 RG:Z:1 NM:i:0 AS:i:49 XS:i:21
variant.chr12-art4601880 81 chr12 1528347 60 44S56M = 1527056 -
1347
TACCTCTGAAGGATACATAGCTAGAAGGATTTTTAAAGACGTTATTATAAATAAGACTAACATAAACTCCTTTTA
AAATCTGAGTAGAAAAGGACCCC
HIHIIHHIHI7BHFFFHBFIIIHIAHIIIIGIHDEHDIHHIIFHDHSHIGEIFAHIHIGFHIIIIGIHFFIIFDF8FGGDDDDDD
DD?@<??A5A SA:Z:chr12,1527279,-,49M51S,60,0; MC:Z:100M MD:Z:56 RG:Z:1 NM:i:0 MQ:i:60
AS:i:56 XS:i:20
```

[Response]

Thank you for your further explanation and providing examples regarding proper pair flag setting in BWA for primary alignment and supplementary alignment. In your case, both primary and supplementary alignments were no “proper pair” bits being set in their flags.

- Flag 161 = read paired, mate reverse strand, second in pair
- Flag 81 = read paired, read reverse strand, first in pair
- flag 2129 = read paired, read reverse strand, first in pair, supplementary alignment)

We re-posted the alignment case for a paired reads (<E00512:209:HFLVKCCXY:1:2120:26514:67427>) from last Point-to-Point Responses below for convenience/comparison to show a different case – **both the primary and the supplementary alignment were marked as “proper paired” in their flags** (<https://broadinstitute.github.io/picard/explain-flags.html>). Again, as we responded last time, **we relied on the aligner (BWA-MEM)** to correctly set the bit for properly aligned reads, and the tool “samtools stats” to collect read mapping information from the aligned bams (by BWA-MEM). Thus, we should have no issue on this subject.

flag 163 Read paired, proper pair, second in pair, mate reverse strand

Thank you for your suggestion, and collectively, our analysis showed that somatic SNVs/SVs results with personalized genome reference were indeed better.

> Our findings demonstrated that use of a personalized genome with individual specific haplotypes was essential for accurate detection of the full-spectrum somatic mutations in paired tumor-normal samples Missing an “of”; you haven’t demonstrated that it’s essential. AFAIK, the current state-of-the-art in short read somatic calling is the GRIDSS2+GRIPSS+PURPLE+LINX Hartwig pipeline which claims a sub-5% FNR/FDR at 100x coverage (<https://genomebiology.biomedcentral.com/articles/10.1186/s13059-021-02423-x>).

[Response]

Thank you for your suggestion, and we have made appropriate changes.

We’ve already demonstrated the essential values of using personalized genome as reference for accurate somatic SNV detection as compared to GRCh38 in the previous version of manuscript (**Figure 5**). We even provided wet-lab validation for some of HCC1395BL_v1.0 specific SNVs with good validation rate (**Supplementary Figure 6 and Supplementary Table 8**).

In this revised manuscript, we have added two more modern SOMATIC callers, GRIDSS2/GRIPSS and Manta as suggested (for replacing Pindel and BreakDancer) and expanded our analysis to cover all 12 tumor/normal paired replicates. Our somatic SV analysis using short-reads and long-reads continued supporting the essential values of using the personalized genome reference for detecting sample-specific SVs that would not be detected using the standard GRCh38 reference (**Figure 6, Figure 7, Figure 8, Supplementary Figure 9A/9B, Supplementary Figure 12A-12E**).

Therefore, we believe our claim that “use of a personalized genome with individual specific haplotypes was essential for accurate detection of the full-spectrum of somatic mutations in paired tumor-normal samples” was justified.

> The N50s of HCC1395 assemblies were significantly smaller than that of the corresponding HCC1395BL

Why? Are the contigs breaking at somatic SVs and are otherwise intact or is assembler failing because the genome does not have uniform coverage? HCC1395 probably has chromothripsis so either more genes are broken, or the assembly is worse (probably a combination of both).

[Response]

We thank the reviewer for this comment regarding smaller N50s and fragmentation in tumor cell line assembly.

Our assertion is based on the Extended Data Fig. 5: Karyotyping of HCC1395 and HCC1395BL (<https://www.nature.com/articles/s41587-021-00993-6/figures/9>) from the now named ref5 [Fang, L.T., et al., *Establishing community reference samples, data and call sets for benchmarking cancer mutation detection using whole-genome sequencing*. *Nat Biotechnol*, 2021. **39**(9): p. 1151-1160.]. The cytogenetic analysis in this paper showed that **the tumor HCC1395 cell line was hypertetraploid with chromosome counts ranging from 64-79 and a gain of 38-63 unidentifiable marker chromosomes (P2P-Figure 3)**. Thus, it was **NOT surprising** that assembly of HCC1395 was highly fragmented.

P2P-Figure 3 (a) Karyotype of HCC1395. Cytogenetic analysis was performed on ten G-Banded metaphase cells from HCC1395. Analysis pointed to a hypertetraploid line with chromosome counts ranging from 64-79 and gain of 38-63 unidentifiable marker chromosomes. **(b)** Karyotype of HCC1395BL. Cytogenetic analysis was performed on ten G-banded metaphase cells from HCC1395BL. All ten cells showed loss of a chrX and an unbalanced whole arm translocation between the long-arm of chr6 at band q10 and the short-arm of chr16 at band p10. This resulted in a net loss of one copy of the short-arm of chr6 and loss of one copy of the long-arm of chr16. The abnormal chromosome could be placed in either a chr6 or chr16 locus as we were unable to determine if the centromere belongs to chr6 or chr16.

 Figure 2 could be improved by changing the colour of the discordantly aligned contig fragment to black and plotting them over the top of the others. Eg. the small scf2 is mostly chr1 but there's a bit that goes somewhere else and I can tell where it is. If you also highlight where the centromeric repeats are in the light blue scaffolds that'd answer whether that is chr8 alpha satellite repeat or not.

[Response]

Thank you for the comment. The small part in scaffold_2 connecting to the other chromosome you mentioned here was not a chromosome 8 alpha satellite repeat. Actually, the small part (SCF2:66536407-66830783, ~294kb) from the scaffold_2 (corresponding region chr1:83182174-83489744 on chromosome 1, p31.1) was mapped to a known segmental duplication region (chr7:76651385-76946262) on chromosome 7 (q11.23) on GRCh38 based on our mapping thresholds. This does not imply that this part of the scaffold_2 was misassembled or discordant mapped, as we had another smaller scaffold (39.6 kb, scaffold_934:0-39522) that was also fully mapped to the same segmental duplication region on chromosome 7 (chr7:76924063-76963636). The purpose of the Figure 2 was to demonstrate the **overall mapping consistencies between HCC1395BL_v1.0 and GRCh38**, not to emphasize these very few connections to other chromosomes, which would be very distracting with a suggested "black" line on the plot.

 I'm still a bit confused about the HLA typing result. Does HCC1395BL_v1.0 contain just the HLA-DR51 haplotype, or did you assemble two haplotypes?

[Response]

HLA gene is located on chromosome 6p, and Karyotyping of HCC1395BL cell line (Extended Data Fig. 5: Karyotyping of HCC1395 and HCC1395BL, <https://www.nature.com/articles/s41587-021-00993-6/figures/9>, from Fang, L.T., et al., *Establishing community reference samples, data and call sets for benchmarking cancer mutation detection using whole-genome sequencing*. *Nat Biotechnol*, 2021. **39**(9): p. 1151-1160) showed that this cell line had a net loss of one copy of the short-arm of chr6 and loss of one copy of the long-arm of chr16, thus, the region of HLA for this normal cell line was essentially haploid (**Results**, Page 10), and this suggested that we had only one haplotype for this HLA region (**Figure 3B**). On HCC1395BL_v1.0 assembly, HLA coding genes were located on Scaffold_30. Based on our analysis, the haplotype of HLA-DRB (labels in red) in HCC1395BL_v1.0 consists of HLA-DRB1 and HLA-DRB4 genes (**human HLA-DR53 haplotype group**), while GRC38 primary assembly contains HLA-DRB1 and HLA-DRB5 genes (**human HLA-DR51 haplotype group**). The human HLA-DR53 haplotype was represented only in GRCh38 ALT_REF_LOCI sequences. Scaffold_30 was a reverse complement mapped onto GRCh38 entirely, but the order of HLA genes on the chromosome or scaffold was identical between GRCh38 and HCC1395BL_v1.0 (**Figure 3B**).

Figure 4: it would be good to indicate the **legend labels** that HCC1395BL/HCC1395 refer to the aligned reads from that replicate, not the reference. Even something as simple as having “Reads from” would reduce the cognitive burden of understanding what’s being plotted.

[Response]

Thank you for your suggestion, and we have made appropriate changes. The “HCC1395BL/HCC1395” really refer to specific sample names (normal and tumor sample) related to their read mapping statistics. To avoid confusion with HCC1395BL_v1.0 reference name, we added a “sample” after the existing labels “HCC1395BL/HCC1395” in Figure 4B/4C/4D.

Page 18L33: please add a sentence making it clear when you’re talking about somatic SVs, and when you’re talking about germline.

[Response]

Thank you for your comment. In this revised manuscript with regarding short-read SV calling, we removed germline callers (Pindel and BreakDancer) and added two new modern somatic callers (GRIDSS2/GRIPSS and Manta), therefore, all the text in that section were all about somatic SVs.

[out of scope] Why are there more GATK calls? I would have expected fewer since GATK will filter calls that it can’t genotype and the presence of aneuploidy massively messes up SNV genotyping.

[Response]

This is related to a question you raised last time regarding the comparison between the results using germline caller GATK _(Tumor - Normal) for somatic calling and the somatic calls using somatic caller (e.g., MuTect2).

Firstly, the two callers (GATK and MuTect2) are using different models to call variants, and each caller will report different numbers of variants with some internally set criteria. Secondly, results from

MuTect2 caller can be fit into a handy filtering program (FilterMutectCalls), so that many low-quality or false-positive calls could be removed. But we do not have a similar handy program to do the further filtering for GATK (Tumor - Normal) result, which means GATK (Tumor - Normal) result would contain more low-quality or false positive calls than that from MuTect2+FilterMutectCalls.

> Pindel RPL

One could argue that these aren't really SVs at all (and I've seen RPL-like sequencing artifacts) and could be just a bunch of adjacent SNVs/indels.

[Response]

RPL calls were reported by Pindel only, which is a replacement with inserted sequence around the breakpoint (for example, a 10kb deletion with 5 bp insertion), according to the author's explanation (<https://www.seqanswers.com/forum/bioinformatics/bioinformatics-aa/4089-pindel-improved-version-for-indels-and-structural-variants/page6?postcount=79#post204058>).

But in our revised manuscript, we have removed Pindel (a germline SV caller), as we added two modern somatic SV callers, GRIDSS2/GRIPSS and Manta. Thus we should have no issue.

Reviewer #2:

The revised manuscript addresses the concerns in the review and is substantially improved.

Reviewer #3:

The authors have reworked the m/s substantially and added some missing analyses. The presentation is definitely improved but the story is still rather lackluster overall. So much effort and money spent on this personal reference – but the real outcome is that SD is a bit lower and read mapping has improved. This had been expected from the start, hadn't it? With the personal reference the readers will expect results that are less abstract and closer to a patient, otherwise they will be disappointed. I think there is a way to make this paper exciting, but it requires more work. Given that the authors rebutted some comments as being outside the scope I should say this IS the scope, what's the point of the personal genome otherwise?

[Response]

Thank you very much for your comment. We actually have different views regarding the benefits of using personalized genome as reference. You mentioned the improvement of read-mapping (short-/long-reads), which was really just one part of the story. The differences of somatic SNVs between GRCh38 and personalized genome reference (HCC1395BL_v1.0) were almost 10%, and the exact SNVs were 88.25% (Figure 5C), not mentioning there were additional HCC1395BL_v1.0 specific somatic SNVs, some of which involve important pathways (Figure 5D, Supplementary Table 6, Supplementary Figure 5A-5G). This benefit alone was too big to be ignored!

With additional somatic SV analysis using both short-read and long-read sequencing data, we identified a set of somatic SVs that were not detected with GRCh38 reference due to the missed sequences in those SV regions (Figure 6/7/8, Supplementary Figure 9A/9B/12A-12E), and some of them involve pathways relevant to tumor development and metastasis (Figure 6E/8C). With all these newly identified personalized somatic SNVs and SVs using HCC1395BL_v1.0 as reference, which were **not “abstract” at all**, thus, we disagree that “the real outcome is that SD is a bit lower and read mapping has improved”,

rather, these newly found somatic SNVs/SVs would open a door for researchers or clinicians to investigate further, in line with the spirits of development of personalized or individualized precision medicine.

My current concerns (occurring from incomplete analyses or unaddressed) include:

1. Benefits of a personal genome are now presented in greater detail. However, the points highlighting those are currently dispersed all over the text as written and a reader must dig for it. A succinct summary in a tabular form (column 1 - benefit; column 2 - short proof, maybe some numbers) would greatly benefit the paper in such consideration. Authors’ answers to my Q2.8, with a bit of granularity would work fine there.

[Response]

Thank you for your great suggestion. We have made a supplementary table (**Supplementary Table 13**) as below and **Discussion** section, briefly summarizing all the benefits of using personalized genome as reference as compared to the standard GRCh38 reference.

		Benefits of using personalized genome assembly as reference	Proofs
Personalized genome assembly		Inclusion of the individual specific haplotypes, and better representations for the clinically important genes; no need to deal with ALT loci in NGS secondary analysis.	Figure 3B; Supplementary 9A/9B; Supplementary Figure 12A/12B/12C/12D/12E
Read mapping	Illumina short-reads	Smaller standard deviations for both read coverages and library-insert sizes; fewer mismatches; fewer improperly paired reads; fewer split-reads; fewer soft-/hard-clipped reads	Figure 4; Supplementary Figure 4
	PacBio long-reads	Higher numbers of reads being mapped; fewer mismatches; smaller standard deviations of read coverages with long reads more uniformly mapped	Figure 4; Supplementary Figure 4
Somatic SNVs detection	Illumina short-reads	Total somatic SNV counts increased; novel SNVs discovered and some were confirmed by Sanger sequencing	Figure 5; Supplementary Figure 5; Supplementary Figure 6
Somatic SVs detection	Illumina short-reads	Total somatic SNV counts increased; novel SVs discovered	Figure 6; Figure 7; Supplementary Figure 8A/8B/8C; Supplementary Figure 9A/9B; Supplementary Figure 10
	PacBio long-reads; Assembled contigs	Total somatic SV counts increased; novel SVs discovered	Figure 8; Supplementary Figure 10; Supplementary Figure 11; Supplementary Figure 12A/12B/12C/12D/12E

2. One of the major concerns I had - would such personal genome **improve SV detection** in tumor with short reads? It is addressed to some extent but not completely and the confident tone is over the top. Eg: “Our findings indicated that use of a personal genome as reference had great impacts on SV discovery...” I don’t think the effects shown (like in Fig. 6) are particularly GREAT impacts, I’d say there is SOME impact, even in the ever so numerous calls of BreakDancer. As DEL is the easiest SV, the fact that “the numbers of somatic deletion calls were similar” for BreakDancer in both references argues against so called “great impacts”. “Small but consistent improvements” would be a more fitting expression. **If such personalization would allow for finding variants that could not be found otherwise or clarifying something of the kind – that would be a demonstrable impact.** Especially, if we’re talking about clinical use (and personal reference makes me imagine a patient) – imagine talking to cancer patients about SD of their mapped reads?

[Response]

Thank you very much for your comment, and we have made appropriate modifications regarding the expression of improvements. Clearly, the impacts on “germline” SVs were much greater with HCC1395BL_v1.0 than somatic SVs in terms of the numbers of SVs being detected. However, as suggested by another reviewer, we have dropped both BreakDancer and Pindel results, as BreakDancer is a germline caller that its sensitivity and accuracy depends on read length and library insert sizes. By subtracting SVs in normal sample from SV calls in tumor sample, it may not produce ideal somatic calls. When other modern somatic SV callers are available, e.g., GRIDSS2 and manta, we should use them instead. In this revision, we have included the somatic SV calls by GRIDSS2/GRIPSS and Manta for all 12 pairs of tumor-normal replicates. Our latest somatic SV results showed that a set of somatic SVs that would not be detected with GRCh38 due to the missed sequences in those SV regions (**Figure 6/7/8, Supplementary Fig. 9A/9B/12A-12E**), and some of them involve pathways relevant to tumor development and metastasis (**Figure 6E, 8C**).

Furthermore, as demonstrated in the previous version of manuscript, the differences of somatic SNVs between GRCh38 and personalized genome reference (HCC1395BL_v1.0) were almost 10%, and the exact SNVs were 88.25%% (**Figure 5C**), not mentioning there were additional HCC1395BL_v1.0 specific somatic SNVs, some of which involve important pathways (**Figure 5D, Supplementary Table 6, Supplementary Figure 5A-5G**).

With all these newly identified personalized somatic **SNVs and SVs** using HCC1395BL_v1.0 as reference, our findings really provide more choices for researchers or clinicians to look into further for personalized patient care.

3. Equally troubling is the statement “undoubtedly demonstrated the benefits of using the personal genome HCC1395BLv1.0 in reducing potential false positives” – how could you say this w/o even knowing the false positives? Any examples, maybe illustrated or best – validated? And, hopefully, not with the IGV. Strictly speaking, manual inspection in IGV (unless mapping is really poor) is not different from reading the contents of VCF files, except a graphical viewer is used instead of a text editor.

[Response]

Thank you for your comment. To avoid confusion, we have removed those statements that related to germline callers (Pindel and BreakDancer), which produced significantly more “germline” SVs with the GRCh38 reference. That was why we stated “using the personal genome HCC1395BLv1.0 in reducing potential false positives” in the previous version of the manuscript. Perhaps, “noisy calls” may be better term than “false positives”.

As mentioned, we now used four “somatic” SV callers, including GRIDSS2 (split-reads + read-pairs + breakpoint assembly + GRIPSS somatic filtering), Manta (split-reads + read-pairs + breakpoint assembly), Delly (read-pairs then split-reads), and novoBreak (somatic kmer identification) to represent 4 different somatic SV detection algorithms, we no longer need to deal with the custom filtering when a germline caller was used for somatic SVs. Our latest somatic SV results showed that a set of somatic SVs that were not detected with GRCh38 due to the missed sequences in those SV regions (**Figure 6/7/8, Supplementary Figure 9A/9B/12A-12E**), and some of them involve pathways relevant to tumor development and metastasis (**Figure 6E/8C**).

With respect to VCF vs IGV, we have slightly different opinions – any SV caller could generate a mutation report in VCF format. But the problem is each caller would have its own false positive rate or false negative rate (saying some “truth” set there), particularly for those SVs in repeats or segdup regions, where the short-read mappings and SV calling could have issues. By using a graphical viewer (e.g., IGV) to upload the VCF file along with the read alignment evidence, we would be able to evaluate or exam if the SV caller makes right calls in the region. The NIST-led Genome-in-a-Bottle consortium had used such a strategy to curate germline SVs, particularly those SVs with medically relevant genes [Chapman LM et al. A crowdsourced set of curated structural variants for the human genome. **PLoS Comput Biol.** 2020 Jun 19;16(6); Wagner J et al. Curated variation benchmarks for challenging medically relevant autosomal genes. **Nat Biotechnol.** 2022 May.]

4. One possibility to address points 2 and 3 and add flavor to the story could be: take the SV calling results and show how they may incorrectly predict a potentially relevant deletion with a standard reference but that such result could be avoided with HCC1395BLv1.0, thus avoiding a wrong treatment option. An IGV illustration would work fine here, actually.

[Response]

Thank you very much for your suggestion. This was exactly what we have done in this revision. Our latest somatic SV results using both short-reads and long-reads showed that a set of somatic SVs that were not detected with GRCh38 due to the missed sequences in those SV regions (**Fig. 6/7/8, Suppl. Fig. 9A/9B – IGV snapshots, Suppl. Fig. 12A-12E –IGV snapshots**), and some of them involve pathways relevant to tumor development and metastasis (**Figure 6E, 8C**).

With all these newly identified personalized somatic **SNVs and SVs** using HCC1395BL_v1.0 as reference, our findings really can provide more personalized SNVs/SVs target choices for researchers or clinicians to look into further for personalized patient care, avoiding wrong treatment options (based on those GRCh38-specific SNVs/SVs), or even missing an interventional target specific treatment option (those HCC1395BL_v1.0 specific SNVs/SVs).

5. Further, the authors have improved the presentation of the long-read results but refuse to compare them with short reads. Such comparison is important, as it also would address a potential recommendations (or just thoughts) – should one use short or long reads tech in such cases? Or both? And next – what variant calling tools would be advisable? And possibly – what to watch for, what pitfalls to avoid? All this can be expressed rather succinctly and based on the current observations with the variant callers used here.

[Response]

Thank you for your comments. Personalized genome as reference would improve short-/long-read mapping, and subsequent somatic SNV/SV mutation detection as we have illustrated in our manuscript (**Figure 4, Figure 5/6/7/8**). But using a personalized genome as reference would not imply other researchers to use one sequencing technology over the other one for mutation detection. In this revision, we did the comparison as described in the **Results** section (**Supplementary Figure 13C**). Our analysis showed that both short-reads and long-read technologies were still needed for more complete somatic SV detection even with personalized reference, because of the underlying differences in sequencing technologies, sequencing coverages, region complexities, mapping qualities, and their detecting algorithms ((**Supplementary Figure 13C**)).

The natures/differences of the cancer cell lines being studied may also need to be considered. For example, GRIDSS2 was presented as the “best” somatic caller and novoBreak the worst one when they were applied to the COLO829/ COLO829BL tumor-normal cell lines (<https://genomebiology.biomedcentral.com/articles/10.1186/s13059-021-02423-x>), but this was not the case in our study. Apparently, Manta SV caller outperformed GRIDSS2, and novoBreak outperform Delly (**Figure 6A/6B**). Moreover, each caller has its own sensitivity for SV detection. For example, one 311 basepair deletion (scaffold_17:32976348-32976659 overlapping with CCDC91 gene that enables identical protein binding activity) detected with HCC1395BL_v1.0 reference only (but not with GRCh38 due to missing of this *A/u* sequence) by 3 callers (Manta, Delly, and novoBreak) with PacBio in-read deletions supports, but not by GRIDSS2 (see **IGV snapshot below**). Therefore, we feel that it is not a good idea for us to recommend any particular SV caller for the best use for all situations, because this is really not our study goal.

However, we did **include some general** evaluations (or suggestions, if you would agree) for short-read SV callers in the main text of our manuscript, which is pretty much in line with what you have suggested here – “We also observed that the total SV counts detected by both GRIDSS2 and Manta were higher than that by Delly and novoBreak, indicating that the callers with more sophisticated algorithms (such as GRIDSS2 and Manta that combined split-read, read-pair, and breakpoint assembly approaches) were likely more sensitive than relatively simpler callers (such as Delly and novoBreak) (**Figure 6A**). In particular, Delly reported much fewer DUP and TRA events than any of other callers, suggesting Delly may have lower sensitivity for detecting these two SV types (**Supplementary Figure 8B**).” At least, readers will have some idea regarding these callers being used in this study.

For long-read SV calling, we also included some general recommendation in **Methods** section (**Page 38**) by not just using a single pipeline for PacBio subreads SV calling to avoid aligner/caller bias – “We noticed that there were certain differences in their alignments with two aligners (pbmm2 vs. NGMLR) using PacBio subreads, thus, resulting SV calls (PBSV vs. Sniffles2) were different in some regions. Therefore, to minimize aligner and caller bias and maximize callers’ concordance in subsequent merged callset, in our PacBio long-read SV analysis, Sniffles2 was also applied to PacBio pbmm2 bams, and PBSV was applied to NGMLR bams. SVs were called jointly with both tumor and normal sample together. Therefore, for each genome reference, four VCFs (pbmm2+PBSV, pbmm2+Sniffles2, ngmlr+PBSV, and ngmlr+Sniffles2) were generated for downstream analysis.” In fact, we found that in merged somatic SV callset based on 4 calling methods with HCC1395BL_v1.0 reference, the overall concordance rate between PBSV and Sniffles2 was improved as compared to that based on two calling methods (e.g., using two callers on one aligner’s bam).

6. Figures would strain the readers. I had a hard time reading the axis labels of **Fig. 6**. for example. Fonts are often so puny, they are unreadable, **legends aren't very descriptive. Fig. 1/Table 1 legends are particularly laconic**, and more details would be great.

[Response]

Thank you for the suggestion and we have made appropriate changes with updated figures and have included more detailed descriptions.

7. They also didn't address Q1.7. Let me clarify. The authors underscore the importance of the correct personal assembly in complex regions. They also mention in response to Reviewer 2 the relevance of HLA mutations to cancer. Was anything detected in this region in the tumor sample with short reads? A measure of improvement compared to using the standard reference would be illustrative.

[Response]

Thank you for your comment. We did not address your original question in previous round review (Q1.7 – “Would HLA change be detectable with short reads?”), because it was not that clear to us – we did not know what you meant by “HLA change” in your question. But now this question has become much more clear, in fact, this is a great question, as it is relevant to assemble complex regions (i.e., HLA) and mutation detection in this region. Unfortunately, this HLA region was located at chromosome 6p on GRCh38 (corresponding scaffold_30 on HCC1395BL_v1.0), which was considered lost one copy for this HCC1395BL cell line based on the Karyotyping of HCC1395BL cell line [Extended Data Fig. 5: Karyotyping of HCC1395 and HCC1395BL, <https://www.nature.com/articles/s41587-021-00993-6/figures/9>, from Fang, L.T., et al., *Establishing community reference samples, data and call sets for benchmarking cancer mutation detection using whole-genome sequencing*. **Nat Biotechnol**, 2021. **39**(9): p. 1151-1160]. Thus, **the region of HLA for this normal cell line was essentially haploid (Results, Page 10)**. There were some mutations detected in HLA region with short-reads on both genome references, but the accuracy of the somatic mutations in these **haploid regions** was questionable, therefore, they were all excluded from analysis and comparison (Fang, L.T., et al., *Establishing community reference samples, data and call sets for benchmarking cancer mutation detection using whole-genome sequencing*. **Nat Biotechnol**, 2021. 39). We followed this and excluded these mutations in haploid regions from our analysis too.

However, such improvements, particularly in those HCC1395BL_v1.0 reference specific SNVs/SVs sets as described in **Results** section, have been captured in our SNV/SV somatic mutation analysis already using short- and long-reads. These results actually demonstrated that correct assembling of a personalized genome, whether it is from complex regions or not, was all important for accurate somatic SNV/SV mutation detections. We believe such results were what you've been looking for since the first round of review.

For example, one novel SNV exists in GTF2H2 (General transcription factor IIH subunit 2) gene that encodes the subunit of RNA polymerase II transcription initiation factor IIH, involving in both basal transcription and nucleotide excision repair (**Supplementary Table 7**). Another novel SNV in PTPN13 (Protein tyrosine phosphatase non-receptor type 13) gene that encodes a signaling molecule that belongs to the protein tyrosine phosphatase (PTP) family, regulating a variety of cellular processes such as cell growth, differentiation, mitotic cycle, and oncogenic transformation (**Supplementary Table 7**).

A 311 base pair *SINE/Alu* deletion (scaffold_17:32976348-32976659), which overlaps with CCDC91 (coiled-coil domain containing 91) gene enabling identical protein binding activity, was discovered with

HCC1395BL_v1.0 reference, but not with GRCh38 with the same set of reads due to its absence of this *Alu* sequence (**Supplementary Figure 9A**). A 57 base pair deletion (scaffold_6:44613899-44613956), which overlaps with MED12L (mediator complex subunit 12L) gene involving in transcriptional coactivation of nearly all RNA polymerase II-dependent genes, was detected with HCC1395BL_v1.0 reference, but only a 40 base pair deletion was discovered with GRCh38 using same set of reads due to 17 more adenine (A) nucleotides in this deletion region on HCC1395BL_v1.0 reference (**Supplementary Figure 9B**).

For example, CDH23 (Cadherin related 23) gene, belongs to the cadherin superfamily that encodes calcium dependent cell-cell adhesion glycoproteins. This gene has been reported to play a role in early stages of tumor metastasis through regulating cell-cell adhesion, and upregulation of CDH23 may be associated with breast cancer [36, 37]. A 327 bps homozygous deletion (scaffold_12:20762508-20762835) that overlaps with CDH23, was uncovered in tumor cell line when HCC1395BL_v1.0 was used as reference, which includes a copy of *SINE/AluY* (284 bps), but this *AluY* sequence was not present in GRCh38 thus, mapping the same set of tumor reads to GRCh38 would not identify this deletion (**Supplementary Figure 12A**).

A 128 base pairs homozygous deletion (scaffold_24:40364012-40364140, overlapping with an *LTR/ERVL* repeat) was uncovered in tumor cell line with HCC1395BL_v1.0 reference, but not with GRCh38. This deletion was located in intronic region (exon1 and exon2) of ST14 (ST14 transmembrane serine protease matriptase) gene (**Supplementary Table 12, Supplementary Figure 12D**), which encodes an epithelial-derived, integral membrane serine protease that functions as an epithelial membrane activator for other proteases and latent growth factors. Studies have associated the expression of this protease with breast, colon, prostate, and ovarian tumors, implicating its role in cancer invasion, and metastasis.

The above examples evidently demonstrated the importance of not only correctly assembling the personalized genome, but also using the personalized genome as reference for accurately detecting somatic SNV/SV mutations with short-read sequencing data in tumor-normal cell lines.

Second round of review

Reviewer 3

I'll go over my previous points. Many of them have been addressed but some concerns still remain and require another, smaller revision.

Point 1 - summary table.

The authors have now provided a Supplementary Table 13 and Supplementary data and it has helped the readability paper. Yet it will be better placed in the main text, rather than tucked somewhere to the end of Supplement.

Supplementary Table 13 "Proofs column". There are some presentation issues.

Personal genome assembly - these are all fine examples. A few problems with these examples:

Fig. 3 caption - "Although scaffold_30 maps onto GRCh38 entirely in reverse complement, but..." Use of although and but in the same sentence.

Supp Fig 9A - deletions are easier to interpret when the reads are shown as pairs. There are usually no size limitations in Supplementary Materials so compactness of reads should not be an issue. Reads are shown as paired in Supp Fig 12A-E and the same should be adopted for 9A.

Read mapping - Difficult to confirm due to graph quality issues:

Figure 4 axis labels nearly impossible to read even after zooming in making it difficult to interpret the graphs.

Supplementary Figure 4 axis labels are blurry at actual size.

Somatic SV detection - figures 6a - d support proofs

Figure 6c and 6d labels difficult to read. Figure 6e axis labels difficult to read.

Supplementary Figure 8b labels and legend difficult to read. Supplementary Figures 10 and 11 legends a bit blurry.

It would be helpful to include in the caption that Supplementary Figures 9A - B are from Illumina short reads and Supplementary Figures 12 A-E are from PacBio long reads. One can figure this out from this table or search the text but not find this out if just looking over the figures in Supplementary Materials.

Supplementary Table 13 - further comments. However, numerical values of the improvements would be helpful in the previous column. I suggested that as a "maybe" earlier and the authors didn't do it, but it looks more certain that it is important for an overall picture. Here's where the benefit "big enough to be ignored" could find its place, too. Maybe in conjunction with the next point?

Point 2 - improve SV detection in short reads.

In the response several Figures are referenced and some of these indeed do show an improvement in SV detection using HCC1935BL compared to GRCh38. However, in Fig. 6A and 6B these improvements appear to be very small. There are a lot of numbers given in pages 19 - 22 about the improvements, for example "increases of 34 (4.17%), 50 (5.78%), 34 (7.21%), and 11 (2.09%) SVs for GRIDSS2, Manta, Delly...". The problem is that once again the reader must dig through the text to understand these improvements. A table that summarizes these numbers would be more useful than just embedding the numbers in the text. For example, you could reference Fig. 6A and a table with the numbers rather than putting the numbers in the text. For short reads, examples of finding SVs in HCC1935BL that would not be found using GRCh38 are unclear. It appears most of the examples given (for example Suppl. Fig. 9A/9B,

12A-E) are from PacBio long reads. The captions should include whether the reads are from short reads or long reads. It is difficult for the reader to figure out how many SVs were detected in HCC1935BL that would not be found using GRCh38 and how many of these are significant. In general, it would be good to clarify the improvements in SV detection in short reads by using a table similar to (or maybe combined with) Supplementary Table 13..

In the Discussion it states that "even with short-read sequencing data...the personalized reference enabled us to identify additional somatic SNVs that could not be detected on the GRCh38 reference..." pg 30 line 43. This is a potentially major selling point that should be expanded upon and something that you would want to tell a patient, as I alluded to in an earlier review (or maybe doctor/insurance company, too).

Point 4 - Some of paragraph 2 of their response should probably be added to the Discussion to fully address this point. Problems with captions for Suppl. Fig. 9A/9B, 12A-E previously mentioned.

Other previous points have been adequately addressed.

New point 5: However, an important extra concern has recently appeared - related to a new "elephant in the room" - telomere-2-telomere (T2T) genome. The authors should speculate in the Discussion, how using T2T could change the balance they've observed here with personal vs. GRCh38? How much the benefits reported here could decrease (or would they)? I'm sure a reader would want to know.

Minor comments:

Other comments of Supplementary Material Figures:

The figures are very inconsistent in layout and quality. Fonts and quality vary widely. Some figures have titles on top, other do not. Legends are on top in some cases and at bottom in others. Figures 1 - 2 legends are easy to read. Figure 3 axis labels are a bit small.

Figure 5G difficult to read the BLAST results since the screen capture was likely squeezed in without preserving the aspect ratio.

Reviewer reports and point-by-point responses for GBIO-D-21-00720R2

Reviewer #3:

Reviewer #3: I'll go over my previous points. Many of them have been addressed but some concerns still remain and require another, **smaller revision**.

[Response]

We really appreciate your great comments that has helped improve our manuscript greatly. Our responses to your comments or suggestions are listed below.

Point 1 - summary table.

The authors have now provided a Supplementary Table 13 and Supplementary data and it has helped the readability paper. Yet it will be better placed in the main text, rather than tucked somewhere to the end of Supplement.

[Response]

Thank you for the suggestion. Now a new main table called "Table 2" has been created to supersede the previous "Supplementary Table 13" that summarized all the benefits using personalized assembly as reference for somatic mutation detection in tumor-normal paired samples as compared to GRCh38. This Table is now part of the Main text of this manuscript.

Supplementary Table 13 "Proofs column". There are some presentation issues.

Personal genome assembly - these are all fine examples. A few problems with these examples:

Fig. 3 caption - "Although scaffold_30 maps onto GRCh38 entirely in reverse complement, but..." Use of although and but in the same sentence.

[Response]

Thank you very much for your careful review. This typo has been fixed now in the revised version.

Supp Fig 9A - deletions are easier to interpret when the reads are shown as pairs. There are usually no size limitations in Supplementary Materials so compactness of reads should not be an issue. Reads are shown as paired in Supp Fig 12A-E and the same should be adopted for 9A.

[Response]

Thank you for the suggestion. We have re-made Fig S9A so that the reads from Illumina short-reads and PacBio long-reads are clearly supporting the detection of this deletion. Please note that this deletion is "heterozygous", not like other examples which are all "homozygous", thus we do not expect those IGV snapshots to look "similar". To clarify a little further regarding this deletion, we provided two additional IGV snapshots (one with all Illumina short-reads, and the other one with all PacBio long reads for the verification purpose) to show the read supporting evidence for this heterozygous deletion in this region.

Read mapping - Difficult to confirm due to graph quality issues:

Figure 4 axis labels nearly impossible to read even after zooming in making it difficult to interpret the graphs.

Supplementary Figure 4 axis labels are blurry at actual size.

Somatic SV detection - figures 6a - d support proofs Figure 6c and 6d labels difficult to read. Figure 6e axis labels difficult to read.

Supplementary Figure 8b labels and legend difficult to read. Supplementary Figures 10 and 11 legends a bit blurry.

[Response]

We apologize for the difficulty of reading, and this was caused because we made the image files inserted into the main manuscript for the easy of peer review. Individual figure files actually do have high resolution. However, Figure 4, Fig. S4, Figure 6A-D, Fig S8B, Fig. S10/S11 have all been re-created with high resolution and are included in this submission.

It would be helpful to include in the caption that Supplementary Figures 9A - B are from Illumina short reads and Supplementary Figures 12 A-E are from PacBio long reads. One can figure this out from this table or search the text but not find this out if just looking over the figures in Supplementary Materials.

[Response]

Thank you for another great suggestion. Yes, we have added Illumina short-reads to the caption for Fig. 9A/9B, and PacBio long-reads to the caption for Fig. S12A-E.

Supplementary Table 13 - further comments. However, numerical values of the improvements would be helpful in the previous column. I suggested that as a "maybe" earlier and the authors didn't do it, but it looks more certain that it is important for an overall picture. Here's where the benefit "big enough to be ignored" could find its place, too. **Maybe in conjunction with the next point?**

[Response]

Thank you very much. Yes, as mentioned earlier, a new main table called "Table 2" has been created to supersede the previous "Supplementary Table 13" that summarized all the benefits using personalized assembly as reference for somatic mutation detection in tumor-normal paired samples as compared to GRCh38. We have added detailed numbers in the "column 2" with respective proofs in the "column 3" as suggested in conjunction with your comments regarding the "short-reads SV improvement".

Point 2 - improve SV detection in short reads.

In the response several Figures are referenced and some of these indeed do show an improvement in SV detection using HCC1935BL compared to GRCh38. However, in Fig. 6A and 6B these improvements appear to be very small. There are a lot of numbers given in pages 19 - 22 about the improvements, for example "increases of 34 (4.17%), 50 (5.78%), 34 (7.21%), and 11 (2.09%) SVs for GRIDSS2, Manta, Delly...". The problem is that once again the reader must dig through the text to understand these improvements. A table that summarizes these numbers would be more useful than just embedding the numbers in the text. For example, you could reference Fig. 6A and a table with the numbers rather than putting the numbers in the text.

[Response]

Yes, great suggestion. We have added two new supplementary tables (Table S9 and Table S10) to summarize these respective improvements and referred these tables in the main text as suggested, thus reducing many "boring" numbers in the main texts.

For short reads, examples of finding SVs in HCC1935BL that would not be found using GRCh38 are unclear. It appears most of the examples given (for example Suppl. Fig. 9A/9B, 12A-E) are from PacBio long reads. The captions should include whether the reads are from short reads or long reads. It is difficult for the reader to figure out how many SVs were detected in HCC1935BL that would not be found using GRCh38 and how many of these are significant.

In general, it would be good to clarify the improvements in SV detection in short reads **by using a table similar to (or maybe combined with) Supplementary Table 13.**

[Response]

As mentioned earlier, we have clarified the underlying reads in the captions for Fig. S9A/B and Fig. S12A-E. The examples from Fig. S12A-E are indeed from PacBio long-reads, but examples from Fig. S9A/B are actually from Illumina short-reads, within which the PacBio long-reads are shown for verification purposes of the detected short-read deletions.

In the main text, it is our intention to illustrate the benefits of using personalized assembly reference for short-reads in Figure 6 (as well as Figure 7). Figure 6D shows novel SVs detected (59) requiring support

by at least 2 callers, while Figure 7D shows novel SVs detected by each of the 4 short-read somatic callers requiring support by at least 2 replicates.

Creating a separate/similar table to just summarize short-read SVs improvement is redundant to the effort of documenting all the benefits of using personalized assembly as reference for somatic mutation detection. We felt that your alternative suggestion for combining with previous Suppl. Table 13 (now main Table 2 below) would be better option. Clearly, now it is much easier for readers to grasp the messages, regarding, e.g., how many somatic SVs were detected in HCC1935BL_v1.0 that would not be found using GRCh38 with short-reads (59, including 17 gene-overlapping SVs, e.g., CCDC91), PacBio long-reads (279, including 91 gene-overlapping SVs, e.g., CDH23, ST14, GNG7), somatic SNVs using short-reads (1,017, 177 overlapping with 71 genes, e.g., GTF2H2 and PTPN13).

		Benefits of using personalized genome assembly as reference	Proofs
Personalized genome (PG) assembly		 Individualized assembly with inclusion of the individual specific haplotypes, and better representations for the clinically important genes; No need to deal with ALT loci in NGS secondary analysis. 	 Figure 3B, Fig. S5A-5G, Fig. S9A/9B, Fig. S12A-12E; HLA genes, GSTT1, KIR2DL5A;
Read mapping	Illumina short-reads	 More properly-paired reads (HCC1395BL 0.5%, HCC1395 0.46%), fewer improperly paired reads (HCC1395BL 41.4%, HCC1395 38.2%); Fewer mismatches (HCC1395BL 18.2%, HCC1395 16.6%%); Fewer soft-clipped reads (HCC1395BL 11.7%, HCC1395 11.6%%), fewer hard-clipped reads (HCC1395BL 32.0%, HCC1395 28.7%); Fewer split-reads (HCC1395BL 31.9%, HCC1395 28.8%); Better read placements with smaller standard deviations for library-insert sizes (HCC1395BL 2.76, HCC1395 2.83); More uniformly read placements with smaller standard deviations for read coverages (HCC1395BL 4.31, HCC1395 4.92); 	 Figure 4A/4B; Figure 4C; Fig. S4A/S4B; Figure 4D; Fig. S4C; Figure 4E;
	PacBio long-reads	 Higher numbers of reads being mapped (HCC1395BL 1.65%, HCC1395 2.98%); Fewer mismatches (HCC1395BL 1%, HCC1395 1%); Lower non-primary/supplementary alignments (HCC1395BL 6.73%/14.5%, HCC1395 1.8%/10.39%) More uniformly read placements with smaller standard deviations for read coverages (HCC1395BL 12.08, HCC1395 11.48); 	 Fig. S4D; Fig. S4D; Fig. S4D; Figure 4F;
Somatic SNVs detection	Illumina short-reads	 Total somatic SNV counts increased by, on average, 1,689 SNPs and 415 InDels; Novel SNVs discovered (1,017), 177 overlapping with 71 genes, e.g., GTF2H2 and PTPN13, and some were confirmed by Sanger sequencing (8 out of 10 selected SNVs); Context sequences of somatic SNVs more accurate, some with germline SNVs (3,995); Avoid GRCh38-only, non-personalized SNVs (901); 	 Figure 5A/5B, Table S5; Figure 5D, Table S6/S7/S8, Fig. S6; Figure 5C, Table S6, Fig. S5A-5G; Figure 5C;
Somatic SVs detection	Illumina short-reads	 Somatic SV counts increased by 82/GRIDSS2, 189/Manta, 54/Delly, and 86/novoBreak; Novel SVs discovered (59), including 17 gene-overlapping SVs, e.g., CCDC91; SV resolution more accurate, e.g., SV with MED12L gene; Avoid GRCh38-only, non-personalized SVs (29); 	 Figure 6A, Table S9/S10, Figure 7A, Fig. S13A/13B; Figure 6D/6E, Table S11, Fig. S9A/9B, Figure 7D; Fig. S9B; Figure 6C, Figure 7C;
	PacBio long-reads; Assembled contigs	 Somatic SV counts increased by 194 (with supports by 2 or more calling methods); Novel SVs discovered (279), including 91 gene-overlapping SVs, e.g., CDH23, ST14, GNG7; SV resolution more accurate, e.g., SV with MED12L gene; Avoid GRCh38-only, non-personalized SVs (213); 	 Figure 8A, Fig. S10, Fig. S13A/13B; Figure 8C/8D, Table S14, Fig. S12A-12E; Fig. S9B; Figure 8B, Table S13;

In the Discussion it states that "even with short-read sequencing data...the personalized reference enabled us to identify additional somatic SNVs that could not be detected on the GRCh38 reference..." pg 30 line 43. This is a potentially major selling point that should be expanded upon and something that

you would want to tell a patient, as I alluded to in an earlier review (or maybe doctor/insurance company, too).

[Response]

This point seems to relate to your Point 4 below. We have added a paragraph to the **Discussion** section as suggested:

“Our approach using a personal assembly (HCC1395BL_v1.0) as reference identified many additional personalized somatic SNVs and SVs which were missed using GRCh38 as reference. These more personalized SNVs/SVs provide additional target choices for patients, researchers or clinicians, and physicians to look into further for personalized patient care. They may reduce the pursuit of incorrect treatment options based on GRCh38-specific or other non-personalized reference somatic SNVs/SVs, and missed opportunities for interventional target specific treatment options.”

Point 4 - Some of paragraph 2 of their response should probably be added to the Discussion to fully address this point. Problems with captions for Suppl. Fig. 9A/9B, 12A-E previously mentioned. Other previous points have been adequately addressed.

[Response]

Firstly, we have addressed the caption issues you mentioned earlier regarding “Suppl. Fig. 9A/9B, 12A-E”. Secondly, regarding adding “paragraph 2” of the previous response, this seems to be related to the point you mentioned immediately above. We have added a paragraph to the **Discussion** section as suggested:

“Our approach using a personal assembly (HCC1395BL_v1.0) as reference identified many additional personalized somatic SNVs and SVs which were missed using GRCh38 as reference. These more personalized SNVs/SVs provide additional target choices for patients, researchers or clinicians, and physicians to look into further for personalized patient care. They may reduce the pursuit of incorrect treatment options based on GRCh38-specific or other non-personalized reference somatic SNVs/SVs, and missed opportunities for interventional target specific treatment options.”

New point 5: However, an important extra concern has recently appeared - related to a new "elephant in the room" - telomere-2-telomere (T2T) genome. The authors should speculate in the Discussion, how using T2T could change the balance they've observed here with personal vs. GRCh38? How much the benefits reported here could decrease (or would they)? I'm sure a reader would want to know.

[Response]

We have added a paragraph in the end of the **Discussion** section as below:

“Recently, the Telomere-to-Telomere (T2T) consortium finished the first gapless telomere-to-telomere human genome assembly (T2T-CHM13) [40], and illustrated its advantages as a reference over GRCh38 for germline variant detection in population genetic analyses [41]. Theoretically, this new reference would improve somatic mutation detection as compared to GRCh38, but the extent of such improvements for somatic mutation discovery in tumor-normal samples has not yet been investigated. Some of the benefits we reported in this study may be impacted. For instance, read mappings to T2T-CHM13 are anticipated to be better than those to GRCh38, but a personal genome (especially a complete T2T personal genome) reference would still probably outperform T2T-CHM13. Although HCC1395 (<https://www.atcc.org/products/crl-2324>) is from a Caucasian sample and CHM13 is mostly of

European origin [40], there are likely some T2T-CHM13 specific somatic mutations that should be avoided, as well as some personal genome specific somatic mutations that we would like to use as additional choices for personalized patient care and precision oncology medicine. If the ultimate goal of our patient care is individualized or personalized, then use of a personalized assembly rather than GRCh38 or T2T-CHM13 as reference to identify the full spectrum of somatic mutations in tumor-normal samples is advocated.”

Minor comments:

Other comments of Supplementary Material Figures:

The figures are very inconsistent in layout and quality. Fonts and quality vary widely. Some figures have titles on top, other do not. Legends are on top in some cases and at bottom in others.

Figures 1 - 2 legends are easy to read. Figure 3 axis labels are a bit small.

Figure 5G difficult to read the BLAST results since the screen capture was likely squeezed in without preserving the aspect ratio.

[Response]

Yes, we have fixed the inconsistency in the revised version. We have recreated Figure 3, Fig. S5G and some other supplementary figures mentioned earlier so they are more clear and more consistent as possible.